# Targeted protein degradation via intramolecular bivalent glues

Oliver Hsia[1,8], Matthias Hinterndorfer[2,8], Angus D. Cowan[1,8], Kentaro Iso[1,3], Tasuku Ishida[1,3], Ramasubramanian Sundaramoorthy[4], Mark A. Nakasone[1], Hana Imrichova[2], Caroline Schätz[2], Andrea Rukavina[2], Koraljka Husnjak[5], Martin Wegner[5], Alejandro Correa-Sáez[1], Conner Craigon[1], Ryan Casement[1], Chiara Maniaci[1,6], Andrea Testa[1,7], Manuel Kaulich[5], Ivan Dikic[5], Georg E. Winter[2 ✉] & Alessio Ciulli[1 ✉]

Targeted protein degradation is a pharmacological modality that is based on the induced proximity of an E3 ubiquitin ligase and a target protein to promote target ubiquitination and proteasomal degradation. This has been achieved either via proteolysis-targeting chimeras (PROTACs)—bifunctional compounds composed of two separate moieties that individually bind the target and E3 ligase, or via molecular glues that monovalently bind either the ligase or the target[1–4]. Here, using orthogonal genetic screening, biophysical characterization and structural reconstitution, we investigate the mechanism of action of bifunctional degraders of BRD2 and BRD4, termed intramolecular bivalent glues (IBGs), and find that instead of connecting target and ligase in *trans* as PROTACs do, they simultaneously engage and connect two adjacent domains of the target protein in *cis*. This conformational change 'glues' BRD4 to the E3 ligases DCAF11 or DCAF16, leveraging intrinsic target–ligase affinities that do not translate to BRD4 degradation in the absence of compound. Structural insights into the ternary BRD4–IBG1–DCAF16 complex guided the rational design of improved degraders of low picomolar potency. We thus introduce a new modality in targeted protein degradation, which works by bridging protein domains in *cis* to enhance surface complementarity with E3 ligases for productive ubiquitination and degradation.

The cullin RING E3 ubiquitin ligase (CRL) substrate receptor DCAF15 has been utilized for the pharmacological degradation of the mRNA splicing factor RBM39 via the aryl sulfonamide molecular glues indisulam and E7820[5–9]. Efforts to leverage aryl sulfonamides as E3-binding ligands for PROTACs have so far met with limited success[10–12] (Extended Data Fig. 1a,b and Supplementary Fig. 1). However, a recent patent filing described a PROTAC-like degrader, referred to here as IBG1 (Fig. 1a and Supplementary Methods), which comprises the BET family bromodomain inhibitor JQ1 tethered to E7820. IBG1 results in potent BRD4 degradation (half-maximal degradation concentration (DC$_{50}$) = 0.15 nM) and pronounced growth inhibition in various cancer cell lines[13]. We synthesized IBG1 and confirmed efficient killing of diverse cell lines (Extended Data Fig. 1c) and BET protein degradation that was specific for BRD2 and BRD4 compared with their paralogue BRD3 (Fig. 1b,c, Extended Data Fig. 1d,e and Supplementary Table 1). The proteasome inhibitor MG132 and the neddylation inhibitor MLN4924 blocked BET protein degradation and/or ubiquitination (Fig. 1d,e and Extended Data Fig. 1f), indicating that IBG1 functions via CRL-mediated ubiquitination and proteasomal degradation. This degradation was unaffected by DCAF15 perturbation (Fig. 1f and

Extended Data Fig. 1g), suggesting an unexpected DCAF15-independent mechanism.

## IBG1 recruits DCAF16 to degrade BRD4

To identify the factors required for the activity of IBG1, we set up a series of CRL-focused fluorescence-activated cell sorting (FACS)-based BRD4 degradation CRISPR screens using a dual fluorescence BRD4 stability reporter (Fig. 2a, Extended Data Table 1 and Supplementary Fig. 2). In the DMSO control screen, we found that the 20S proteasome, the COP9 signalosome and the CRL3–SPOP complex controlled BRD4 stability, recapitulating the known endogenous BRD4 turnover machinery[14,15] (Fig. 2b and Supplementary Table 3). For MZ1, we identified subunits of the CRL2–VHL complex, consistent with the known engagement of VHL by MZ1[16] (Fig. 2b). When focusing on the genes required for BRD4 degradation by IBG1, we found that the compound functioned independently of DCAF15, in line with our previous observations. Instead, we identified members of the CRL4–DCAF16 complex, notably the CUL4A backbone, RBX1, the adapter DDB1 and the substrate receptor DCAF16, to be required for IBG1 function, as recently reported for the

[1]Centre for Targeted Protein Degradation, School of Life Sciences, University of Dundee, Dundee, UK. [2]CeMM Research Center for Molecular Medicine of the Austrian Academy of Sciences, Vienna, Austria. [3]Tsukuba Research Laboratory, Eisai Co., Ibaraki, Japan. [4]Centre for Gene Regulation and Expression, School of Life Sciences, University of Dundee, Dundee, UK. [5]Institute of Biochemistry II, Faculty of Medicine, Goethe University Frankfurt, Frankfurt am Main, Germany. [6]Present address: Medical Research Council (MRC) Protein Phosphorylation and Ubiquitylation Unit, School of Life Sciences, University of Dundee, Dundee, UK. [7]Present address: Amphista Therapeutics, Cambridge, UK. [8]These authors contributed equally: Oliver Hsia, Matthias Hinterndorfer, Angus D. Cowan. ✉e-mail: gwinter@cemm.oeaw.ac.at; a.ciulli@dundee.ac.uk

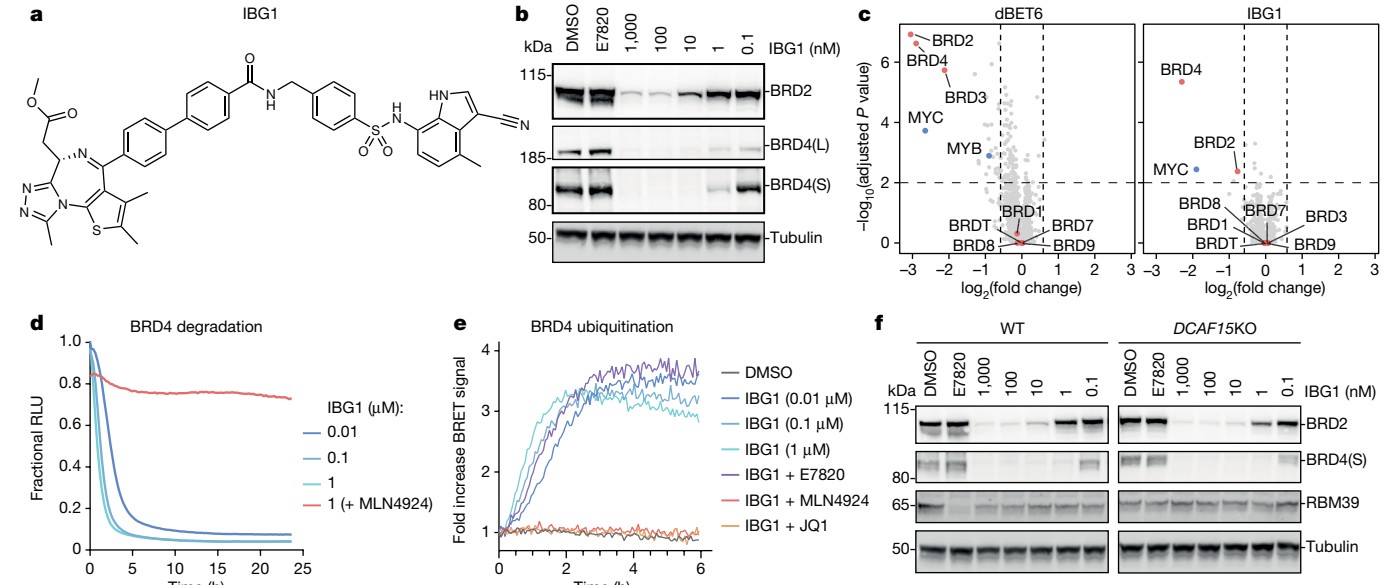

**Fig. 1 | IBG1 degrades BRD2 and BRD4 independently of DCAF15.**
**a**, Structure of IBG1. **b**, BET protein degradation activity of IBG1. HEK293 cells were treated for 6 h with DMSO, E7820 (1 μM) or increasing concentrations of IBG1. BET protein was quantified by immunoblot. Data representative of $n$ = 3 independent experiments. **c**, Whole-proteome changes after degrader treatment. Quantitative proteomics in KBM7 cells was performed after 6 h of treatment with DMSO, IBG1 (1 nM) or dBET6 (10 nM). $\log_2$-transformed fold change and $-\log_{10}$-transformed Benjamini–Hochberg adjusted one-way analysis of variance (ANOVA) $P$ value compared with DMSO treatment. $n$ = 3 biological replicates. **d**, NanoBRET kinetic degradation assay. BromoTag–HiBiT–BRD4

knock-in HEK293 cells were treated with IBG1 with or without MLN4924 (10 μM) pre-treatment for 1 h. Mean of $n$ = 3 biological replicates. RLU, relative light units. **e**, NanoBRET kinetic ubiquitination assay. LgBiT-transfected HiBiT–BromoTag–BRD4 knock-in HEK293 cells were treated with IBG1 at indicated concentrations or at 10 nM following pre-treatment with JQ1, E7820 (both 10 μM) or MLN4924 (1 μM) for 1 h. Mean of $n$ = 4 biological replicates. **f**, DCAF15-independent BET protein degradation. Wild-type (WT) and *DCAF15*-knockout (KO) HCT-116 cells were treated with increasing concentrations of IBG1 for 6 h and BET protein was quantified by immunoblot. Data representative of $n$ = 3 independent experiments.

monovalent BET degrader GNE-0011[17–20] (Extended Data Fig. 2a). We also found DCAF16 alongside the CUL4-associated ubiquitin-conjugating enzyme UBE2G1[21,22] as the top hits mediating resistance to IBG1 in an orthogonal viability-based CRISPR screen (Fig. 2c and Supplementary Tables 4 and 5).

In validation assays in KBM7 and HCT-116 cells, knockout or knockdown of CRL4–DCAF16 complex subunits prevented degradation of BRD4–BFP as well as endogenous BRD2 and BRD4 (Fig. 2d and Extended Data Fig. 2b–e), whereas ectopic expression of single guide RNA (sgRNA)-resistant DCAF16 restored degradation (Fig. 2e and Extended Data Fig. 2f). Finally, knockout of DCAF16 prevented the induction of apoptosis by IBG1 (Fig. 2f and Extended Data Fig. 2g) and led to enhanced tolerance of KBM7 cells (Fig. 2g), whereas IBG1 still induced pronounced MYC downregulation, in line with retained DCAF16-independent BET bromodomain inhibition by its JQ1 moiety (Extended Data Fig. 2g). Together, these data show that despite the incorporation of a DCAF15-targeting aryl sulfonamide moiety[9], IBG1 critically depends on the structurally unrelated CRL4 substrate receptor DCAF16 for BET protein degradation and cancer cell killing. We thus investigated a potential affinity of IBG1 for DCAF16. As expected, we observed dose-dependent binding of a fluorescein isothiocyanate (FITC)-labelled E7820 probe to recombinant DCAF15, whereas it showed no affinity for DCAF16 (Fig. 2h). Additionally, the presence of excess amounts of E7820 or sulfonamide-containing truncations of IBG1 (compounds **1a**–**d**) did not prevent BRD4 ubiquitination or degradation (Fig. 1e and Extended Data Fig. 3a). These results indicated that the IBG1 sulfonamide moiety is not involved in the recruitment of DCAF16 in a PROTAC-like manner. However, IBG1 fragments containing truncations of the sulfonamide moiety (compounds **1e**–**g**) did not promote BRD4 degradation despite efficient binding to BRD4 (Extended Data Fig. 3b,c), suggesting that the E7820 moiety is required for IBG1 activity in a role outside of direct E3 ligase recruitment.

## IBG1 enhances the affinity of DCAF16 for BRD4

We next sought to characterize the possible interactions between DCAF16, BRD4 and IBG1 in vitro. Using isothermal titration calorimetry (ITC), we observed the formation of a ternary complex between IBG1, DCAF16 and BRD4[Tandem], a BRD4 construct containing both bromodomains (BD1 and BD2) connected by the native linker (dissociation constant ($K_d$) = 567 nM; Fig. 3a). Similarly, a time-resolved fluorescence resonance energy transfer (TR-FRET) complex-formation assay showed that a ternary complex formed between DCAF16 and BRD4[Tandem] in a dose-dependent manner upon IBG1 titration (half-maximal effective concentration ($EC_{50}$) = 44 nM; Fig. 3b). A complementary TR-FRET-based complex-stabilization assay confirmed an interaction upon titrating DCAF16 into BRD4[Tandem] in the presence of IBG1 ($K_d$ = 712 nM; Fig. 3c). Unexpectedly, we also observed an intrinsic affinity of DCAF16 to BRD4[Tandem] in the absence of IBG1 using TR-FRET ($K_d$ = 1 μM; Fig. 3c) and ITC ($K_d$ = 4 μM; Extended Data Fig. 3d). No such intrinsic affinity was observed with isolated BRD4–BD1 or BRD4–BD2 (Fig. 3c). Comparison of the ITC titrations for DCAF16 into unbound versus IBG1-bound BRD4[Tandem] revealed that IBG1 strengthens ($K_d$ of 0.6 μM versus 4 μM) and thermodynamically alters the BRD4–DCAF16 interaction. Although IBG1 changes the binding from exothermic to endothermic (binding enthalpy ($\Delta H$) of −8 kJ mol$^{-1}$ versus 38 kJ mol$^{-1}$), this unfavourable enthalpy change is more than compensated for by a substantial change in the entropic term ($T\Delta S$), which becomes much more favourable in the presence of IBG1 ($T\Delta S$ of 22.5 kJ mol$^{-1}$ versus 73.9 kJ mol$^{-1}$). This enthalpy–entropy compensation, a well-known phenomenon in biological systems[23], leads to a greater binding energy ($\Delta G$) in the presence versus absence of IBG1 ($\Delta G$ of −35.7 versus −30.6 kJ mol$^{-1}$), resulting in a favourable binding energy change ($\Delta\Delta G$) of −5.1 kJ mol$^{-1}$. Together, these marked differences in thermodynamic behaviour are consistent with a different mode of DCAF16 binding for IBG1-bound compared

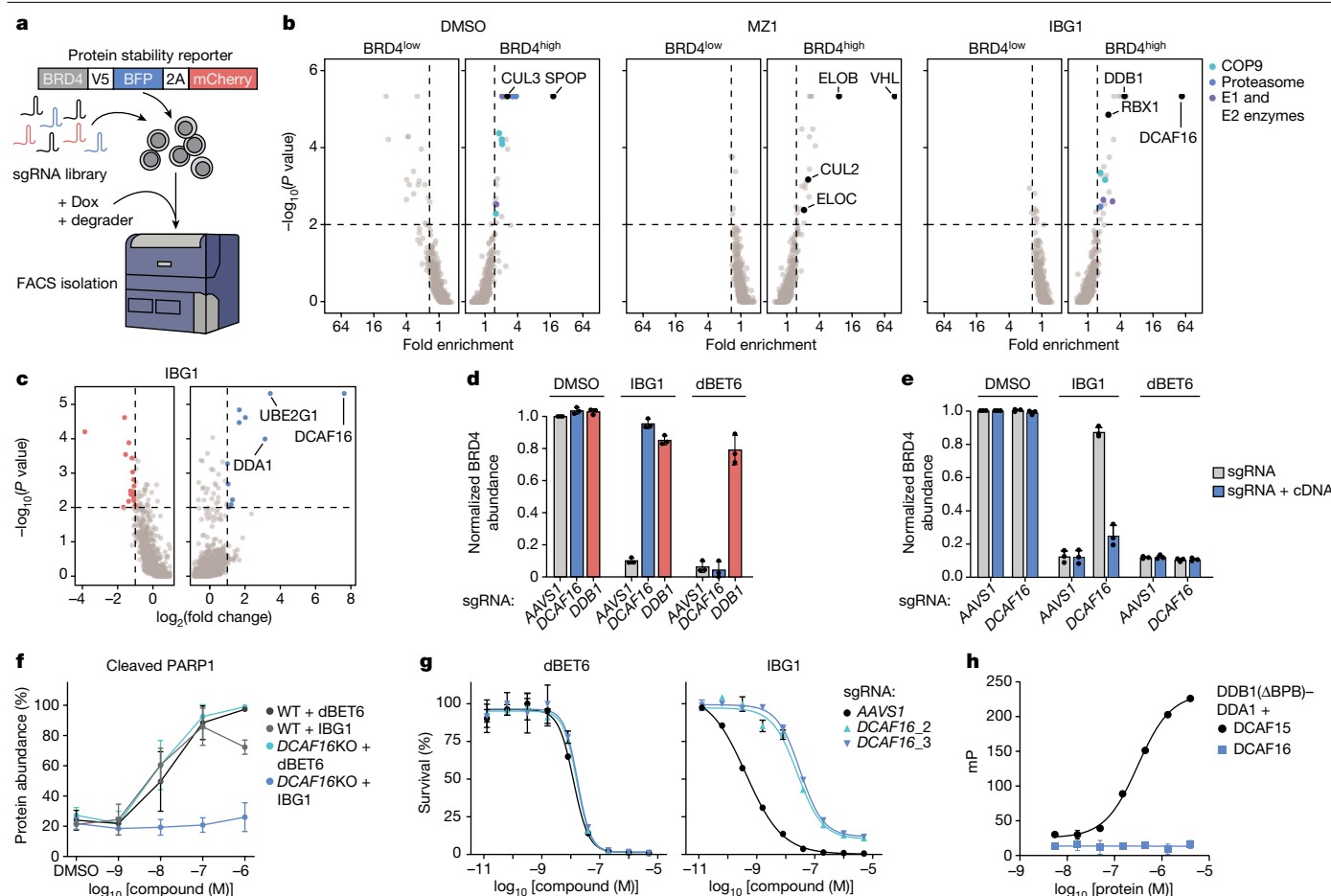

**Fig. 2 | IBG1-induced degradation of BRD2 and BRD4 is dependent on CRL4–DCAF16. a**, Schematic of FACS-based CRISPR–Cas9 screens. Doxycycline (Dox)-inducible Cas9 (iCas9) KBM7 BRD4–BFP reporter cells were transduced with a CRL-focused sgRNA library, treated with BET degraders and sorted based on BRD4–BFP/mCherry ratios. **b**, FACS-based CRISPR screens for BRD4 stability. KBM7 iCas9 BRD4 reporter cells were treated with DMSO, MZ1 (10 nM) or IBG1 (1 nM) for 6 h before sorting. 20S proteasome subunits, COP9 signalosome subunits and E1 and E2 ubiquitin enzymes inside the scoring window (one-sided MAGeCK $P$ value < 0.01, fold change > 1.5) are highlighted. **c**, CRISPR–Cas9 viability screen. HCT-116 cells were transduced with Cas9 and a ubiquitin–proteasome system-focused sgRNA library and treated with IBG1 (58 nM; fourfold half-maximal inhibitory concentration (IC$_{50}$)) for 6 days. Genes with a fold change > 2 and one-sided MAGeCK $P$ value < 0.01 are highlighted. **d**, Screen validation. KBM7 iCas9 BRD4–BFP reporter cells were transduced with *AAVS1*, *DCAF16* or *DDB1*-targeting sgRNAs, treated with

DMSO, IBG1 (1 nM) or dBET6 (10 nM) for 6 h, and BRD4–BFP was quantified by FACS. **e**, *DCAF16* knockout and rescue. KBM7 iCas9 BRD4–BFP reporter cells were transduced with *AAVS1* or *DCAF16*-targeting sgRNAs, with or without sgRNA-resistant *DCAF16* cDNA. After knockout of endogenous DCAF16, cells were treated for 6 h as in **b** and BRD4–BFP was quantified by FACS. **f**, Apoptosis induction. Wild-type or *DCAF16*-knockout KBM7 cells were treated with indicated concentrations of dBET6 or IBG1 for 16 h. Cleaved PARP1 was evaluated by immunoblotting. **g**, Viability assay. Wild-type or *DCAF16*-knockout KBM7 cells were treated with IBG1 or dBET6 for 72 h and cell viability was evaluated by CellTiterGlo assay. Mean ± s.d. of $n$ = 3 biological replicates. **h**, Fluorescence polarization binary binding assay. FITC-labelled sulfonamide probe (Supplementary Methods) was titrated into DCAF15–DDB1(ΔBPB)–DDA1 or DCAF16–DDB1(ΔBPB)–DDA1. DDB1(ΔBPB) lacks the cullin-binding domain (BPB). $n$ = 3 technical replicates. **d**–**f**, $n$ = 3 independent experiments. **d**–**h**, Mean ± s.d.

with unbound BRD4$^{Tandem}$. These observations were corroborated by size-exclusion chromatography (SEC) experiments, in which DCAF16 and BRD4$^{Tandem}$ co-eluted in the absence of compound and this interaction was stabilized by IBG1, whereas no interaction was observed with isolated BD1 and BD2 (Fig. 3d,e). In alphaLISA displacement assays, we found significantly enhanced affinity of IBG1 to BRD4$^{Tandem}$ in the presence of DCAF16 (IC$_{50}$ = 12.8 nM) compared with IBG1 and BRD4$^{Tandem}$ alone (IC$_{50}$ = 462 nM; cooperativity ($\alpha$) = 36; Extended Data Fig. 3e), further supporting a role of IBG1 in the formation of a tight BRD4–IBG1–DCAF16 ternary complex. Again, DCAF16 did not induce the binding of IBG1 to isolated BRD4–BD1, corroborating that both bromodomains are required for complex formation. Together, these orthogonal assays establish an intrinsic affinity between BRD4 and DCAF16, which is stabilized by IBG1 and requires the presence of both bromodomains.

To further explore the different behaviour of individual bromodomains and BRD4$^{Tandem}$, we focused on cellular assays based on the

BRD4–BFP reporter. We generated a panel of KBM7 cell lines stably expressing either wild-type or truncated reporters (Fig. 3f and Extended Data Fig. 3f) and assessed the degradation of these constructs by IBG1 or the CRBN-based PROTAC dBET6[24] using FACS. As expected, we observed potent degradation of wild-type BRD4 by both degraders. Deletion of the N-terminal phosphorylation site (NPS), basic residue-enriched interaction domain (BID), extraterminal (ET) and serine, glutamic acid and aspartic acid-rich region (SEED) domains did not affect degradation (Extended Data Fig. 3f) and BRD4$^{Tandem}$ was sufficient for degradation (Fig. 3f). Whereas isolated BD1 and BD2 were potently degraded by dBET6, we observed no degradation with IBG1 (Fig. 3f). Disruption of the JQ1 binding sites within the acetyllysine binding pockets in either bromodomain via single asparagine to phenylalanine changes (N140F or N433F, respectively) was sufficient to prevent degradation by IBG1, whereas simultaneous mutation of both domains was required to disrupt dBET6-based degradation (Extended Data Fig. 3f). We validated

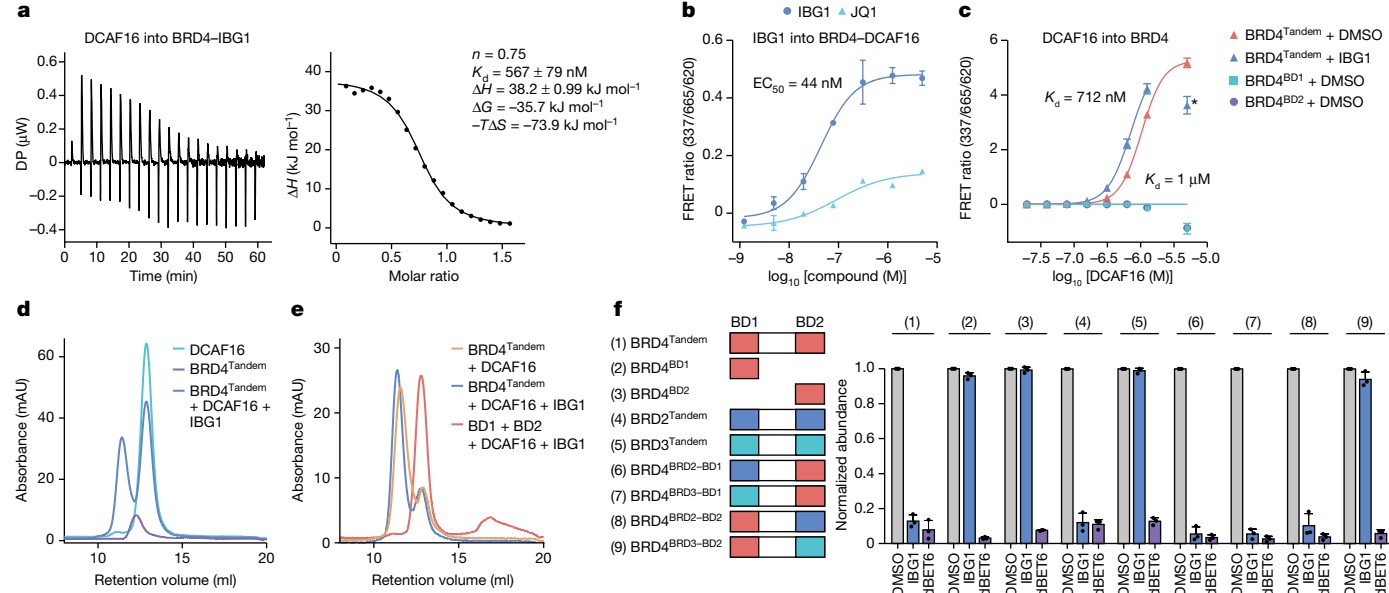

**Fig. 3 | IBG1 enhances the intrinsic interaction between the tandem bromodomain region of BRD4 and DCAF16. a**, ITC measurement of DCAF16–DDB1(ΔBPB)–DDA1 binding to pre-incubated BRD4[Tandem]–IBG1 complex (1:1.1 molar ratio). Data representative of $n = 2$ independent experiments. DP, differential power. **b**, TR-FRET ternary complex-formation assay. Europium-labelled anti-His bound to BRD4[Tandem] was incubated with equimolar Cy5-labelled DCAF16–DDB1(ΔBPB)–DDA1 and increasing concentrations of IBG1 or JQ1. Mean ± s.d. of $n = 3$ technical replicates. **c**, TR-FRET complex-stabilization assay. His-tagged BRD4[Tandem]- or BRD4[BD1] (200 nM) bound to anti-His–europium was incubated with increasing concentrations of Cy5-labelled DCAF16–DDB1(ΔBPB)–DDA1 in the presence or absence of 1 μM IBG1. Mean ± s.d. of $n = 2$ independent experiments, each with 2 technical replicates. The asterisk denotes a datapoint that was excluded from non-linear regression fitting.

**d,e**, UV chromatograms from SEC analysis. DCAF16–DDB1(ΔBPB)–DDA1 and BRD4[Tandem] alone or mixed at a 2:1 molar ratio in the presence of excess IBG1 (**d**), DCAF16–DDB1(ΔBPB)–DDA1 and BRD4[Tandem] mixed at a 1:1 molar ratio in the absence or presence of excess IBG1 (**e**), or DCAF16–DDB1(ΔBPB)–DDA1 mixed with BRD4[BD1] and BRD4[BD2] at a molar ratio of 1:1:1 with excess IBG1 (**e**) were run on an S200 10/300 column. Data representative of $n = 2$ independent experiments. mAU, milli-absorbance units. **f**, BET protein stability reporter assay. Tandem mTagBFP fusions with BRD2, BRD3 or BRD4 bromodomains, isolated BRD4 bromodomains or bromodomain chimeras were expressed in KBM7 cells and protein stability was quantified by FACS following treatment with DMSO, IBG1 (1 nM) or dBET6 (10 nM) for 6 h. Mean ± s.d. of $n = 3$ independent experiments.

the requirement for tandem bromodomains using the 'bump-and-hole' BromoTag approach[25], where a BromoTag–MCM4 fusion was efficiently degraded by a 'bumped' VHL-based PROTAC (ABG1) but not a derivative of IBG1 (bIBG1; Extended Data Fig. 3g,h). These data confirm that, unlike other BET PROTACs, IBG1 requires the simultaneous engagement of both BRD4 bromodomains, and that a single bromodomain is not sufficient for degradation.

We also used the BRD4–BFP reporter assay to identify the determinants of IGB1 selectivity for BRD4 over BRD3 (Fig. 1c and Extended Data Fig. 1d,e). As expected, we observed potent degradation of BRD2, BRD3 and BRD4 tandem constructs by dBET6, whereas IBG1 selectively degraded BRD2 and BRD4 but not BRD3 (Fig. 3f). When we exchanged the linker from BRD4[Tandem] with the corresponding regions in BRD2 or BRD3, or deleted the known SPOP degron[14,15], we observed no influence on degradation (Extended Data Fig. 3f). Next, we swapped either BD1 or BD2 from BRD4[Tandem] with the corresponding domain from BRD2 or BRD3. Whereas exchange of BD1 had minimal influence on protein degradation, for BD2 only a swap with BRD2 was tolerated. By contrast, replacement by the BRD3 BD2 fully disrupted degradation by IBG1 (Fig. 3f). Thus, BD2 determines the selectivity of IBG1 for BRD2 and BRD4 over BRD3.

## IBG1 bivalently binds both BRD4 bromodomains

To gain molecular insights into the mechanism underpinning IBG1-induced BRD4 degradation, we solved the structure of the ternary complex formed between BRD4[Tandem], IBG1 and DCAF16–DDB1(ΔBPB)–DDA1 by cryo-electron microscopy at a resolution of approximately 3.77 Å (Fig. 4a, Extended Data Fig. 4 and Extended Data Table 2). DCAF16 adopts a unique fold consisting of 8 helices, several loops

and a structural zinc ion coordinated by residues C100 and C103 in the loop between α3 and α4 and C177 and C179 of α8 (Extended Data Fig. 5a). Helices 4–6 bind the central cleft between β-propellers A and C of DDB1 in a binding mode distinct from those of other CRL4 substrate receptors[7,26,27] (Extended Data Fig. 5b). Helices 1, 3, 7 and 8 fold into a bundle that sits on the outer surface of β-propeller C blades 5 and 6, as well as the loop between strands c and d of blade 7. Consistent with its role as a CRL substrate receptor, this helical bundle of DCAF16 bridges DDB1 with BRD4. Both bromodomains are simultaneously bound to DCAF16 with a single continuous density representing one molecule of IBG1 located between DCAF16, BD1 and BD2 (Fig. 4b). Although the JQ1 moiety of IBG1 binds canonically to the acetyllysine pocket of BD2, we found that the E7820 moiety unexpectedly binds to the equivalent pocket of BD1. The binding mode of the E7820 portion of IBG1 overlays well with other sulfonamide-containing BET inhibitors that have been co-crystallized with BD1[28,29], with the nitrogen atom of the cyano group taking a position that is occupied by a conserved water molecule in BET bromodomain crystal structures[30] (Extended Data Fig. 5c). In line with these observations, we found that E7820 and other arylsulfonamide derivatives show weak binding to BRD4[Tandem] as well as isolated bromodomains (Extended Data Fig. 5d,e). SEC showed increased retention of IBG1-bound BRD4[Tandem] compared with unbound or JQ1-bound BRD4[Tandem], indicating a decreased hydrodynamic radius consistent with compaction through intramolecular dimerization of BRD4 bromodomains (Fig. 4c). Thus, both bromodomains are simultaneously engaged and bridged by the opposing ends of a single IBG1 molecule. Such a conformational change would also explain the marked increase in entropy observed by ITC for BRD4 binding to DCAF16 in the presence of IBG1 (Fig. 3a and Extended Data Fig. 3d), as the entropic penalty for

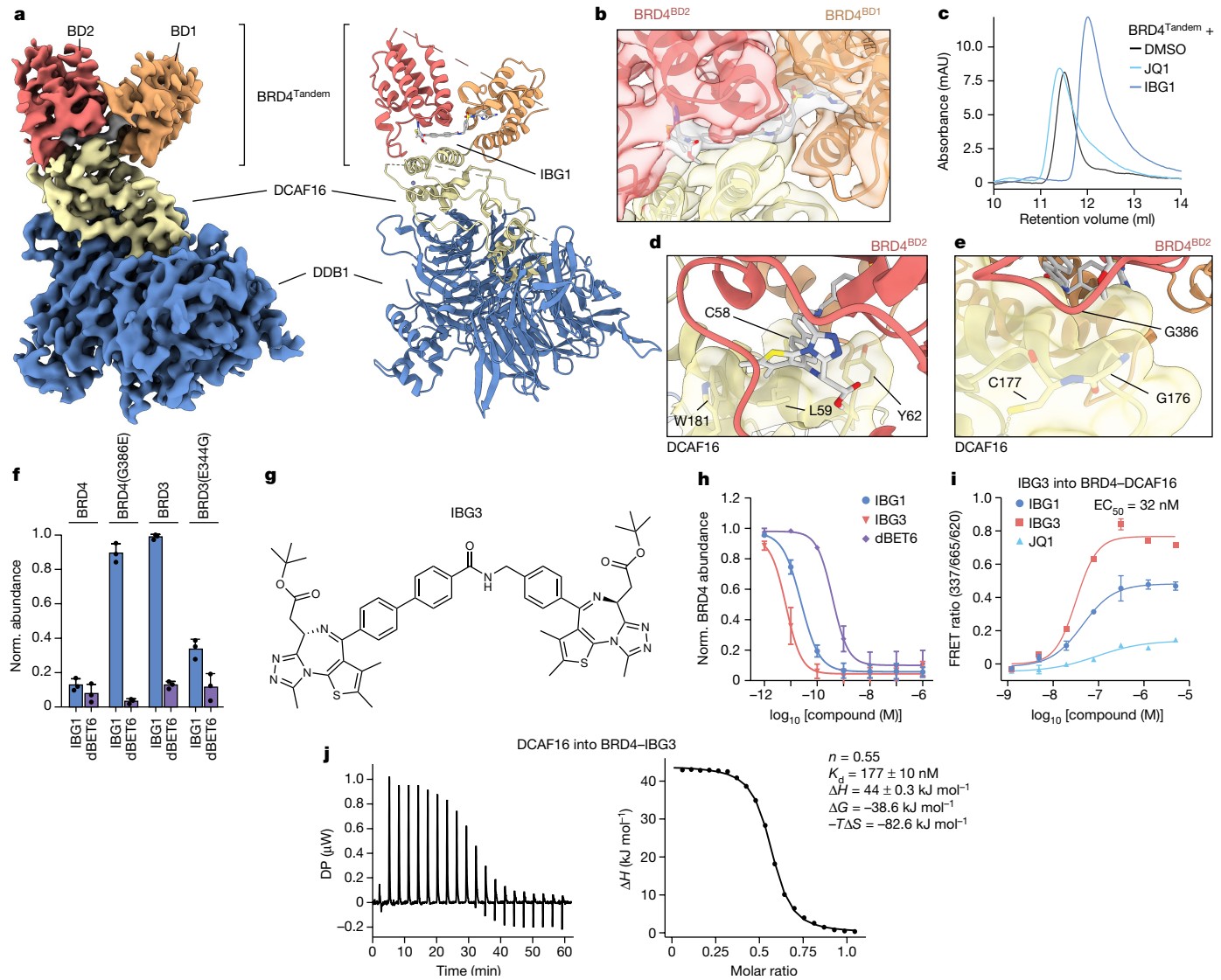

**Fig. 4 | IBG1 engages both BRD4 bromodomains simultaneously and glues BRD4 to DCAF16. a**, Electron density (left) and model (right) of the complex formed between DCAF16, DDB1(ΔBPB), BRD4$^{Tandem}$ (BD1 and BD2) and IBG1. **b**, Electron density at the DCAF16–IBG1–BRD4 interface. The JQ1 moiety binds to BD2, and the sulfonamide engages BD1. **c**, UV chromatograms from SEC analysis. Recombinant BRD4$^{Tandem}$ was incubated with DMSO, JQ1 or IBG1 at a 1:2 molar ratio and run on an S200 10/300 column. Data representative of $n = 2$ independent experiments. **d**, A hydrophobic cage formed by DCAF16 residues C58, L59, Y62 and W181 encloses the JQ1 moiety and linker phenyl ring of IBG1. **e**, Selectivity-determining residue G386 of BD2 at the interface with DCAF16. Colours in **b,d,e** as in **a. f**, FACS reporter assay. KBM7 reporter cells expressing wild-type BRD3, BRD4 or indicated single-point mutant bromodomain tandems were treated with IBG1 (1 nM) or dBET6 (10 nM) for 6 h and BET protein stability

was evaluated by FACS. Mean ± s.d. of $n = 3$ independent experiments. **g–j**, Structure (**g**) and mechanistic characterization (**h–j**) of the dual-JQ1-containing BET degrader IBG3. **h**, BRD4 degradation. KBM7 reporter cells expressing BRD4$^{Tandem}$ were treated for 6 h with increasing concentrations of IBG1, IBG3 or dBET6, and BRD4 protein stability was assessed by FACS. Mean ± s.d. of $n = 3$ independent experiments. **i**, TR-FRET ternary complex-formation assay. Anti-His–europium bound to BRD4$^{Tandem}$ was incubated with equimolar Cy5-labelled DCAF16–DDB1(ΔBPB)–DDA1 and increasing concentrations of IBG1, IBG3 or JQ1. Data for JQ1 and IBG1 as in Fig. 3b. Mean ± s.d. of $n = 3$ technical replicates. **j**, ITC measurements of DCAF16–DDB1(ΔBPB)–DDA1 complex binding to pre-incubated BRD4$^{Tandem}$–IBG3 (1:1.1 molar ratio). Data representative of $n = 2$ independent experiments.

intramolecularly engaging and stabilizing the bromodomains is paid for by IBG1 binding prior to complex formation with DCAF16.

At the ternary interface, DCAF16 encloses the hydrophobic dimethylthiophene and phenyl groups of the JQ1 moiety as well as the linker phenyl, shielding them from solvent (Fig. 4d). DCAF16 also contacts BD1 through residue W54, which binds into a hydrophobic pocket on BD1 (Extended Data Fig. 5f). The ternary complex is further stabilized by intramolecular contacts between the two bromodomains, including the sandwiching of M442 between W81 and P375 in the WPF shelves of BD1 and BD2, respectively (Extended Data Fig. 5g). This series of interactions buries a large hydrophobic surface area upon complex

formation, which is consistent with the highly entropically favourable interaction of IBG1-bound BRD4$^{Tandem}$ with DCAF16 (Fig. 3a). G386 of BD2 is positioned at a crucial interface in close contact with DCAF16, with only limited space available for the amino acid side chain (Fig. 4e). The corresponding residue in BRD2 is also a glycine (G382), whereas in BRD3 it is a glutamate (E344), suggesting a role for this residue in determining the BRD2 and BRD4 selectivity of IBG1. Indeed, a G386E mutation in the BRD4 completely abrogated degradation, and the reciprocal E344G mutation in BRD3 sensitized it to IBG1 (Fig. 4f).

We hypothesized that bifunctional compounds with two high-affinity bromodomain ligands should stabilize the degradation-competent

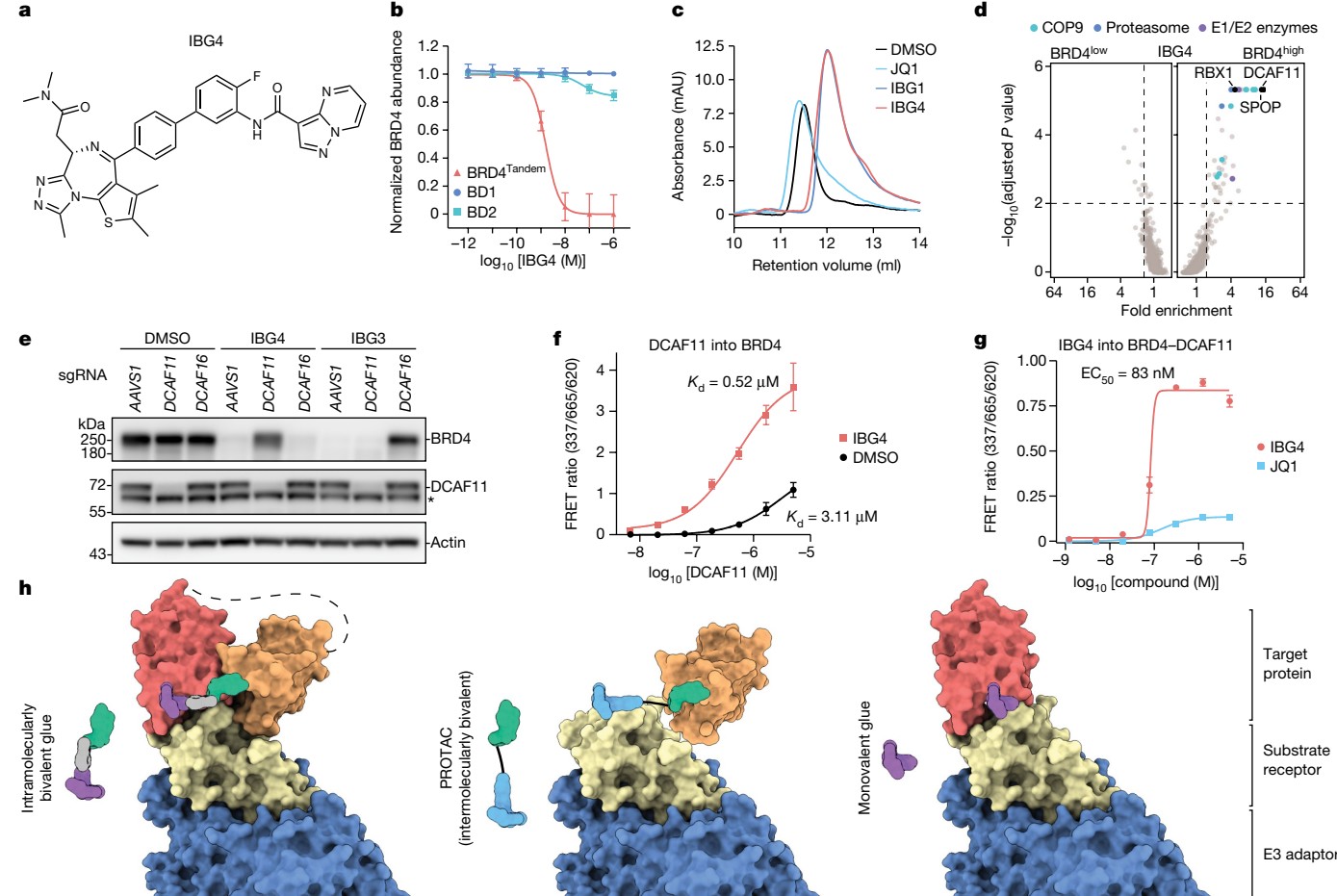

**Fig. 5 | IBG4 is a DCAF11-dependent intramolecular bivalent glue degrader.**
**a**, Structure of IBG4. **b**, Tandem bromodomain requirement of IBG4. KBM7 reporter cells expressing BRD4$^{Tandem}$, BD1 or BD2 were treated for 6 h with increasing concentrations of IBG4, and protein degradation was evaluated by FACS. Mean ± s.d. of $n = 3$ independent experiments. **c**, UV chromatograms from SEC analysis of BRD4$^{Tandem}$ incubated with DMSO, JQ1, IBG1 or IBG4. Data representative of $n = 2$ independent experiments. Data for DMSO, JQ1 and IBG1 as in Fig. 4c. **d**, CRISPR–Cas9 BRD4 stability screen. KBM7 iCas9 BRD4 reporter cells expressing a CRL-focused sgRNA library were treated with IBG4 (100 nM) for 6 h before FACS as in Fig. 2b. The 20S proteasome subunits, COP9 signalosome subunits and E1 and E2 ubiquitin enzymes inside the scoring window (one-sided MAGeCK $P$ value < 0.01, fold change > 1.5; dashed lines) are highlighted. **e**, Immunoblot-based screen validation. *AAVS1* control, *DCAF11*- or

*DCAF16*-knockout KBM7 cells were treated with DMSO, IBG4 (100 nM) or IBG3 (0.1 nM) for 6 h and BRD4 was quantified by immunoblotting. The asterisk denotes a nonspecific band. Data representative of $n = 3$ independent experiments. **f**, TR-FRET complex-stabilization assay. His-tagged BRD4$^{Tandem}$ (100 nM) bound to anti-His–europium was incubated with increasing concentrations of Cy5-labelled DCAF11–DDB1(ΔBPB)–DDA1 and 500 nM IBG4 or DMSO. Mean ± s.d. of $n = 2$ independent experiments, each with 3 technical replicates. **g**, TR-FRET complex-formation assay. His-tagged BRD4$^{Tandem}$ bound to anti-His–europium was incubated with equimolar Cy5-labelled DCAF11–DDB1(ΔBPB)–DDA1 and increasing concentrations of IBG4 or JQ1. Mean ± s.d. of $n = 3$ technical replicates. **h**, Schematic model of the different modes of molecular recognition with traditional monovalent glues and bivalent PROTACs versus intramolecularly bivalent glues revealed in this work.

bromodomain conformation even more efficiently, potentially enabling the generation of more effective DCAF16-based degraders. We synthesized a series of compounds in which we replaced E7820 with a second JQ1 moiety while keeping the IBG1 linker architecture intact (IBG2 and IBG3; Fig. 4g and Extended Data Fig. 6a) and found that BRD4 and BRD2 degradation efficiencies exceeded those of IBG1, with IBG3 showing degradation in a low picomolar range (DC$_{50}$ = 6.7 pM and 8.6 pM, respectively; Fig. 4h and Extended Data Fig. 6b,c). IBG3 also showed improved 'gluing' of the BRD4–DCAF16 complex by TR-FRET (EC$_{50}$ = 32 nM; Fig. 4i), increased affinity of DCAF16 for BRD4–IBG3 by ITC while maintaining an endothermic ITC profile consistent with IBG1 (Fig. 4j), and more pronounced compaction of bromodomains by SEC (Extended Data Fig. 6d). Similar to its parental compound, IBG3 was specific for BRD2 and BRD4 over BRD3 (Extended Data Fig. 6b,c), selective for tandem bromodomains over isolated bromodomains (Extended Data Fig. 6e), and mediated by DCAF16 (Extended Data Fig. 6f,g), indicating degradation via the same intramolecular glue mechanism. BRD4

constructs with two copies of either BD1 or BD2 were fully resistant to IBG1 and IBG3 (Extended Data Fig. 6h), supporting the importance of the explicit relative arrangement of the bromodomains for ternary complex architecture. This also probably explains the functional difference to the previously published bivalent bromodomain-targeting compounds MT1 and MS645, which potently inhibit but do not degrade BET proteins[31–33] (Extended Data Fig. 6i,j). Thus, on the basis of mechanistic and structural insights, we rationally designed IBG3 as an improved intramolecular bivalent glue with higher potency.

## A DCAF11-based intramolecular glue

As bridging of two domains of BRD4 induced a potent gain of function by stabilizing interactions with DCAF16, we surmised that bivalent domain engagement by intramolecular glues might be utilized to more broadly modify protein–protein interactions to rewire protein function. A recently reported BRD4 degrader consisting of a pyrazolopyrimidine

moiety connected to JQ1 via a short rigid linker[34] (hereafter referred to as IBG4; Fig. 5a) caught our attention since it, similar to IBG1, showed efficient degradation of BRD4[Tandem], while sparing isolated bromodomains and acetyllysine pocket mutants N140F and N433F (Fig. 5b and Extended Data Fig. 7a). In SEC, IBG4 induced a similar compaction of BRD4[Tandem] as IBG1 (Fig. 5c) and in NanoBRET conformational biosensor assays both compounds induced comparable levels of intramolecular bromodomain interactions (Extended Data Fig. 7b), together indicating that IBG4 induces—similar to IBG1—bromodomain dimerization in *cis*. Finally, the pyrazolopyrimidine moiety of IBG4 showed similar affinity to BRD4 bromodomains as the E7820 moiety in IBG1 (Extended Data Fig. 7c). Thus, despite being structurally differentiated, IBG4 phenotypically mimics the cellular mechanism of action of IBG1, suggesting that both compounds share an intramolecular glue-like mechanism. Unlike IBG1, IBG4 showed high specificity for BRD4 and did not efficiently degrade BRD2 (Extended Data Fig. 7d), pointing towards different structural requirements of a potential ternary BRD4–IBG4–E3 ligase complex. Indeed, whereas degradation was blocked by the neddylation inhibitor MLN4924 (Extended Data Fig. 7e), DCAF16 knockout had no effect on IBG4-mediated BRD4 degradation (Extended Data Fig. 7f). We thus performed a BRD4 degradation CRISPR screen and in addition to the endogenous BRD4 turnover factor SPOP identified the CRL4–DCAF11 complex to mediate resistance to IBG4 (Fig. 5d,e). Despite no predicted structural similarity to DCAF16 (Extended Data Fig. 7g), DCAF11 showed measurable intrinsic affinity for BRD4 in TR-FRET (Fig. 5f) and this interaction was significantly enhanced in the presence of IBG4 (Fig. 5f,g). Finally, in line with stabilization of the ternary complex, the addition of IBG4 induced co-elution of BRD4 with DCAF11 in SEC (Extended Data Fig. 7h,i). IBG4 thus recapitulates all cellular and biophysical properties of the intramolecular glue degraders described above, but extends the mechanistic scope to another structurally unrelated E3 ligase. Collectively, our data establish intramolecular dimerization of protein domains as a novel strategy for efficient targeted protein degradation that can be rationally engineered following principles of structure-based drug design.

## Discussion

Most molecular glue degraders reported so far, such as the plant hormone auxin[35] and the immune modulatory drug (IMiD) lenalidomide[27,36–39], function via binary engagement of an E3 ligase that subsequently recruits neosubstrates for ubiquitination. Thus, only targets that can be productively paired to a chemically accessible ligase can be addressed via this strategy, and very few glues have been developed from a given target protein ligand[40,41]. Here we define the mechanism of chemically distinct BET protein degraders as simultaneously engaging two separate sites on the target protein to nucleate formation of stable ternary complexes and induce target protein degradation. Thus, we reveal a new strategy distinct from conventional bivalent PROTACs and monovalent glues, which we designate 'intramolecular bivalent gluing', that enables the development of potent and target-selective degraders (Fig. 5h). On the basis of our mechanistic and structural insights, we rationally improved the first-generation intramolecular bivalent glue degrader IBG1 by enhancing its affinity to tandem bromodomains and gluing to DCAF16. This resulted in the second-generation IBG3, which showed half-maximal degradation at single digit picomolar concentrations, demonstrating that this novel class of degraders can reach efficiencies higher than any PROTAC reported to date[42].

Around 60–80% of all human proteins feature at least two distinct domains and are thus potentially accessible to targeted degradation via intramolecular bivalent gluing[43,44]. Both IBG1 and IBG4 feature only a single high-affinity BET ligand, whereas the second moiety shows only low affinity for its respective target domain. Nevertheless, both compounds trigger degradation at nanomolar concentrations, suggesting that these glues can efficiently degrade target proteins even when utilizing suboptimal ligands. Even though the intramolecular bivalent glue degraders presented here are currently focused on a single family of target proteins, these relatively lenient requirements for target binding suggest that this approach might be applicable for a much broader range of targets. Conversely, our work also highlights the challenges of using sub-specific or low-affinity ligands—such as E7820—as E3-binding 'handles' for conventional PROTAC mechanism, sounding a note of caution as the field expands to E3 ligases beyond CRBN and VHL.

Even though IBG1 and IBG4 share the same mechanism, we find that they utilize two structurally unrelated E3 ligases to induce ubiquitination and degradation: IBG1 functions via CRL4–DCAF16, whereas IBG4 functions via CRL4–DCAF11. We identified intrinsic affinities between BRD4 and either E3 ligase even in the absence of ligands. This reinforces the emerging concept that molecular glue degraders often stabilize pre-existing, albeit functionally inconsequential E3–target interactions[41,45,46] and suggests that these affinities may be essential for (intra)-molecular glue degraders. The exclusive requirement of DCAF16 and DCAF11 for IBG1 and IBG4, respectively, suggests that the varying arrangements and linker architectures align the BRD4 bromodomains in different orientations relative to each other, generating distinct protein–ligand surfaces that are selectively recognized by the two ligases. Our work suggests that both DCAF11 and DCAF16 are primed for BET bromodomain recognition and that relatively mild modifications of the interaction surface could be sufficient to trigger productive complex stabilization and ubiquitination. The apparent affinity of BET proteins for various E3 ligases might be a potential explanation for their eminent accessibility for chemically induced protein degradation[47].

In conclusion, we show that structurally distinct BET degraders converge on a shared novel mechanism of action: intramolecular dimerization of two domains to modify protein surface and modulate protein–protein interactions. So far, this concept is limited to degradation of a single target protein family and generalizability to other targets remains to be shown. However, protein surface modulation via intramolecular, chemical bridging of binding sites in *cis* could outline a strategy to pharmacologically utilize intrinsic interactions with diverse effector proteins and rewire cellular circuits for protein degradation and beyond.

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

## Methods

### Chemical synthesis
Additional details are provided in the Supplementary Methods.

### Plasmids and oligonucleotides
The design and construction of the human CRL-focused sgRNA library used for BRD4 stability screens, lentiviral sgRNA expression vectors used for single gene knockouts, as well as viral vectors used for the engineering of inducible Cas9 cell lines have been described previously[48,49]. For the engineering of the fluorescent protein stability reporters, the short isoform of *BRD4* (*BRD4*(S)) (Twist Bioscience), *BRD2* (Addgene plasmid #65376, a gift from K. Miller[50]) or *BRD3* (Addgene plasmid #65377, a gift from K. Miller[50]) were cloned into a pRRL lentiviral vector, fused to a 3×V5 tag and mTagBFP, and coupled to mCherry for normalization. For knockout and rescue studies, *DCAF16* open reading frame cDNA (Twist Bioscience) was synonymously mutated to remove the sgRNA protospacer adjacent motif and seed sequence, coupled to a Flag tag and cloned into a pRRL lentiviral vector expressing iRFP670 for flow-cytometric detection. All plasmids and sgRNAs used in this study are shown in Extended Data Table 1, and the CRL-focused sgRNA libraries used for FACS-based and viability-based CRISPR–Cas9 screens are shown in Supplementary Tables 2 and 4, respectively.

### Cell culture
HEK293, HCT-116, HeLa and MV4;11 cell lines, originally sourced from ATCC, were provided by the MRC PPU reagents facility at the University of Dundee. KBM7 iCas9 cells were a gift from J. Zuber. HEK293, HeLa, Lenti-X 293 T lentiviral packaging cells (Clontech) and HCT-116 were cultured in DMEM (Gibco) supplemented with 10% fetal bovine serum (FBS; Thermo Fisher), 100 U ml$^{-1}$ penicillin-streptomycin (Thermo Fisher) and 2 mM L-glutamine (Thermo Fisher). MV4;11 and KBM7 cells were cultured in IMDM (Gibco), supplemented with the same additives as above. All cell lines were grown in a humidified incubator at 37 °C and 5% CO$_2$ and routinely tested for mycoplasma contamination. All cell lines were authenticated by short tandem repeat profiling.

### Lentivirus production and transduction
Semiconfluent Lenti-X cells were co-transfected with lentiviral plasmids, the lentiviral pCMVR8.74 helper (Addgene plasmid #22036, a gift from D. Trono) and pMD2.G envelope (Addgene plasmid #12259, a gift from D. Trono) plasmids using polyethylenimine (PEI) transfection (PEI MAX MW 40,000, Polysciences) as previously described. Virus containing supernatant was clarified by centrifugation. Target cells were infected at limiting dilutions in the presence of 4 μg ml$^{-1}$ polybrene (Santa Cruz Biotechnology).

### CRISPR–Cas9 *DCAF15*-knockout cell line generation
The HCT-116 *DCAF15*-knockout cell line was generated via ribonuclear protein (RNP) transfection using sgRNAs (IDT) targeting *DCAF15* exon 2 and exon 4 (Extended Data Table 1), spCas9 Nuclease V3 (IDT) and TransIT-X2 (Mirus Bio). Following transfection for 48 h, cells were trypsinized and re-plated in 96-well plates at low density and allowed to grow for >2 weeks. Single colonies were isolated and expanded and verified for *DCAF15* knockout via western blotting, using an optimized RBM39 degradation assay as well as via genomic DNA sequencing.

### CRISPR–Cas9 HiBiT and BromoTag knock-in cell line generation
HiBiT BRD2, BRD3 and BRD4 cell lines were generated via RNP transfection of single-stranded DNA oligonucleotides (IDT) as the ssODN donor templates, spCas9 (Sigma-Aldrich) and target-specific sgRNA (IDT) (Extended Data Table 1). HEK293 cells were resuspended in buffer R (Thermo Fisher), along with the RNP complex and ssODN template, and electroporated using a 10 μl Neon Electroporation System cuvette tip (Thermo Fisher). Immediately following electroporation, cells were added to pre-warmed DMEM supplemented with 10% FBS and 100 U ml$^{-1}$ penicillin–streptomycin added for BromoTag cell lines only. Edited pools were analysed for HiBiT insertion by assaying for luminescence on a PHERAstar spectrophotometer (BMG Labtech) 48–72 h post-electroporation. Successful knock-in of HiBiT three days post-electroporation was first established using HiBiT lytic assay (Promega) on the mixed cell population. Following identification of luminescent signal these cells underwent single cell sorting using an SH800 cell sorter (Sony Biotechnology). Single cells were sorted into 3× 96-well plates per experiment in 200 μl of 50% filtered preconditioned media from healthy cells and 50% fresh DMEM. After two weeks, all visible colonies were expanded, validated using the HiBiT lytic assay.

BromoTag cell lines were generated in HEK293 cells via simultaneous transfection of two vectors at a 4:1 reagent:DNA ratio with FuGENE 6 (Promega). The first vector was a pMK-RQ vector containing 500-bp homology arms on either side of either an eGFP-IRES-BromoTag or eGFP-IRES-HiBiT-BromoTag sequence for integration into MCM4 and BRD4, respectively (Extended Data Table 1). The second vector was a custom pBABED vector harbouring U6-sgRNA, Cas9 and puromycin expression cassettes. Following transfection, cells were repeatedly washed with PBS and then treated with 1 μg ml$^{-1}$ puromycin for one week before FACS sorting. Single cell clones were generated by FACS sorting of single GFP$^+$ cells using an SH800 cell sorter and sorting between 2 to 10 96-well plates in 200 μl of 50% filtered preconditioned media from healthy cells mixed with 50% fresh media.

### siRNA-mediated knockdown
Cells were transfected for 48 h using ON-TARGETplus SMARTPool siRNAs for *DCAF15*, *DCAF16*, *DDB1*, *RBX1*, *CUL4A* and *CUL4B* (all from Dharmacon) and RNAiMAX (Invitrogen) following the manufacturer's instructions, with 35 pmol of siRNA per well in 6-well plates. When simultaneously targeting two genes, half the amount of siRNA was used for each gene.

### Cell viability assay
MV4;11, HCT-116 or KBM7 cells were plated in 96-well plates at a density of 0.5 × 10$^6$ (MV4;11 and HCT-116) or 0.1 × 10$^6$ (KBM7) cells per ml in 50 μl cell suspension per well. The following day, 2× stocks of compounds were added for a final volume of 100 μl. Cells were treated for 24 h (MV4;11), 72 h (KBM7) or 96 h (HCT-116) in a humidified incubator at 37 °C and 5% CO$_2$. CellTiterGlo (G7570, Promega) or CellTiterGlo 2.0 reagent (G924A, Promega) was added to the plates per manufacturer instructions, before shaking the plate for 3–20 min at 300 rpm and measuring the luminescence using a PHERAstar (BMG Labtech) operated on PHERAstar software (firmware v1.33) or VICTOR X3 (PerkinElmer) multilabel plate reader operated on PerkinElmer 2030 software (v4.0). The results were normalized to DMSO controls and analysed using Graphpad Prism (v9.5.1) to derive EC$_{50}$ values by four-parameter non-linear regression curve fitting or interpolation of a sigmoidal standard curve.

### Degradation assays and western blotting
HEK293 and HCT-116 cells were plated in 6-well plates at varying densities (0.2 to 0.6 × 10$^6$ cells per ml) depending on experimental setup. In all experiments, media was changed prior to compound treatment. Stock solutions of compounds were prepared in DMSO at a concentration of 10 mM and stored at −20 °C. Working dilutions were made fresh using DMEM media and added dropwise to 6-well plates. For competition assays, cells were pre-treated with 10 μM of the competition compounds, 3 μM MLN4924 or 50 μM MG132 for 1 h, before treating with IBG1 at 10 nM for 2 h.

For cell collection, cells were washed once with ice-cold PBS before lysis for 15 min on ice with RIPA buffer supplemented with benzonase (1:1,000, Sigma or Millipore 70746) and cOmplete EDTA-free Protease Inhibitor Cocktail (11873580001, Roche). Following clearance via

centrifugation, protein concentration of lysates was determined using the Pierce BCA Protein Assay (23225, Fisher Scientific) and 20–30 μg of lysate was prepared using 4× LDS sample buffer (Thermo Fisher) and 10% 2-mercaptoethanol or 50 mM dithiothreitol (DTT) and run on NuPAGE 4–12% bis-tris gels (Thermo Fisher). Proteins were transferred to nitrocellulose membranes, blocked for 1 h in 5% milk TBS-T at room temperature, before incubating with primary antibodies overnight at 4 °C. The following primary antibodies were used: BRD2 (1:1,000, no. Ab139690, Abcam), BRD3 (1:2,000, Ab50818, Abcam), BRD4 (1:1,000, E2A7X, 13440, Cell Signaling Technology and Ab128874, Abcam), BromoTag (1:1,000, NBP3-17999, Novus Biologicals), CUL4A (1:2,000, A300-738A, Bethyl Laboratories), CUL4B (1:2,000, 12916-1-AP, Proteintech), DDB1 (1:1,000, A300-462A, Bethyl Laboratories), MCM4 (1:1,000, ab4459, Abcam) RBM39 (1:1,000, HPA001591, Atlas Antibodies), RBX1 (1:1,000, D3J5I, 11922, Cell Signalling Technology), DCAF11 (1:2,000, A15519, ABclonal), cleaved caspase-3 (1:1,000, D3E9, 9579, Cell Signalling Technology), PARP1 (1:1,000, 9542, Cell Signalling Technology), MYC (1:500, D84C12, 5605, Cell Signalling Technology), β-actin (1:10,000, AC-15, A5441, Sigma-Aldrich), α-tubulin (1:500, DM1A, T9026, Sigma-Aldrich). Membranes were then washed in TBS-T and incubated with fluorescent or horseradish peroxidase (HRP)-conjugated secondary antibodies for 1 h at room temperature, before further washes and imaging on a ChemiDoc Touch imaging system (Bio-Rad) operated on Image Lab software (v2.4.0.03). Secondary antibodies used were HRP anti-rabbit IgG (1:2500, 7074, Cell Signaling Technology), HRP anti-mouse IgG (1:5,000, 7076, Cell Signaling Technology), IRDye 680RD anti-mouse (1:5,000, 926-68070, Li-Cor), IRDye 800CW anti-rabbit (1:5,000, 926-32211, Li-Cor), StarBright blue 520 goat anti-mouse (1:5,000, 12005866, Bio-Rad) and hFABTM rhodamine anti-tubulin (1:5,000, 12004165, Bio-Rad). Western blots were quantified using Image Lab software (v6.1 build 7).

## HiBiT degradation assays
Endogenously tagged HiBiT cells were plated in 96-well plates (PerkinElmer) at a density of $0.5 × 10^6$ cells per ml, with 50 μl of cell suspension per well. The following day, 2× stocks of compounds were added for a final volume of 100 μl. Cells were treated for 5, 6 or 24 h as indicated in the respective figure legends before lysis using the HiBiT lytic assay buffer (Promega) per manufacturer instructions. Plates were then read on a BMG Pherastar plate reader for luminescence detection. Treated wells were normalized to a DMSO-only control and analysed using GraphPad Prism (v9.3.1) via fitting of non-linear regression curves for extraction of $DC_{50}$ and maximal degradation ($D_{MAX}$) values.

## Kinetic ubiquitination and degradation assays
For kinetic ubiquitination assays, HiBiT-tagged HEK293 cells were seeded in 6-well plates at a density of $8 × 10^6$ cells per ml in 2 ml volume. After 5 h, LgBiT and Halo-Ub cDNA (Promega) were transfected using FuGENE HD (Promega) with 1 μg of each plasmid at a 3:1 transfection reagent:plasmid ratio. The following day, cells were trypsinized and resuspended in phenol red-free OptiMEM (Gibco) supplemented with 4% FBS and seeded in 96-well plates at a density of $3.5 × 10^5$ cells per ml in the presence or absence of 0.1 mM HaloTag NanoBRET ligand (Promega). Following overnight incubation, media was removed from the wells and replaced with 90 μl OptiMEM (4% FBS) with a 1:100 dilution of Vivazine substrate. The plates were incubated at 37 °C for 1 h before 10× stocks of experimental compounds were added and the plates were analysed on a GloMAX Discover microplate reader (software v4.0.0, firmware v4.92; Promega) in kinetic mode for NanoBRET ratio metric (460 nm donor and 618 nm acceptor emissions) signal detection for 6 h, with measurements taken every 3–5 min. Data was processed by subtracting NanoBRET ligand-free controls before plotting NanoBRET signal versus time in GraphPad Prism (v9.3.1).

Kinetic degradation assays were performed as previously described[51], using the HiBiT-tagged cells with exogenous LgBiT transfection as described above for the kinetic ubiquitination assays. Cells were incubated in Endurazine substrate (1:100) for 2.5 h at 37 °C prior to 10× compound addition, with luminescence measurements taken on a GloMAX Discover microplate reader (Promega) every 15 min for 24 h. Data were normalized to DMSO-only controls and plotted for luminescence signal versus time in GraphPad Prism (v9.3.1).

## NanoBRET bromodomain confirmational sensor assay
Transient transfection of the dual NanoLuc and Halo-Tagged tagged BRD4$^{Tandem}$ plasmid (Promega) was performed as described previously[51]. In brief, 0.02 μg of plasmid and 2 μg of carrier DNA were combined with FuGENE HD (Promega) at a 3:1 ratio and added per well of a 6-well plate seeded with 70% confluent HEK293 cells. The following day, cells were trypsinized and resuspended in phenol red-free OptiMEM (Gibco) supplemented with 4% FBS and 100 μl were seeded per well in 96-well plates at a density of $2 × 10^5$ cells per ml in the presence or absence of 0.1 mM HaloTag NanoBRET ligand (Promega). The following morning, the media was aspirated and replaced with phenol red-free media containing MG132 (10 μM final concentration) for 1 h, before cells were incubated with test compounds for 3 h. For cell lysis and detection, 100 μl of 2× NanoBRET substrate solution was added per well, the plate was incubated in darkness while shaking at 400 RPM for 3 min, before reading on a BMG Pherastar plate reader equipped with a NanoBRET filter (618/460 nm). Wells lacking Halo ligand were subtracted from wells containing Halo ligand, and the fold increase in signal compared to DMSO was plotted using GraphPad Prism (v9.3.1).

## FACS-based CRISPR–Cas9 BRD4 stability screens
For pooled FACS-based CRISPR–Cas9 BRD4 protein stability screens, a CRL-focused sgRNA library[49] was packaged in lentivirus using polyethylenimine (PEI MAX MW 40,000, Polysciences) transfection of Lenti-X cells and the lentiviral pCMVR8.74 helper (Addgene plasmid #22036, a gift from D. Trono) and pMD2.G envelope (Addgene plasmid #12259, a gift from D. Trono) plasmids. The virus containing supernatant was cleared of cellular debris by filtration through a 0.45-μm polyethersulfone filter and used to transduce KBM7 BRD4–BFP reporter cells harbouring a doxycycline-inducible Cas9 allele (iCas9) at a multiplicity of infection of 0.05 and 1,000-fold library representation. Library-transduced cells were selected with G418 (1 mg ml$^{-1}$, Gibco) for 14 days, expanded and Cas9 expression was induced with doxycycline (0.4 μg ml$^{-1}$, PanReac AppliChem).

Three days after Cas9 induction, 25 million cells per condition were treated with DMSO (1:1,000), MZ1 (10 nM), IBG1 (1 nM), GNE-0011 (1 μM), IBG3 (0.1 nM) or IBG4 (100 nM) for 6 h in 2 biological replicates. Cells were washed with PBS, stained with Zombie NIR Fixable Viability Dye (1:1,000, BioLegend) and APC anti-mouse Thy1.1 (also known as CD90.1) antibody (1:400, 202526, BioLegend) in the presence of Human TruStain FcX Fc Receptor Blocking Solution (1:400, 422302, BioLegend), and fixed with 0.5 ml methanol-free paraformaldehyde 4% (Thermo Scientific Pierce) for 30 min at 4 °C, while protected from light. Cells were washed with and stored in FACS buffer (PBS containing 5% FBS and 1 mM EDTA) at 4 °C overnight. The next day, cells were strained trough a 35-μm nylon mesh and sorted on a BD FACSAria Fusion (BD Biosciences) operated on BD FACSDiva software (v8.0.2) using a 70-μm nozzle. Aggregates, dead (Zombie NIR positive), Cas9-negative (GFP) and sgRNA library-negative (THY1.1–APC) cells were excluded, and the remaining cells were sorted based on their BRD4–BFP and mCherry levels into BRD4$^{high}$ (5–10% of cells), BRD4$^{mid}$ (25–30%) and BRD4$^{low}$ (5–10%) fractions. For each sample, cells corresponding to at least 1,500-fold library representation were sorted per replicate.

Next-generation sequencing (NGS) libraries of sorted cell fractions were prepared as previously described[48]. In brief, genomic DNA was isolated by cell lysis (10 mM Tris-HCl, 150 mM NaCl, 10 mM EDTA, 0.1% SDS), proteinase K treatment (New England Biolabs) and DNAse-free RNAse digest (Thermo Fisher Scientific), followed by two rounds of

phenol extraction and 2-propanol precipitation. Isolated genomic DNA was subjected to several freeze–thaw cycles before nested PCR amplification of the sgRNA cassette.

Barcoded NGS libraries for each sorted population were generated using a two-step PCR protocol using AmpliTaq Gold polymerase (Invitrogen). The resulting PCR products were purified using Mag-Bind TotalPure NGS beads (Omega Bio-tek) and amplified in a second PCR introducing the standard Illumina adapters. The final Illumina libraries were bead-purified, pooled and sequenced on HiSeq 3500 or NovaSeq 6000 platforms (Illumina).

Screen analysis was performed as previously described[48]. In brief, sequencing reads were trimmed using fastx-toolkit (v0.0.14), aligned using Bowtie2 (v2.4.5) and quantified using featureCounts (v2.0.1). The crispr-process-nf Nextflow workflow is available at https://github.com/ZuberLab/crispr-process-nf/tree/566f6d46bbcc2a3 f49f51bbc96b9820f408ec4a3. For statistical analysis, we used the crispr-mageck-nf Nextflow workflow, available at https://github.com/ZuberLab/crispr-mageck-nf/tree/c75a90f670698bfa78bfd8be-786d6e5d6d4fc455. To calculate gene-level enrichment, the sorted populations (BRD4[high] or BRD4[low]) were compared to the BRD4[mid] populations in MAGeCK (0.5.9)[52], using median-normalized read counts.

## Viability-based CRISPR–Cas9 screen

The ubiquitin–NEDD8 system CRISPR-knockout library (Supplementary Table 4) was generated using the covalently closed circular-synthesized (3Cs) technology, as previously described[53,54]. The library contained 3,347 gRNAs cloned under the U6 promoter in a modified pLentiC-RISPRv2-puromycin vector containing a modified gRNA scaffold sequence starting with GTTTG. Each gene was represented by four gRNAs selected with the Broad Institute CRISPick tool[55–57]. Additionally, the library included a set of essential genes, non-targeting as well as AAVS1-targeting control sgRNAs.

HCT-116 cells were transduced with the ubiquitin–NEDD8 system lentiviral CRISPR–Cas9 library at a multiplicity of infection of 0.5 and a coverage of 500. Cells were selected with 1 μg ml⁻¹ puromycin for 12 days. Eight million selected cells per condition were then plated in T175 flasks. Cells were treated with DMSO or IBG1 (58 nM), corresponding to 4 times the $IC_{50}$ value for 3 days, followed by replating and treatment for additional 3 days. After a total of 6 days of treatment, cells were trypsinized, washed three times with PBS, followed by genomic DNA isolation. Sequencing libraries were prepared via PCR as previously described[54] and purified via GeneJET Gel Extraction Kit (Thermo Fisher Scientific).

Raw sequencing data were demultiplexed with bcl2fastq v2.20.0.422 (Illumina) to generate raw fastq files. To determine the abundance of individual gRNAs per samples, the fastq files were trimmed using cutadapt (v2.8) to retain only the putative gRNA sequences. These sequences were then aligned to the original gRNA library with Bowtie2 (v2.3.0) and only perfect matches were counted. Statistical analysis was performed via MAGeCK[52], using median or total read count normalization and removal of gRNAs with zero counts in the control samples. Genes with a $\log_2$-transformed fold change (LFC) > 1 or < −1 and a $P$ value < 0.01 were labelled as significantly depleted or enriched hits.

## Flow-cytometric BRD4 reporter assay

KBM7 iCas9 cells were transduced with lentivirus expressing wild-type, mutated or truncated versions of the SFFV–BRD4(S)–mTagBFP–P2A–mCherry reporter to generate stable reporter cell lines. For evaluation of reporter degradation, cells were treated with DMSO (1:1,000), IBG1 (1 nM), dBET6 (10 nM), IBG3 (0.1 nM) or IBG4 (100 nM) for 6 h before flow cytometry analysis on an LSR Fortessa (BD Biosciences) operated on BD FACSDiva software (v9.0).

To quantify the influence of genetic perturbations on compound-induced reporter degradation, stable BRD4(S) or BRD4[Tandem] reporter cell lines were transduced with lentiviral sgRNA (pLenti-U6-sgRNA-IT-EF1αs-THY1.1-P2A-NeoR) and/or transgene expression vectors

(pRRL-SFFV-3xFlag-DCAF16-EF1αs-iRFP670) to 30–50% transduction efficiency. Cas9 expression was induced with doxycycline (0.4 μg ml⁻¹) for 3 days, followed by 6 h of degrader treatment. Cells were stained for sgRNA expression with an APC-conjugated anti-mouse Thy1.1 antibody (202526, BioLegend; 1:400) and human TruStain FcX Fc receptor blocking solution (422302, BioLegend; 1:400) for 5 min in FACS buffer (PBS containing 5% FBS and 1 mM EDTA) at 4 °C. Cells were washed and resuspended in FACS buffer and analysed on an LSR Fortessa (BD Biosciences).

Flow-cytometric data analysis was performed in FlowJo v10.8.1. BFP and mCherry mean fluorescence intensity values for were normalized by background subtraction of the respective values from reporter-negative KBM7 cells. BRD4 abundance was calculated as the ratio of background subtracted BFP to mCherry mean fluorescence intensity, and is displayed normalized to DMSO-treated, sgRNA and cDNA double-negative cells.

## Quantitative proteomics

For unbiased identification of degrader target proteins, $50 \times 10^6$ KBM7 iCas9 cells per condition were treated with DMSO (1:1,000), IBG1 (1 nM) or dBET6 (10 nM) for 6 h in biological triplicates. Cells were collected via centrifugation, washed three times in ice-cold PBS and snap-frozen in liquid nitrogen. Cell pellets were lysed in 500 μl of freshly prepared lysis buffer (50 mM HEPES pH 8.0, 2% SDS, 1 mM PMSF and protease inhibitor cocktail (Sigma-Aldrich)). Samples incubated at room temperature for 20 min before heating to 99 °C for 5 min. DNA was sheared by sonication using a Covaris S2 high-performance ultrasonicator. Cell debris was removed by centrifugation at 16,000g for 15 min at 20 °C. Supernatant was transferred to fresh tubes and protein concentration determined using the BCA protein assay kit (Pierce Biotechnology). Filter-aided sample preparation was performed using a 30 kDa molecular weight cut-off centrifugal filters (Microcon 30, Ultracel YM-30, Merck Millipore) as previously described[58]. In brief, 200 μg of total protein per sample was reduced by the addition of DTT to a final concentration of 83.3 mM, followed by incubation at 99 °C for 5 min. Samples were mixed with 200 μl freshly prepared 8 M urea in 100 mM Tris-HCl (pH 8.5) (UA solution) in the filter unit and centrifuged at 14,000g for 15 min at 20 °C to remove SDS. Residual SDS was washed out by a second wash step with 200 μl UA. Proteins were alkylated with 100 μl of 50 mM iodoacetamide in the dark for 30 min at room temperature. Thereafter, three washes were performed with 100 μl of UA solution, followed by three washes with 100 μl of 50 mM TEAB buffer (Sigma-Aldrich). Proteolytic digestion was performed using trypsin (1:50) overnight at 37 °C. Peptides were recovered using 40 μl of 50 mM TEAB buffer followed by 50 μl of 0.5 M NaCl. Peptides were desalted using Pierce Peptide Desalting Spin Columns (Thermo Scientific). TMTpro 16plex Label Reagent Set was used for labelling according to the manufacturer's instructions (Pierce). After the labelling reaction was quenched, the samples were pooled, the organic solvent removed in a vacuum concentrator, and the labelled peptides purified by C18 solid phase extraction.

For offline fractionation via reverse phase high-performance liquid chromatography (HPLC) at high pH as previously described[59], tryptic peptides were re-buffered in 10 mM ammonium formate buffer (pH 10). Peptides were separated into 96 time-based fractions on a Phenomenex C18 reverse phase column (150 × 2.0 mm Gemini-NX, 3 μm C18 110 Å, Phenomenex) using an Agilent 1200 series HPLC system fitted with a binary pump delivering solvent at 50 μl min⁻¹. Acidified fractions were consolidated into 36 fractions via a concatenated strategy as previously described[59]. After removal of solvent in a vacuum concentrator, samples were reconstituted in 0.1% TFA prior to liquid chromatography–mass spectrometry (LC–MS/MS) analysis.

Mass spectrometry analysis was performed on an Orbitrap Fusion Lumos Tribrid mass spectrometer coupled to a Dionex Ultimate 3000 RSLCnano system (via a Nanospray Flex Ion Source) (all Thermo Fisher Scientific) interface and operated via Xcalibur (v4.3.73.11) and Tune

(v3.4.3072.18). Peptides were loaded onto a trap column (PepMap 100 C18, 5 μm, 5 × 0.3 mm, Thermo Fisher Scientific) at a flow rate of 10 μl min$^{-1}$ using 0.1% TFA as loading buffer. After loading, the trap column was switched inline with an Acclaim PepMap nanoHPLC C18 analytical column (2.0 μm particle size, 75 μm internal diameter × 500 mm, 164942, Thermo Fisher Scientific). The column temperature was maintained at 50 °C. Mobile phase A consisted of 0.4% formic acid in water, and mobile phase B consisted of 0.4% formic acid in a mixture of 90% acetonitrile and 10% water. Separation was achieved using a 4-step gradient over 90 min at a flow rate of 230 nl min$^{-1}$. In the liquid junction setup, electrospray ionization was enabled by applying a voltage of 1.8 kV directly to the liquid being sprayed, and non-coated silica emitter was used. The mass spectrometer was operated in a data dependent acquisition (DDA) mode using a maximum of 20 dependent scans per cycle. Full MS1 scans were acquired in the Orbitrap with a scan range of 400–1,600 $m/z$ and a resolution of 120,000 at 200 $m/z$. Automatic gain control (AGC) was set to 'standard' and a maximum injection time (IT) of 50 ms was applied. MS2 spectra were acquired in the Orbitrap at a resolution of 50,000 at 200 $m/z$ with a fixed first mass of 100 $m/z$. To achieve maximum proteome coverage, a classical tandem MS approach was chosen instead of the available synchronous precursor selection (SPS)-MS3 approach. To minimize TMT ratio compression effects by interference of contaminating co-eluting isobaric peptide ion species, precursor isolation width in the quadrupole was set to 0.5 Da and an extended fractionation scheme applied. Monoisotopic peak determination was set to 'peptides' with inclusion of charge states between 2 and 5. Intensity threshold for MS2 selection was set to $2.5 \times 10^4$. Higher energy collision induced dissociation (HCD) was applied with a normalized collision energy (NCE) of 34%. Normalized AGC was set to 200% with a maximum injection time of 86 ms. Dynamic exclusion for selected ions was 90 s.

The acquired raw data files were processed using Proteome Discoverer (v.2.4.1.15), via the TMT16plex quantification method. Sequest HT database search engine and the Percolator validation software node were used to remove false positives with FDR 1% at the peptide and protein level. All MS/MS spectra were searched against the human proteome (Canonical, reviewed, 20 304 sequences) and appended known contaminants and streptavidin, with a maximum of two allowable miscleavage sites. The search was performed with full tryptic digestion with or without deamidation on amino acids asparagine, glutamine, and arginine. Methionine oxidation and protein N-terminal acetylation, as well as methionine loss and protein N-terminal acetylation with methionine loss were set as variable modifications, while carbamidomethylation of cysteine residues and tandem mass tag (TMT) 16-plex labelling of peptide N termini and lysine residues were set as fixed modifications. Data were searched with mass tolerances of ±10 ppm and ±0.025 Da for the precursor and fragment ions, respectively. Results were filtered to include peptide spectrum matches with Sequest HT cross-correlation factor (Xcorr) scores of ≥1 and high peptide confidence assigned by Percolator. MS2 signal-to-noise (S/N) values of TMTpro reporter ions were used to calculate peptide or protein abundance values. Peptide spectrum matches with precursor isolation interference values of ≥70% and average TMTpro reporter ion S/N ≤ 10 were excluded from quantification. Both unique and razor peptides were used for TMT quantification. Correction of isotopic impurities was applied.

Data were normalized to total peptide abundance and scaled 'to all average'. Abundances were compared to DMSO-treated cells and protein ratios were calculated from the grouped protein abundances using an ANOVA hypothesis test. Adjusted $P$ values were calculated using the Benjamini–Hochberg method. Proteins with less than three unique peptides detected were excluded from downstream analysis.

### Protein construction, expression and purification
His$_6$–TEV–BRD4 bromodomain 1 (BRD4$^{BD1}$) (amino acids 44–178) and His$_6$–TEV–BRD4 bromodomain 2 (BRD4$^{BD2}$) (amino acids 333–460) were expressed in *Escherichia coli* BL21(DE3) and purified as described

previously[60]. In brief, proteins were purified by nickel affinity chromatography and SEC. His$_6$ tag cleavage and reverse nickel affinity was performed prior to SEC for some applications, for others the tag was left on. Purified proteins in 20 mM HEPES, 150 mM sodium chloride, 1 mM DTT, pH 7.5 were aliquoted and flash frozen in liquid nitrogen and stored at −80 °C.

His$_6$–SUMO–TEV–BRD4$^{Tandem}$ (residues 1–463) was prepared as previously described[51]. In brief, protein was expressed in *E. coli* BL21(DE3) and purified sequentially by nickel affinity on a HisTrap HP 5 ml column (Cytiva), His$_6$ tag cleavage by SENP1 followed by reverse nickel affinity, cation exchange on a HiTrap SP HP 5 ml column (Cytiva), and size exclusion on a HiLoad 16/600 Superdex 200 pg column (Cytiva). Purified protein in 20 mM HEPES, 100 mM sodium chloride, 1 mM TCEP, pH 7.5 was aliquoted and flash frozen in liquid nitrogen then stored at −80 °C.

BRD4$^{Tandem}$ (residues 43–459) was cloned into pRSF-DUET or a modified pGEX4T1 with an N-terminal His$_{10}$ tag and HRV3C cleavage site or a His$_{12}$-GST tag and TEV cleavage site, respectively.

His$_{10}$–3C-BRD4$^{Tandem}$ (residues 43–459) was transformed into *E. coli* BL21(DE3) and overnight expression at 18 °C was induced with 0.35 mM IPTG at OD$_{600}$ ~ 0.8–1. Cells were collected by centrifugation and pellets were resuspended in ice-cold PBS then spun down again. Supernatant was removed and pellets were flash frozen in liquid nitrogen and stored at −80 °C. Cells were thawed and resuspended in lysis buffer (50 mM HEPES, 500 mM NaCl, 0.5 mM TCEP, pH 7.5) supplemented with 2 mM magnesium chloride, DNAse and cOmplete EDTA-free Protease Inhibitor Cocktail (Roche, 1 tablet per litre initial culture volume) and lysed at 30,000 psi using a CF1 Cell Disruptor (Constant Systems). The lysate was cleared by centrifugation at 20,000 rpm for 30 min at 4 °C then syringe-filtered using a 0.45-μm filter. The lysate was supplemented with 40 mM imidazole and loaded on to a 5 ml HisTrap HP column (Cytiva) equilibrated in lysis buffer with 40 mM imidazole, washed at 60 mM imidazole and eluted with a gradient up to 100% elution buffer (50 mM HEPES, 500 mM NaCl, 0.5 mM TCEP, 500 mM imidazole, pH 7.5). The prep was split as required for tag cleavage or for purification of the His$_{10}$–3C-tagged form. For tag cleavage, the sample was buffer exchanged into lysis buffer on a HiPrep 26/10 Desalting column and HRV3C protease was added to cleave the tag overnight at 4 °C. Imidazole was added to 20 mM to the cleaved BRD4$^{Tandem}$ and the sample was run on a 5 ml HisTrap HP column equilibrated in lysis buffer with 20 mM imidazole and washed with the same imidazole concentration. The flow-through and wash containing BRD4$^{Tandem}$ were pooled and, along with uncleaved His$_{10}$–3C-BRD4$^{Tandem}$, were concentrated in 10,000 MWCO Amicon centrifugal filter units (Merck Millipore). The proteins were each loaded separately onto a HiLoad 26/600 Superdex 200 pg column (GE LifeSciences) equilibrated in 20 mM HEPES, 150 mM NaCl, 0.5 mM TCEP, pH 7.5. Fractions containing either pure BRD4$^{Tandem}$ or His$_{10}$–3C-BRD4$^{Tandem}$ were confirmed by SDS–PAGE, then pooled, concentrated and aliquoted for storage at −80 °C until use.

For use in cryo-electron microscopy (cryo-EM) with DCAF16 and IBG1, His$_{12}$–GST–TEV–BRD4$^{Tandem}$ (residues 43–459) expression in *E. coli* BL21(DE3) cells was induced at OD$_{600}$ = 2 with 0.5 mM IPTG at 20 °C for 16 h. Cells were collected by centrifugation and resuspended in lysis buffer (50 mM HEPES, 500 mM NaCl, 20 mM imidazole, 0.5 mM TCEP, pH 7.5) (10 ml g$^{-1}$ pellet weight) supplemented with DNAse and 1 cOmplete EDTA-free Protease Inhibitor Cocktail tablet (Roche) per 2 l of culture. Cells were lysed at 30,000 psi using a CF1 Cell Disruptor (Constant Systems) and lysate was clarified by centrifugation. Lysate was filtered through a BioPrepNylon Matrix Filter (BioDesign) then incubated with 1 ml Ni-NTA resin per litre culture for 1 h. The lysate–resin slurry was poured into a Bio-Rad Econo-column and resin was washed with >10 column volumes lysis buffer. Bound protein was eluted with elution buffer (50 mM HEPES pH 7.5, 150 mM NaCl, 500 mM imidazole, 0.5 mM TCEP) then incubated with 1 ml glutathione agarose resin per litre culture for 30 min. The mixture was poured into an Econo-column and resin was washed with 20 mM HEPES, 150 mM NaCl, 0.5 mM TCEP,

pH 7.5. TEV protease was added to the resin slurry for on-bead cleavage and the column was incubated overnight on a roller at 4 °C. Protein was eluted from the column then concentrated and run on a HiLoad 16/600 Superdex 75 pg column equilibrated in 20 mM HEPES, 150 mM NaCl, 0.5 mM TCEP, pH 7.5. Fractions containing protein were pooled, concentrated and aliquoted then flash frozen in liquid nitrogen then stored at −80 °C until use.

A DCAF15 construct lacking the proline-rich region (amino acids 276–380; DCAF15Δpro) with N-terminal His$_6$-TEV-Avi tag, DDB1(ΔBPB) (residues 396–705 replaced with a GNGNSG linker), and full-length DDA1 coding sequences were cloned into a pFastBacDual vector. Bacmid was generated using the Bac-to-Bac baculovirus expression system (Thermo Fisher Scientific). Baculovirus was generated via an adapted single-step protocol[61,62]. In brief, bacmid (1 μg ml$^{-1}$ culture volume) was mixed with 2 μg PEI 25 K (Polysciences) per μg bacmid in 200 μl warm PBS and incubated at room temperature for 30 min. The mixture was added to a suspension culture of Sf9 cells at $1 \times 10^6$ cells per ml in Sf-900 II SFM (Gibco) and incubated at 27 °C with shaking at 110 rpm. Viral supernatant (P0) was collected after 4–6 days. For expression, *Spodoptera frugiperda* cells (Sf9) were grown to densities between 1.9 to $3.0 \times 10^6$ cells per ml in Sf-900 II SFM (Gibco) and infected with a total virus volume of 1% per $1 \times 10^6$ cells per ml. Cells were incubated at 27 °C in 2 l Erlenmeyer flasks (~500 ml culture per flask) with shaking at 110 rpm for 48 h. Cells were spun at 1,000$g$ for 10 min and supernatant was discarded. Pellets were resuspended in lysis buffer (50 mM HEPES, 200 mM NaCl, 2 mM TCEP, pH 7.5) with magnesium chloride (to 2 mM), benzonase (to 1 μg ml$^{-1}$) and cOmplete EDTA-free Protease Inhibitor Cocktail (Roche, 2 tablets per litre initial culture volume). The suspension was frozen and stored at −80 °C, and then thawed. Cell suspensions were sonicated and lysates were centrifuged at 40,000 rpm for 30 min. The supernatant was incubated with 1.5 ml Ni-NTA agarose resin (Qiagen) on a roller at 4 °C for 1.5 h. The lysate–resin slurry was loaded into a glass bench top column. Supernatant was allowed to flow through then the resin was washed with wash buffer (50 mM HEPES, 200 mM NaCl, 2 mM TCEP, 20 mM imidazole, pH 7.5). Bound protein was eluted with elution buffer (50 mM HEPES pH 7.5, 200 mM NaCl, 2 mM TCEP, 500 mM imidazole). TEV protease was added to protein and dialysed with buffer (50 mM HEPES, 200 mM NaCl, 2 mM TCEP, pH 7.5). Cleaved protein was run over 1.5 ml Ni-NTA agarose resin and the flow-through and washes with binding buffer were collected and pooled. Protein was diluted with buffer (25 mM HEPES, 2 mM TCEP, pH 7.5) to adjust the NaCl concentration to 50 mM, then loaded onto a HiTrap Q HP 5 ml column (Cytiva). The column was washed with IEX buffer A and bound protein was eluted with a 0–100% IEX buffer B (25 mM HEPES, 1 M NaCl, 2 mM TCEP, pH 7.5) gradient. Fractions containing protein were pooled and concentrated to ~1–2 ml then run on 16/600 Superdex 200 pg column in GF buffer (25 mM HEPES, 300 mM NaCl, 1 mM TCEP, pH 7.5). Fractions containing the purified protein complex were pooled, concentrated and aliquoted then flash frozen in liquid nitrogen for storage at −80 °C.

The coding sequences for full-length DCAF16 or DCAF11 with TEV-cleavable N-terminal His$_6$-tags were cloned into a pFastBacDual vector under the control of the *polh* promoter. Coding sequences for full-length DDB1 or DDB1(ΔBPB) and full-length DDA1 were cloned into a pFastBacDual vector under the control of polh and p10 promoters, respectively. Bacmid was generated using the Bac-to-Bac baculovirus expression system (Thermo Fisher Scientific). Baculovirus was generated as described above and viral supernatant (P0) was collected after 5–7 days. For expression, *Trichoplusia ni* High Five cells were grown to densities between 1.5 to $2 \times 10^6$ cells per ml in Express Five SFM (Gibco) supplemented with 18 mM L-glutamine and infected with a total virus volume of 1% per $1 \times 10^6$ cells per ml, consisting of equal volumes of DCAF16/DCAF11 and DDB1 + DDA1 baculoviruses. Cells were incubated at 27 °C in 2 l Erlenmeyer flasks (~600–650 ml culture per flask) with shaking at 110 rpm for 72 h. Cells were spun at 1,000$g$ for 20 min and supernatant was discarded. Pellets were resuspended in 25 ml binding buffer (50 mM HEPES, 500 mM NaCl, 1 mM TCEP, pH 7.5), flash frozen in liquid nitrogen and stored at −80 °C. Pellets were thawed and diluted with binding buffer to ~100 ml l$^{-1}$ original culture volume. Tween-20 (to 1% (v/v)), magnesium chloride (to 2 mM), benzonase (to 1 μg ml$^{-1}$) and cOmplete EDTA-free Protease Inhibitor Cocktail (Roche, 2 tablets per litre initial culture volume) were added to the cell suspension and stirred at room temperature for 30 min. Cell suspensions were sonicated, and lysates were centrifuged at 23,000 rpm for 60 min. Supernatants were filtered through 0.45-μm filters and supplemented with 10 mM imidazole then incubated with 2 ml cobalt agarose resin per litre culture on a roller at 4 °C for 1 h. The lysate–resin slurry was loaded into a glass bench top column. Supernatant was allowed to flow through then the resin was washed with wash buffer (50 mM HEPES, 500 mM NaCl, 1 mM TCEP, 15 mM imidazole, pH 7.5). Bound protein was eluted with elution buffer (50 mM HEPES, 500 mM NaCl, 1 mM TCEP, 250 mM imidazole, pH 7.5) and buffer exchanged on a 26/10 HiPrep Desalting column (Cytiva) into Binding Buffer. TEV protease was added to protein and incubated for 2 h at room temperature then 4 °C overnight. Imidazole was added to the cleaved protein to a concentration of 10 mM and the sample was run over cobalt agarose resin. Flow-through and washes with binding buffer supplemented with 10 mM imidazole were collected and pooled. Protein was buffer exchanged into ion exchange (IEX) buffer A (50 mM HEPES, 50 mM NaCl, 1 mM TCEP, pH 7.5) on a 26/10 HiPrep Desalting column then loaded onto a HiTrap Q HP 5 ml column (Cytiva). The column was washed with IEX buffer A and bound protein was eluted with a 0–100% IEX buffer B (50 mM HEPES, 1 M NaCl, 1 mM TCEP, pH 7.5) gradient. Fractions containing protein were pooled and concentrated then run on 16/600 Superdex 200 pg column in equilibrated in 20 mM HEPES, 150 mM NaCl, 1 mM TCEP, pH 7.5. Fractions containing the purified protein complex were pooled and concentrated then aliquoted and flash frozen in liquid nitrogen for storage at −80 °C.

## Sulfo-Cy5 NHS ester labelling

For DCAF16 labelling, sulfo-Cy5 NHS ester (Lumiprobe) in DMF was prepared to a final concentration of 800 μM with DCAF16–DDB1(ΔBPB)–DDA1 (100 μM) and sodium bicarbonate (100 mM). For DCAF11 labelling, sulfo-Cy5 NHS ester (Lumiprobe) in DMF was prepared to a final concentration of 1 mg ml$^{-1}$ with DCAF11–DDB1(ΔBPB)–DDA1 (1 mg ml$^{-1}$) and sodium bicarbonate (100 mM). The solutions were protected from light and shaken for 1 h at room temperature. The solutions were spun down at 15,000$g$ for 5 min then run on a Superdex 200 10/300 GL column (Cytiva) to remove free dye and aggregated protein. Fractions containing the sulfo-Cy5-labelled protein were pooled and concentrated, the degree of labelling was calculated to be greater than 100% for each batch of labelled protein. Labelled protein was aliquoted then flash frozen in liquid nitrogen and stored at −80 °C.

## Fluorescence polarization assay

Stock solutions of reaction components including DCAF15(Δpro)–DDB1(ΔBPB)–DDA1, DCAF16–DDB1(ΔBPB)–DDA1, His$_6$–BRD4$^{BD1}$, His$_6$–BRD4$^{BD2}$, BRD4$^{Tandem}$ (residues 43–459), and FITC-sulfonamide probe[7] were prepared in FP assay buffer (25 mM HEPES pH 7.5, 300 mM NaCl, 1.0 mM TCEP). DCAF15(Δpro)–DDB1(ΔBPB)–DDA1, DCAF16–DDB1(ΔBPB)–DDA1, BRD4$^{BD1}$, BRD4$^{BD2}$ and BRD4$^{Tandem}$ were titrated 1:3 in FP assay buffer. Components were added to Corning 384-Well solid black polystyrene microplates to a final volume of 15 μl. Final concentration of 20 nM for FITC-sulfonamide probe was used while DCAF15(Δpro)–DDB1(ΔBPB)–DDA1, DCAF16–DDB1(ΔBPB)–DDA1, BRD4$^{BD1}$, His$_6$–BRD4$^{BD2}$ and BRD4$^{Tandem}$ were titrated from 4 μM to 5.5 nM. Background subtraction was performed with 20 nM FITC-sulfonamide probe and no protein constructs. Components were mixed by spinning down plates at 50$g$ for 1 min and the plate was covered and incubated at room temperature for 1 h, before analysis on a PHERAstar FS (BMG LABTECH) with fluorescence excitation and emission wavelengths of 485 and 520 nm, respectively, with a settling time of 0.3 s.

## AlphaLISA displacement assay

The alphaLISA assays were performed as described previously[51] using $His_6$-BRD4[BD1], $His_6$-BRD4[BD2] or $His_{10}$-BRD4[Tandem] and the biotinylated JQ1 probe. Assay conditions in the present work used were as follows: 100 nM bromodomain protein, 10 nM Bio-JQ1 probe, 25 µg ml$^{-1}$ acceptor (nickel chelate) and donor (anti-His–europium; both PerkinElmer). All components were diluted to working concentrations in alphaLISA buffer (50 mM HEPES, 100 mM NaCl, 0.1% BSA, 0.02% CHAPS, pH 7.5). Bromodomain protein was co-incubated with test compounds using 384-well AlphaPlates (PerkinElmer) in the absence or presence of DCAF16 (1 µM) for 1 h, before adding the acceptor and donor beads simultaneously in a low light environment and incubating the plate at room temperature for a further 1 h. The plate was then read on a BMG Pherastar equipped with an alphaLISA module. Data were normalized to a DMSO control and expressed as % bound vs log[concentration] of compound and analysed by non-linear regression, with extraction of binding affinity values ($IC_{50}$) from the curves. Where applicable, $K_d$ values were calculated from a titration of bromodomain protein on the same assay plate alone into the probe, as described previously[63].

## TR-FRET proximity assay

Stock solutions of reaction components including sulfo-Cy5-labelled DCAF16–DDB1(ΔBPB)–DDA1, sulfo-Cy5-labelled DCAF11–DDB1(ΔBPB)–DDA1, $His_6$-BRD4[BD1], $His_{10}$-BRD4[BD2], $His_{10}$-BRD4[Tandem], experimental compounds and LANCE Eu-W1024 Anti-$His_6$ donor (PerkinElmer) were prepared in TR-FRET assay buffer (50 mM HEPES pH 7.5, 100 mM NaCl, 1 mM TCEP, 0.05% Tween-20). Two types of TR-FRET assay were performed: titration of compound into protein (complex-formation assay) and titration of sulfo-Cy5-labelled DCAF into BRD4 vs BRD4–compound (complex-stabilization assay). For the former, compounds were titrated 1:4 into 100 nM BRD4 and 100 nM Cy5-DCAF to a PerkinElmer OptiPlate-384 (white) to a final well volume of 16 µl. For the complex-stabilization assay, sulfo-Cy5-labelled DCAF16–DDB1(ΔBPB)–DDA1 or DCAF11–DDB1(ΔBPB)–DDA1 were titrated 1:4 and 1:3 respectively in TR-FRET assay buffer. Components were added to PerkinElmer OptiPlate-384 (white) to a final well volume of 16 µl. Final concentrations of 100 or 200 nM for BRD4 constructs and 0.5 µM or 1 µM for IBG1 respectively were used. LANCE Eu-W1024 anti-$His_6$ donor and DMSO concentrations were kept constant across the plate for both assay formats at 2 nM and 0.5%, respectively. Background subtraction was performed with using concentration matched samples containing sulfo-Cy5-labelled DCAF complexes but not BRD4. Components were mixed by spinning down plates at 50$g$ for 1 min and plates were covered and incubated at room temperature for 30 min. Plates were read on a PHERAstar FS (BMG LABTECH) with fluorescence excitation and dual emission wavelengths of 337 and 620/665 nm, respectively, with an integration time between 70 and 400 µs. Data were processed in GraphPad Prism (v9.3.1), curve fitting for the IBG1 curve was performed by setting the maximum as DMSO-only 5 µM sulfo-Cy5-labelled DCAF16–DDB1(ΔBPB)–DDA1 datapoint.

## Analytical SEC

For DCAF16 experiments, DCAF16–DDB1(ΔBPB)–DDA1, BRD4[Tandem] (residues 1–463), BRD4[BD1] ($His_6$ tag removed), BRD4[BD2] ($His_6$ tag removed), and IBG1 were incubated alone and in various combinations in buffer (20 mM HEPES, 150 mM NaCl, 1 mM TCEP, 2% DMSO, pH 7) on ice for 50 min. Final concentrations used for Fig. 4a and Extended Data Fig. 4a were 10 µM DCAF16–DDB1(ΔBPB)–DDA1, 5 µM BRD4[Tandem], 25 µM IBG1 in 250 µl reaction volumes. Final concentrations used for Fig. 4b were 5 µM DCAF16–DDB1(ΔBPB)–DDA1, 5 µM BRD4[Tandem], 5 µM BRD4[BD1], 5 µM BRD4[BD2], 12.5 µM IBG1 in 200 µl reaction volumes. Samples were run on a Superdex 200 Increase 10/300 gl column in 20 mM HEPES, 150 mM NaCl, 1 mM TCEP, pH 7.

For DCAF11 experiments, DCAF11–DDB1(ΔBPB)–DDA1, BRD4[Tandem] (residues 43–463) and IBG4 were incubated alone and in various combinations in buffer (20 mM HEPES, 150 mM NaCl, 0.5 mM TCEP, 2% DMSO, pH 7.5) at final concentrations of 5 µM, 5 µM and 10 µM, respectively. Samples were run on a Superdex 200 Increase 10/300 gl column in 20 mM HEPES, 150 mM NaCl, 0.5 mM TCEP, pH 7.5.

For BRD4 intramolecular dimerization experiments, BRD4[Tandem] (residues 43–463) and compounds were incubated in buffer (20 mM HEPES, 150 mM NaCl, 0.5 mM TCEP, 2% DMSO, pH 7.5) at final concentrations of 5 µM and 10 µM, respectively. Samples were run on a Superdex 200 Increase 10/300 gl column in 20 mM HEPES, 150 mM NaCl, 0.5 mM TCEP, pH 7.5.

## Isothermal titration calorimetry

Titration experiments were performed with an ITC200 instrument (Malvern) in 100 mM Bis-tris propane, 50 mM NaCl, 0.5 mM TCEP, pH 7.5 at 298 K. Protein samples were prepared by dialysing in buffer in D-Tube Dialyzer Midi, MWCO 6–8 kDa (Millipore). BRD4[Tandem] (residues 43–459) was pre-incubated alone, or with either IBG1 or IBG3 at a 1:1.1 molar ratio for 30 min at room temperature prior to titrations at a DMSO concentration of 2% (v/v). DCAF16–DDB1(ΔBPB)–DDA1 at 2% DMSO (v/v) was titrated into either BRD4[Tandem] alone, pre-complexed BRD4[Tandem]–IBG1 or pre-complexed BRD4[Tandem]–IBG3. The titration consisted of 0.4 µl initial injection (discarded during data analysis) followed by 19 injections of 2 µl at 180 s intervals between injections. Data were fitted using a one-set-of-site binding model to obtain dissociation constant ($K_d$), binding enthalpy ($\Delta H$) and stoichiometry ($N$) using MicroCal PEAQ-ITC Analysis Software1.1.0.1262.

## Cryo-EM sample and grid preparation

Protein complexes for cryo-EM were prepared by first co-incubating BRD4[Tandem] (residues 43–459) with IBG1 in 20 mM HEPES, 50 mM NaCl, 0.5 mM TCEP-HCl, 2% (v/v) DMSO, pH 7.5 for 10 min at room temperature. DCAF16–DDB1(ΔBPB)–DDA1 was added to the mixture to give final concentrations of 14 µM BRD4[Tandem], 14 µM DCAF16–DDB1(ΔBPB)–DDA1 and 35 µM IBG1 in a final reaction volume of 200 µl and incubated on ice for 50 min. The sample was loaded onto a Superdex 200 Increase 10/300 GL column in 20 mM HEPES, 50 mM NaCl, 0.5 mM TCEP-HCl, pH 7.5. Due to incomplete complex formation and to avoid monomeric proteins, only the earliest eluting fraction containing the ternary complex was taken and concentrated to 4.8 µM. Quantifoil R1.2/1.3 Holey Carbon 400 mesh gold grids (Electron Microscopy Sciences) were glow discharged for 60 s with a current of 35 mA under vacuum using a Quorum SC7620. The complex (3.5 µl) was dispensed onto the grid, allowed to disperse for 10 s, blotted for 3.5 s using blot force 3, then plunged into liquid ethane using a Vitrobot Mark IV (Thermo Fisher Scientific) with the chamber at 4 °C and 100% humidity.

## Cryo-EM data acquisition

Cryo-EM data were collected on a Glacios transmission electron microscope (Thermo Fisher) operating at 200 keV. Micrographs were acquired using a Falcon4i direct electron detector, operated in electron counting mode. Movies were collected at 190,000× magnification with the calibrated pixel size of 0.74 Å per pixel on the camera. Images were taken over a defocus range of −3.2 µm to −1.7 µm with a total accumulated dose of 12.7 e$^-$ Å$^{-2}$ using single-particle EPU (Thermo Fisher Scientific, v3.0) automated data software. A total of 2,075 movies were collected in EER format and after cleaning up for large motion and poor contrast transfer function (CTF) a total of 1,896 movies were used for further processing. Cryo-EM data collection, refinement and validation statistics are presented in Extended Data Table 2.

## Cryo-EM image processing

Movies were imported into cryosparc[64] (v4.1.2) and the EER movie data was fractionated into 8 fractions to give a dose of 1.59 e$^-$ Å$^{-2}$ per

fraction. Movies were processed using patch motion correction and CTF correction then manually curated to remove suboptimal movies. Manual picking of 153 particles was performed on 20 micrographs, which were used for blob tuner with minimum and maximum diameters of 70 and 130 Å, respectively. 12,579 particles were picked by blob tuner, extracted with a box size of 324 pix (240 Å) and run through initial 2D classification. Good classes with diverse views were selected and used as templates for template picking on 1,895 movies. Picks were inspected and curated, and 1.35 million particles were extracted with box size 324 pix and used for 2D classification. Particles from the well-resolved, diverse classes were used for ab initio reconstruction with 3 classes. One class contained primarily empty DDB1(ΔBPB) and a second class contained biased views upon testing of the particle set with 2D re-classification, leading to smeared maps. The third class unambiguously contained density corresponding to DDB1(ΔBPB), two bromodomains, and density likely corresponding to DCAF16 between them. Particles belonging to the second and third class were run through heterogenous refinement. The best class yielded a map into which DDB1(ΔBPB) and two bromodomains could be placed with confidence. To improve the resolution, movies were re-imported in cryosparc and fractionated into 18 fractions to give a lower dose of ~0.7 $e^-$ Å$^{-2}$ per fraction. 50 templates for particle picking were generated using the create templates job with the input map from the previous heterogeneous refinement. The templates were used in the template picker to pick particles from 1,132 curated movies with a minimum CTF fit resolution cut-off of 3.5. Picks were curated with thresholds of NCC score > 0.4, local power >368 and <789, resulting in 564,575 particles that were extracted with a box size of 324 pixels and used for ab initio reconstruction with 4 classes. Resulting classes were subjected to a heterogeneous refinement, with one class clearly containing all components of the complex and the others either junk, DDB1(ΔBPB) alone or biased views. The map and particles (192,014) from the best class were used for homogenous refinement with the dynamic mask threshold set to 0.5. Local refinement with a dynamic map threshold of 0.5 produced a map with a gold-standard Fourier shell correlation (GSFSC) resolution of 3.77 Å at cut-off 0.143. The workflow, GSFSC curve, local resolution estimation, angular distribution plot, and posterior position directional distribution plot are presented in Extended Data Fig. 4.

## Cryo-EM model building

DDB1(ΔBPB), BRD4[BD1] and BRD4[BD2] extracted from PDB entries 5FQD[27], 3MXF[65] and 6DUV, respectively, were manually placed into the map in WinCoot[66] (v0.9.8.1) by rigid body fitting. Despite co-purifying with DCAF16 and DDB1(ΔBPB), we did not see density for DDA1, as was observed in another DDB1-substrate receptor structure from a recent publication[67]. Correct placement of each bromodomain was aided by manual inspection of residues Asn93 and Gly386 in equivalent positions in the ZA loops of BD1 and BD2, respectively. In one bromodomain, this position was facing solvent while in the other it was at a protein–protein interface with density corresponding to DCAF16. Given that mutation of Gly386 to Glu prevents degradation of BRD4 by IBG1 (Fig. 3i), BD2 was placed in the position where Gly386 was adjacent to the DCAF16 density. The BD2 ZA loop is three residues longer than the BD1 ZA loop, further confirming the correct positioning of each domain based on the map around these positions. Both bromodomains were joined onto a single chain designation. Initial restraints for IBG1 were generated using a SMILES string with eLBOW (in Phenix v1.20.1-4487)[68], then run through the GRADE webserver (Grade2 v1.3.0). IBG1 was fitted into density by overlaying the JQ1 moiety with its known binding mode in either the BRD4[BD1] or BRD4[BD2]. Positioning the ligand in BD2 was compatible with electron density, whereas positioning in BD1 caused a clash with DCAF16 due to the rigid linker. DCAF16 was built using a combination of models from ColabFold[69,70] (v1.3), ModelAngelo[71] (v0.2.2) and manual building in Coot (v0.9.8.1). ColabFold correctly predicted the α5 and α6 helices that bind the DDB1 central cavity while

ModelAngelo correctly built the 4-helical bundle of α3, 4, 7 and 8, as well as α6 in the DDB1 cavity. Correctly built parts of the models were combined, and the structure was refined with rounds of model building in Coot, fitting with adaptive distance restraints in ISOLDE[72] (v1.6) and refinement with Phenix (v1.20.1-4487) real-space refinement[73,74]. Figures were generated in ChimeraX[75] (v1.6) and The PyMOL Molecular Graphics System[76] (v2.5.2, Schrödinger, LLC).

## Reporting summary

Further information on research design is available in the Nature Portfolio Reporting Summary linked to this article.

## Data availability

Source data for Figs. 1c, 2b,c and 5d and Extended Data Figs. 2a and 6f are included as Supplementary Tables 1–5. Cryo-EM density maps have been deposited in the Electron Microscopy Data Bank (EMDB) with the accession code EMD-17172. The atomic model has been deposited at the Protein Data Bank under accession 8OV6. Quantitative proteomics data have been deposited to the ProteomeXchange Consortium PRIDE repository[77] with the accession ID PXD040570. Full versions of all gels and blots are provided in Supplementary Fig. 1. Schematics of gating strategies applied for FACS analyses and cell sorting are provided in Supplementary Fig. 2. All biological materials are available upon reasonable requests under material transfer agreements (MTA) with The Centre for Targeted Protein Degradation, University of Dundee, or CeMM Research Center for Molecular Medicine of the Austrian Academy of Sciences, respectively.

## Code availability

Code for analysis of FACS-based screens is available on GitHub (https://github.com/ZuberLab/crispr-process-nf, https://github.com/ZuberLab/crispr-mageck-nf).

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

**Acknowledgements** The authors thank all members of the Ciulli and Winter laboratories and J. Konc for experimental advice and helpful discussions, in particular S. Ramachandran, A. Wijaya, Z. Rutter, T. Webb and V. Spiteri for assistance with biophysical assay troubleshooting and productive discussions; M. Nanao for assistance and support in using beamline ID23-2; J. Zuber and members of the Zuber laboratory for sharing iCas9 cell lines, reagents and plasmids, in particular F. Andersch for help with analysing flow cytometry-based CRISPR screens; the Core Facility Flow Cytometry of the Medical University of Vienna for access to flow cytometry instruments and assistance with cell sorting; the CeMM Biomedical Sequencing Facility for NGS sample processing, sequencing and data curation; T. Hannich and the Proteomics Facility of the CeMM Molecular Discovery Platform for access to instruments; D. Cassidy from CeTPD for the assistance with the maintenance of cell lines and support with DNA prep and sequencing; the MRC PPU CRISPR services at the University of Dundee for constructing plasmids used for the generation of knock-in cell lines; N. Larsen at Foghorn Therapeutics for sharing plasmids; and K. Riching at Promega for the gift of the bromodomain conformational sensor plasmid. We acknowledge the European Synchrotron Radiation Facility (ESRF) for provision of synchrotron radiation facilities and the MRC PPU DNA sequencing and services facility for additional cloning and sequencing support. This work was funded by Eisai and by the pharmaceutical companies supporting the Division of Signal Transduction and Therapy (Boehringer Ingelheim, GlaxoSmithKline, Merck KaaG) as sponsored research funding to A.C. Funding is also gratefully acknowledged from the European Union's Horizon 2020 research and innovation programme under the Marie Skłodowska-Curie grant agreement no. 101024945 (H2020-MSCA-IF-2020-101024945 DELETER, Marie Skłodowska-Curie Actions Individual Fellowship to A.D.C.). The work of the Ciulli laboratory on targeting E3 ligases and TPD has received funding from the European Research Council (ERC) under the European Union's Seventh Framework Programme (FP7/2007-2013) as a Starting Grant to A.C. (grant agreement ERC-2012-StG-311460 DrugE3CRLs), and the Innovative Medicines Initiative 2 (IMI2) Joint Undertaking under grant agreement no. 875510 (EUbOPEN project). The IMI2 Joint Undertaking receives support from the European Union's Horizon 2020 research and innovation programme, European Federation of Pharmaceutical Industries and Associations (EFPIA) companies, and associated partners KTH, OICR, Diamond and McGill. We acknowledge the University of Dundee Cryo-EM facility for access to the instrumentation, funded by Wellcome (223816/Z/21/Z), MRC (MRC World Class Laboratories PO 4050845509). CeMM and the Winter laboratory are supported by the Austrian Academy of Sciences. The Winter laboratory is further supported by funding from the European Research Council (ERC) under the European Union's Horizon 2020 research and innovation programme (grant agreement 851478), as well as by funding from the Austrian Science Fund (FWF, projects P32125, P31690 and P7909).

**Author contributions** O.H., M.H., A.D.C., G.E.W. and A.C. conceived and planned this project. O.H., M.H. and A.D.C. designed and conducted experiments with help from K.I., T.I. and A.R. O.H., M.H., A.D.C., G.E.W. and A.C. analysed and interpreted original data. K.I., T.I., R.C., C.M. and A.T. designed and synthesized compounds. H.I. and C.S. analysed FACS-based CRISPR screens and K.H. and M.W. performed and analysed viability-based CRISPR screens with input from M.K. and I.D. A.R. performed and analysed quantitative expression proteomics. A.D.C. performed cryo-EM imaging, data processing and 3D reconstruction with help from R.S. and M.A.N. M.A.N., A.C.-S., C.C., C.M. and A.T. established critical reagents and methodology. O.H., M.H., A.D.C., G.E.W. and A.C. co-wrote the manuscript with input from all co-authors.

**Competing interests** A.C. is a scientific founder, shareholder and advisor of Amphista Therapeutics, a company that is developing targeted protein degradation therapeutic platforms. The Ciulli laboratory receives or has received sponsored research support from Almirall, Amgen, Amphista Therapeutics, Boehringer Ingelheim, Eisai, Merck KaaG, Nurix Therapeutics, Ono Pharmaceutical and Tocris-Biotechne. A.T. is currently an employee of Amphista Therapeutics. G.E.W. is scientific founder and shareholder of Proxygen and Solgate. The Winter laboratory has received research funding from Pfizer. The other authors declare no competing interests.

**Additional information**
**Correspondence and requests for materials** should be addressed to Georg E. Winter or Alessio Ciulli.

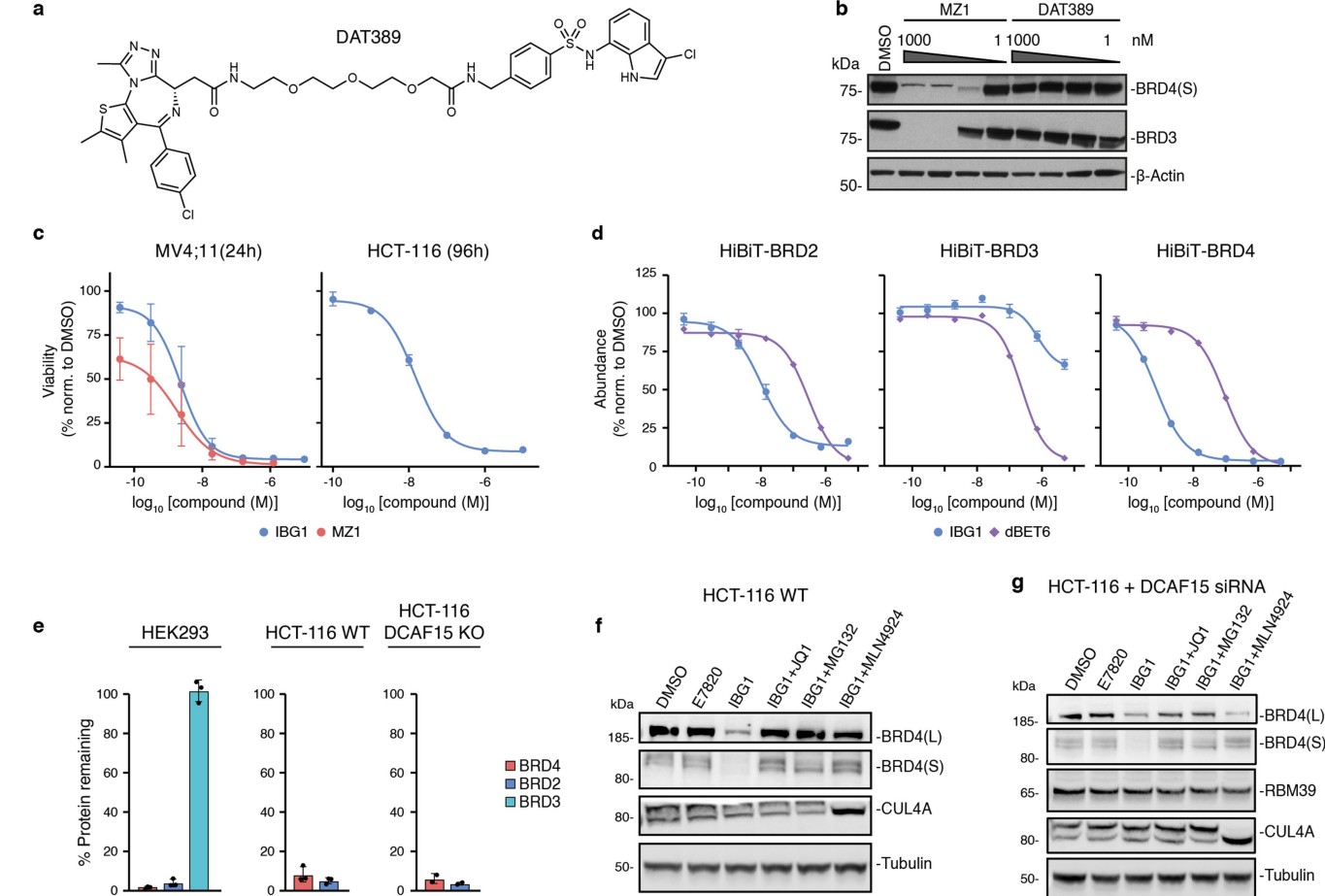

**Extended Data Fig. 1 | IBG1 degrades BRD2 and BRD4 independent of DCAF15. a,b,** Structure (**a**) and BET protein degradation (**b**) of sulfonamide-based PROTAC DAT389. HeLa cells were treated with increasing concentrations of MZ1 or DAT389 for 16 h and BET protein levels were analysed by immunoblot (n = 1). **c,** Cytotoxicity of IBG1 and VHL-based PROTAC MZ1. MV4;11 and HCT-116 cells were treated with increasing concentrations of compounds for 24 or 96 h, respectively, and cell viability was assessed via CellTiterGlo assay. Dose-response curves were fitted using non-linear regression. n = 2 biological replicates, mean +/– s.d. **d,** End-point HiBiT protein degradation. BRD2, BRD3 or BRD4 HiBiT knock-in HEK293 cells were treated with the indicated compounds for 5 h and levels of HiBiT-tagged proteins were quantified via the HiBiT lytic detection system. Dose-response curves were fitted using non-linear regression. n = 3 independent experiments, mean +/– s.d. **e,** Degradation activities of IBG1. BET protein levels were quantified by immunoblotting after compound treatment in HEK293, HCT-116 WT and DCAF15 KO cells. n = 3 independent experiments, mean +/– s.d. Source data, Supplementary Fig. 1. **f,g,** In-cell mechanistic evaluation of IBG1. HCT-116 WT (**f**) or DCAF15 knockdown (**g**) cells were treated for 2 h with E7820 (1 μM) or IBG1 (10 nM) alone, or after 1 h pre-treatment with JQ1 (10 μM), MG132 (50 μM) or MLN4924 (3 μM). Western blot representative of 3 (**f**) or 2 (**g**) independent experiments.

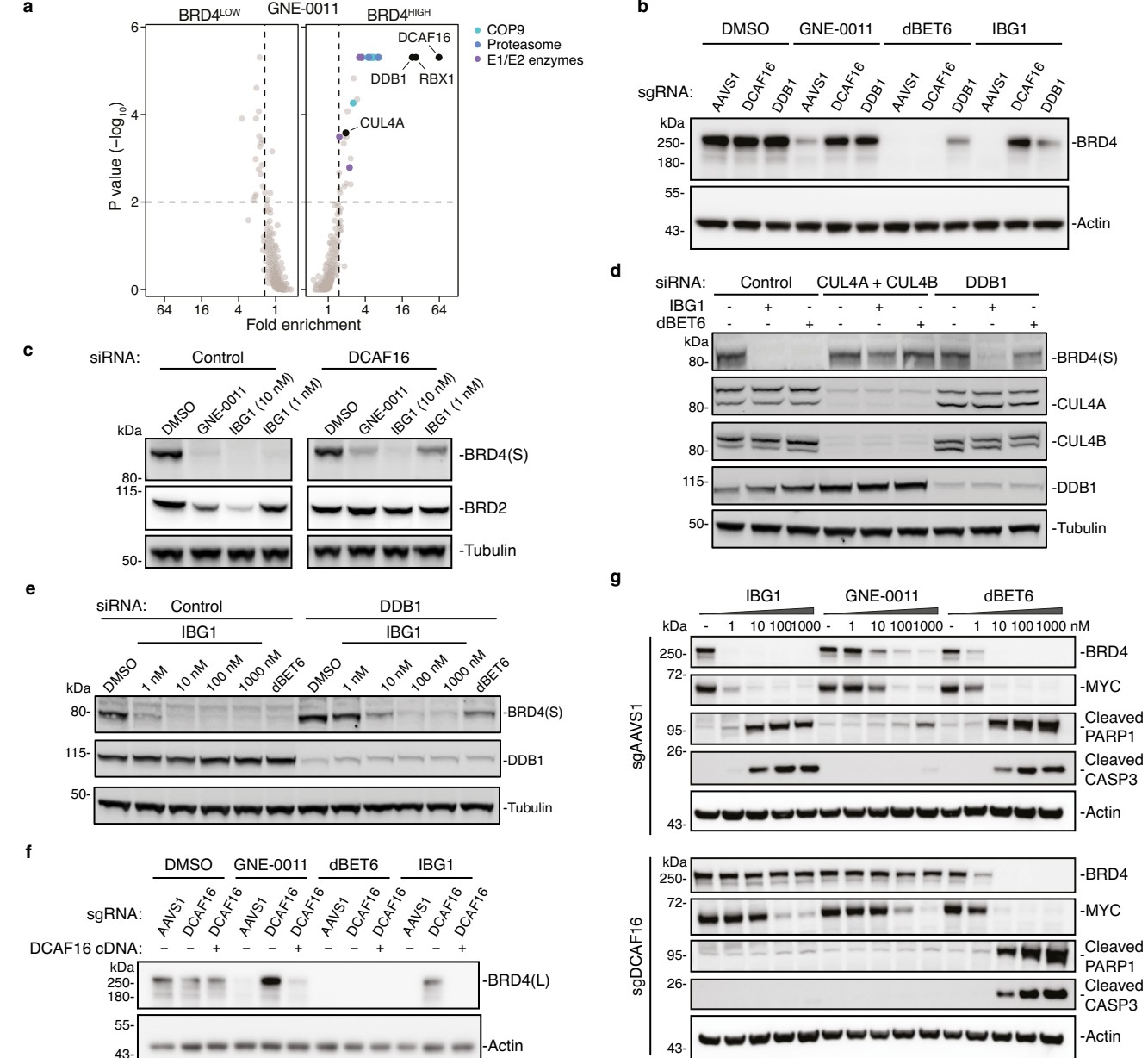

**Extended Data Fig. 2 | IBG1 degrades BRD2/4 via CRL4$^{DCAF16}$. a**, BRD4 stability CRISPR screen. KBM7 iCas9 BRD4 dual fluorescence reporter cells expressing a CRL-focused sgRNA library were treated with GNE-0011 (1 μM) for 6 h before flow cytometric cell sorting into BRD4$^{low}$, BRD4$^{mid}$ and BRD4$^{high}$ fractions as in Fig. 2b. 20 S proteasome subunits (blue), COP9 signalosome subunits (cyan) and E1 or E2 ubiquitin enzymes (purple) inside the scoring window (one-sided MAGeCK p-value < 0.01, fold-change > 1.5; dashed lines) are highlighted. **b–f**, Immunoblot-based CRISPR/Cas9 screen validation. **b**, CRISPR-based validation. KBM7 iCas9 cells were lentivirally transduced with sgRNAs targeting AAVS1, DCAF16 or DDB1 and 3 days after Cas9 induction, cells were treated with GNE-0011 (1 μM), dBET6 (10 nM) or IBG1 (1 nM) for 6 h and BRD4 levels were analysed via immunoblot. Data are representative of n = 2 independent experiments. **c–e**, siRNA-based validation. HCT-116 cells were

transfected with siRNA pools targeting the indicated genes and treated with DMSO, IBG1, GNE-0011 or dBET6 for 2 h at the indicated concentrations and BET protein levels were analysed via immunoblotting. Data are representative of n = 2 independent experiments. **f**, DCAF16 knockout/rescue. KBM7 iCas9 cells were lentivirally transduced with *DCAF16*-targeting or *AAVS1* control sgRNAs, as well as a *DCAF16* cDNA in which the sgRNA target sites were removed by synonymous mutations. After knockout of endogenous DCAF16 and compound treatment for 6 h as above, BRD4 expression levels were assessed via immunoblotting (n = 1). **g**, Induction of apoptosis. KBM7 iCas9 WT or DCAF16 knockout cells were treated with increasing concentrations of IBG1, GNE-0011 or dBET6 for 16 h and levels of BRD4, MYC, cleaved PARP1 and cleaved caspase 3 were analysed via immunoblotting as in Fig. 2g. Data are representative of n = 3 independent experiments.

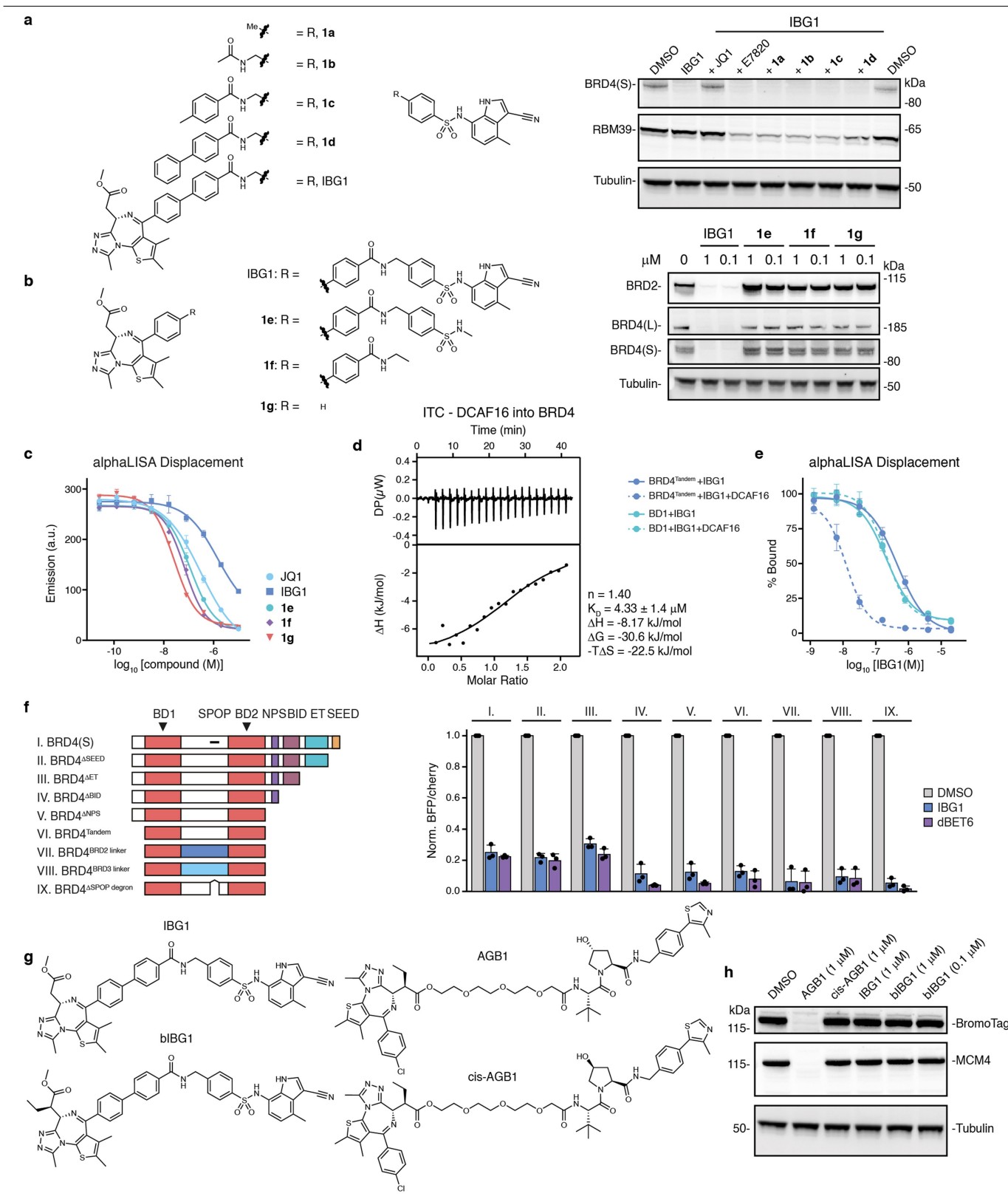

**Extended Data Fig. 3** | See next page for caption.

**Extended Data Fig. 3 | Mechanistic evaluation of IBG1 mechanism of action.**
**a**, Competitive degradation assay. HCT-116 cells were pre-treated for 1 h with 10 µM of sulfonamide-containing truncations of IBG1 (compounds **1a**–**d**), followed by 2-hour treatment with IBG1 (10 nM) and immunoblot analysis. Data is representative of n = 2 independent experiments. **b**, Degradation activities of JQ1-containing truncations of IBG1. HCT-116 cells were treated with indicated concentrations of JQ1-containing truncations of IBG1 (compounds **1e**–**g**) for 6 h and analysed by immunoblotting. Data is representative of n = 2 independent experiments. **c**, alphaLISA displacement assay. His-BRD4$^{BD2}$ preincubated with a biotinylated JQ1 probe was titrated against increasing concentrations of IBG1 or truncated compounds **1e**–**g**. n = 3 technical replicates, mean +/− s.d. **d**, Isothermal titration calorimetry measurement of DCAF16-DDB1ΔBPB-DDA1 binding to BRD4$^{Tandem}$ (n = 1). **e**, alphaLISA displacement assay. His-BRD4$^{Tandem}$ or His-BRD4$^{BD1}$ were preincubated with a biotinylated JQ1 probe and titrated against increasing concentrations of IBG1 in the presence or absence of DCAF16. n = 2 independent experiments each with 3 technical replicates, mean +/− s.d. **f**, Protein stability reporter assay. WT or truncated forms of BRD4 fused to mTagBFP (left) were stably expressed in KBM7 cells and after 6-hour treatment with DMSO, IBG1 (1 nM) or dBET6 (10 nM) protein stability was quantified via flow cytometric evaluation of the mTagBFP/mCherry ratio (right). BD, bromodomain; NPS, N-terminal phosphorylation sites; BID, basic residue-enriched interaction domain; ET, extraterminal domain; SEED, Serine/Glutamic acid/Aspartic acid-rich region. n = 3 independent experiments, mean +/− s.d. **g**,**h**, BromoTag degradation. HEK293 cells stably expressing BromoTag-MCM4 were treated for 5 h with DMSO, BromoTag degrader AGB1 and non-degrader *cis*-AGB1, IBG1, or 'bumped' IBG1 analogue bIBG1 (**g**) and BromoTag-MCM4 levels were analysed by immunoblotting (**h**). Data representative of n = 2 independent experiments.

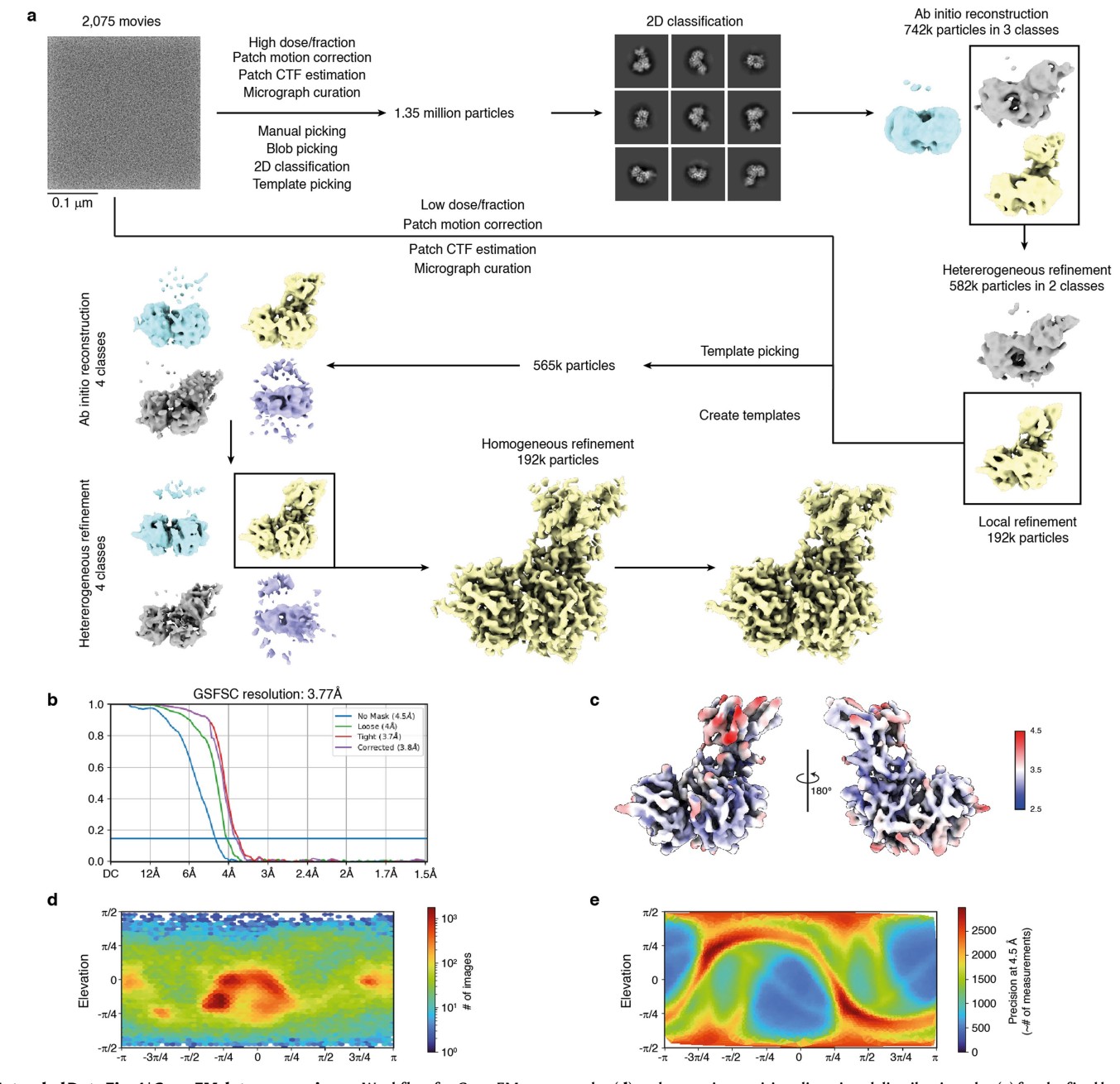

**Extended Data Fig. 4 | Cryo-EM data processing. a**, Workflow for Cryo-EM data processing. **b**, Gold-standard Fourier shell correlation at a cut-off of 0.143. **c**, Local resolution estimation on the unsharpened map. **d**,**e**, Angular distribution plot (**d**) and posterior position directional distribution plot (**e**) for the final local refinement.

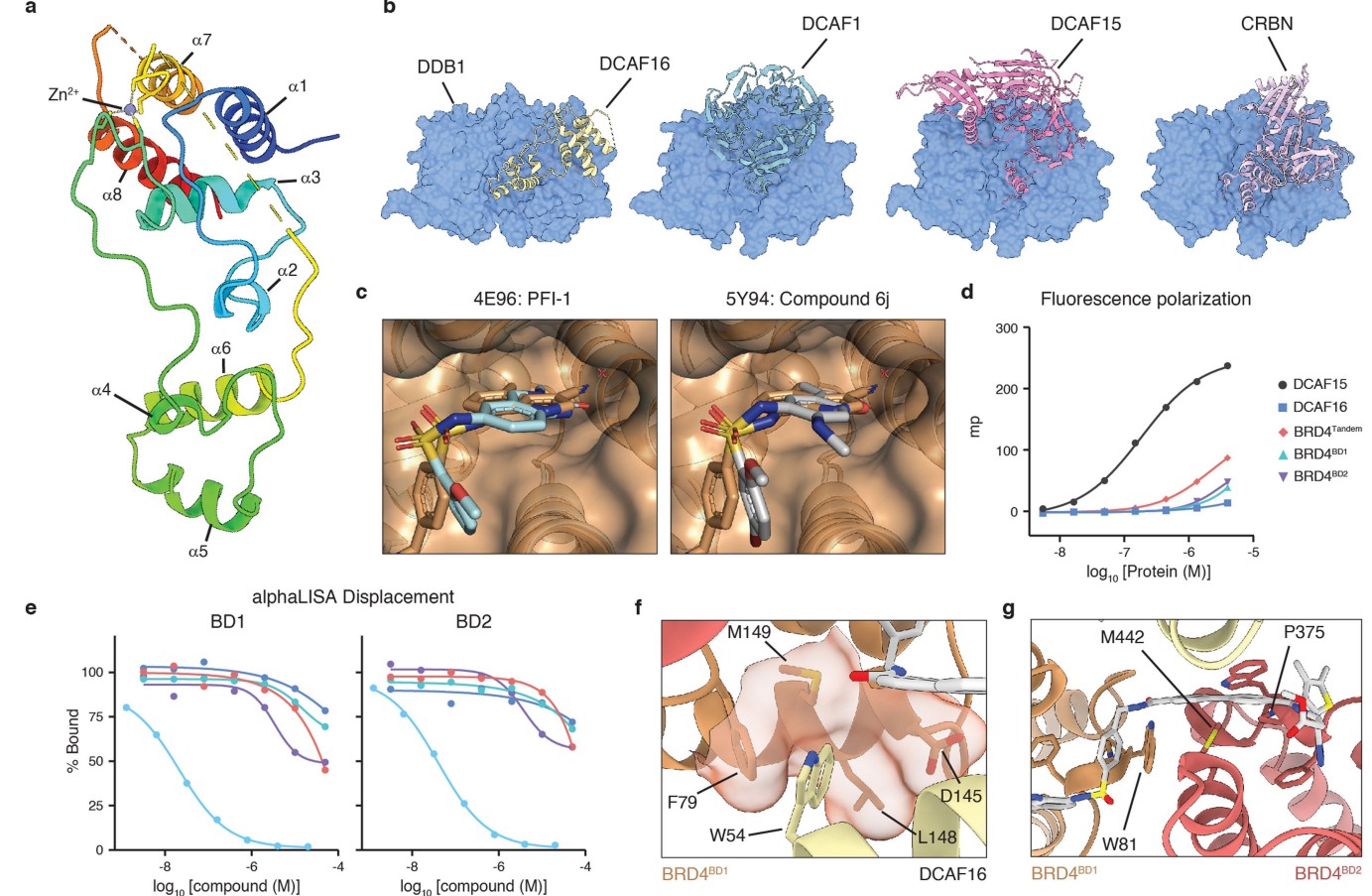

**Extended Data Fig. 5 | Structure-based characterization of the ternary BRD4-IBG1-DCAF16 complex. a**, Structure of DCAF16 coloured rainbow from N- to C-terminus. **b**, Comparison of DCAF16 and structurally distinct CRL4 substrate receptors DCAF1, DCAF15, and CRBN (PDB entries 5JK7, 6UD7, and 5FQD, respectively) bound to DDB1 (blue). **c**, Comparison of binding mode in the acetyl-lysine pocket of BRD4[BD1] (orange surface and cartoon) between IBG1 (orange) and known sulfonamide BET inhibitors PFI-1 (left, light blue; PDB 4E96) and compound 6j (right, grey; PDB 5Y94). The cyano group of IBG1 overlays close to a conserved water molecule found in both crystal structures and other published BD1 structures. **d**, Fluorescence polarization binary binding assay. Proteins were titrated into 20 nM FITC-sulfonamide probe, as in Fig. 2g. n = 3 technical replicates, mean +/− s.d. **e**, alphaLISA displacement assay. Competition of a biotinylated-JQ1 probe following titration of compounds **1a**, **1d**, E7820, Indisulam, or JQ1 into His-BRD4[BD1] (left) or His-BRD4[BD2] (right). Data, mean of n = 2 technical replicates. **f**, Detailed view of DCAF16-BD1 interface. Residue W54 of DCAF16 binds to a hydrophobic pocket on the surface of BD1. **g**, Detail view of BD1-BD2 interface. Residue M442 of BD2 is sandwiched between residues W81 and P375 of the BD1 and BD2 WPF shelves, respectively, as well as the linker of IBG1.

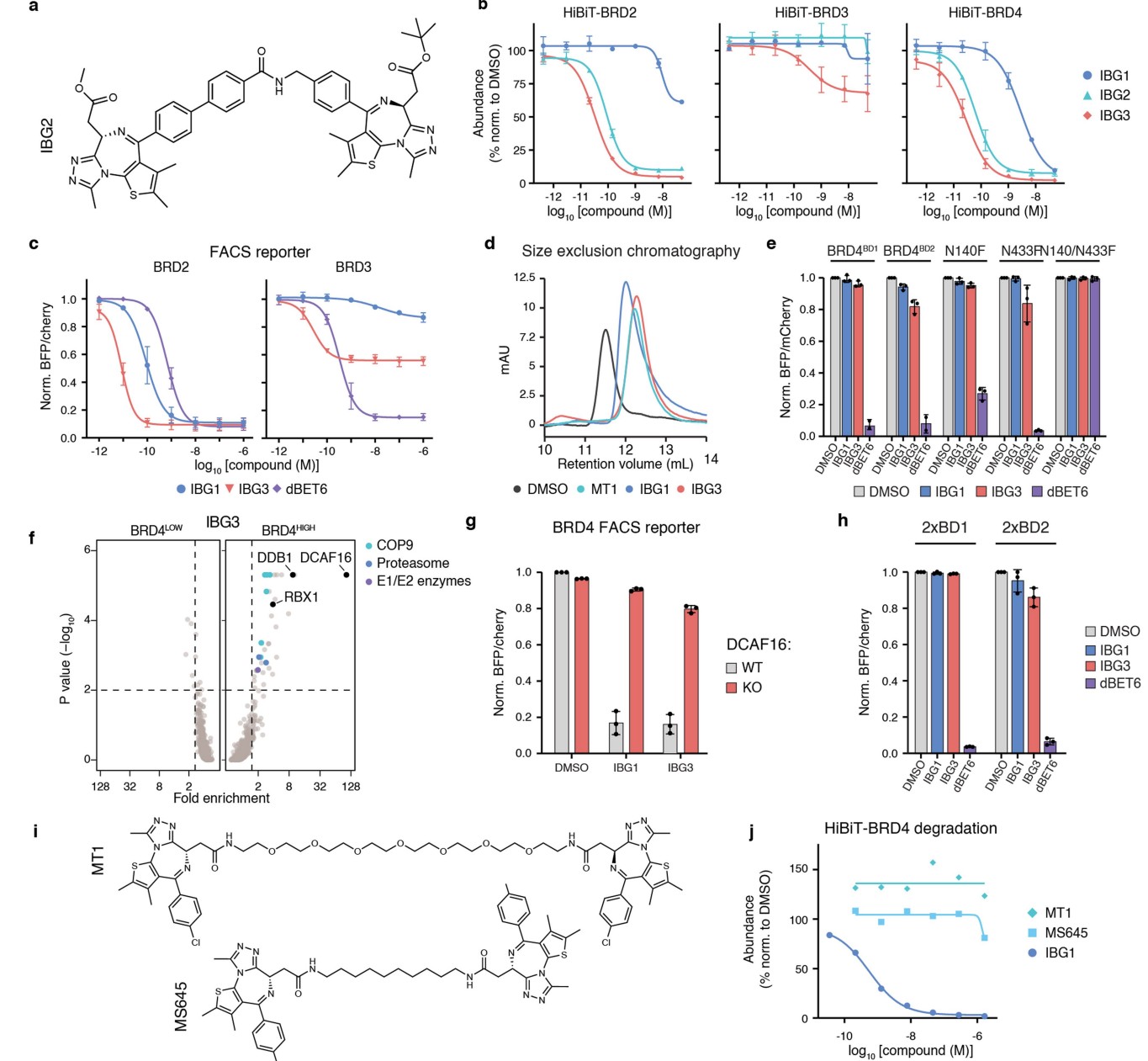

**Extended Data Fig. 6 | Rational design of improved intramolecular bivalent glue BET degraders. a**, Structure of double JQ1 containing intramolecular bivalent glue degrader IBG2. **b**, HiBiT degradation assay. HEK293 HiBiT knock-in cells were treated with IBG1, IBG2 or IBG3 for 5 h and levels of BRD2-, BRD3- and BRD4-HiBiT proteins were quantified via HiBiT lytic detection system. Data, n = 3 independent experiments, mean +/− s.d. **c**, BET protein degradation specificity. KBM7 cells expressing BRD2^Tandem or BRD3^Tandem dual fluorescence reporters were treated with increasing concentrations of IBG1, IBG3 or dBET6 for 6 h and BET protein levels were quantified via flow cytometry. **d**, Size exclusion chromatograms of BRD4^Tandem incubated with DMSO, MT1, IBG1 or IBG3. Data for DMSO and IBG1 as in Fig. 4c, data representative of n = 2 independent experiments. **e**, Bromodomain tandem selectivity. KBM7 cells expressing isolated BRD4 bromodomains or mutated BRD4^Tandem constructs were treated with IBG1 (1 nM), IBG3 (0.1 nM) or dBET6 (10 nM) for 6 h and protein levels were evaluated via flow cytometry. **f**, BRD4 stability CRISPR screen. KBM7 iCas9 BRD4 dual fluorescence reporter cells expressing a CRL-focused sgRNA

library were treated with IBG3 (0.1 nM) for 6 h before flow cytometric cell sorting as in Fig. 2b. 20 S proteasome subunits (blue), COP9 signalosome subunits (cyan) and E1 or E2 ubiquitin enzymes (purple) inside the scoring window (one-sided MAGeCK p-value < 0.01, fold-change > 1.5; dashed lines) are highlighted. **g**, DCAF16 dependency. BRD4(S) dual fluorescence reporter KBM7 iCas9 cells were lentivirally transduced with a *DCAF16*-targeting sgRNA and 3 days post Cas9 induction cells were treated with DMSO, IBG1 (1 nM) or IBG3 (0.1 nM) for 6 h before FACS-based quantification of BRD4 levels. **h**, Bromodomain arrangement. KBM7 cells expressing dual fluorescence reporters harbouring tandems of either BD1 or BD2 of BRD4 were treated with DMSO, IBG1 (1 nM), IBG3 (0.1 nM) or dBET6 (10 nM) for 6 h and analysed by flow cytometry. **i,j**, Structures (**i**) and HiBiT-BRD4 degradation activity (**j**) of bivalent BET inhibitors MT1 and MS645 after treatment for 24 h. Data for **c,e,g,h**, n = 3 independent experiments, mean +/− s.d. Data in **j**, mean of n = 2 independent experiments.

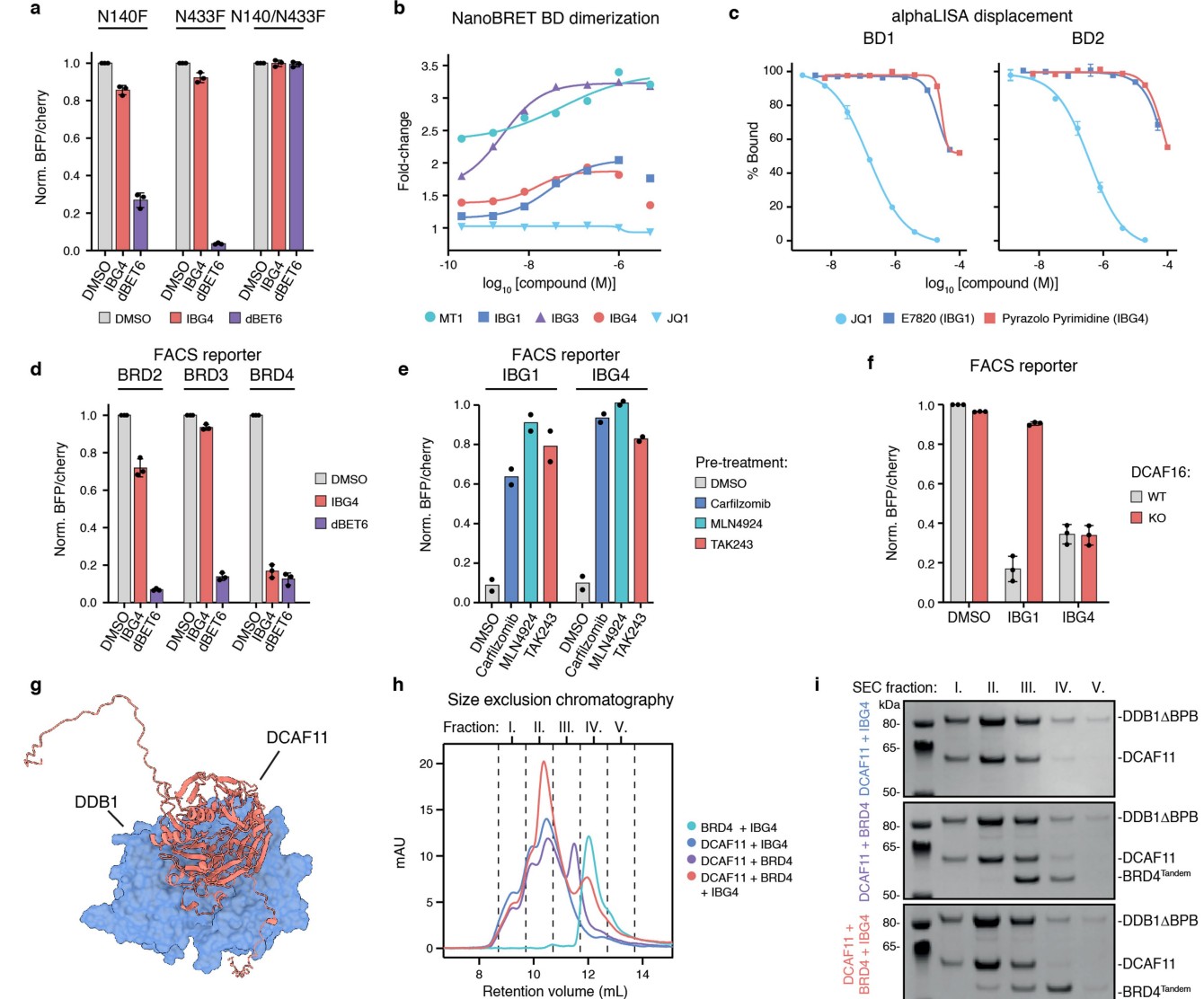

**Extended Data Fig. 7 | IBG4 is a DCAF11-dependent intramolecular bivalent glue degrader. a**, Bromodomain tandem specificity. KBM7 cells expressing bromodomain mutant BRD4[Tandem] dual fluorescence reporters were treated with DMSO, IBG4 (100 nM) or dBET6 (10 nM) for 6 h and analysed by flow cytometry. **b**, NanoBRET bromodomain dimerization assay. Indicated compounds were titrated into HEK293 cells transiently expressing BRD4[Nluc-Tandem-HaloTag]. Data, mean of n = 2 independent experiments. **c**, alphaLISA displacement assay. Increasing concentrations of JQ1, E7820 or the pyrazolo pyrimidine warhead of IBG4 were titrated against His-tagged BRD4 bromodomains and biotinylated JQ1 probe. n = 3 technical replicates, mean +/− s.d. **d**, BET protein selectivity. Bromodomain tandem BRD2, BRD3 or BRD4 dual fluorescence reporter KBM7 cells were treated with DMSO, IBG4 (100 nM) or dBET6 (10 nM) for 6 h and analysed by flow cytometry. **e**, Mechanistic FACS reporter assay. KBM7 BRD4

dual fluorescence reporter cells were co-treated with IBG1 (1 nM) or IBG4 (100 nM) and Carfilzomib (1 μM), MLN4924 (1 μM) or TAK243 (0.5 μM) for 6 h and BRD4 levels were analysed via flow cytometry. Data, mean of n = 2 independent experiments. **f**, DCAF16-independence of IBG4. KBM7 iCas9 WT or DCAF16 knockout cells expressing BRD4(S) dual fluorescence reporter were treated with DMSO, IBG1 (1 nM) or IBG4 (100 nM) for 6 h and BRD4 degradation was assessed via flow cytometry. **g**, AlphaFold (AlphaFold Monomer v2.0 pipeline)[78] prediction of DCAF11 (red) bound to DDB1 (blue). **h,i**, Size exclusion chromatograms of different combinations of DCAF11, BRD4[Tandem] and IBG4 (**h**), data representative of n = 2 independent experiments, and corresponding peak fractions run on SDS-PAGE (**i**). Data for **a,d,f**, n = 3 independent experiments, mean +/− s.d.

**Extended Data Table 1 | Plasmids and sgRNAs used in this study**

| Plasmid name | Application |
|---|---|
| pRRL-SFFV-BRD4(S)-mTagBFP-P2A-mCherry | Protein stability reporter, screen |
| pRRL-U6-sgRNA-EF1αs-Thy1.1-P2A-NeoR | Screen validation |
| pRRL-SFFV-FLAG-DCAF16-EF1αs-iRFP670 | KO/Rescue |
| pRRL-SFFV-BRD4(Tandem)-mTagBFP-P2A-mCherry | Protein stability reporter |
| pRRL-SFFV-BRD4(BD1)-mTagBFP-P2A-mCherry | Protein stability reporter |
| pRRL-SFFV-BRD4(NLS-BD1)-mTagBFP-P2A-mCherry | Protein stability reporter |
| pRRL-SFFV-BRD4(BD2)-mTagBFP-P2A-mCherry | Protein stability reporter |
| pRRL-SFFV-BRD4(NLS-BD2)-mTagBFP-P2A-mCherry | Protein stability reporter |
| pRRL-SFFV-BRD4(Tandem N140F)-mTagBFP-P2A-mCherry | Protein stability reporter |
| pRRL-SFFV-BRD4(Tandem N433F)-mTagBFP-P2A-mCherry | Protein stability reporter |
| pRRL-SFFV-BRD4(Tandem N140F/N433F)-mTagBFP-P2A-mCherry | Protein stability reporter |
| pRRL-SFFV-BRD2(Tandem)-mTagBFP-P2A-mCherry | Protein stability reporter |
| pRRL-SFFV-BRD3(Tandem)-mTagBFP-P2A-mCherry | Protein stability reporter |
| pRRL-SFFV-BRD2-BD1/BRD4-BD2(Tandem)-mTagBFP-P2A-mCherry | Protein stability reporter |
| pRRL-SFFV-BRD3-BD1/BRD4-BD2(Tandem)-mTagBFP-P2A-mCherry | Protein stability reporter |
| pRRL-SFFV-BRD4-BD1/BRD2-BD2(Tandem)-mTagBFP-P2A-mCherry | Protein stability reporter |
| pRRL-SFFV-BRD4-BD1/BRD3-BD2(Tandem)-mTagBFP-P2A-mCherry | Protein stability reporter |
| pRRL-SFFV-BRD4(Tandem G386E)-mTagBFP-P2A-mCherry | Protein stability reporter |
| pRRL-SFFV-BRD4(ΔSEED)-mTagBFP-P2A-mCherry | Protein stability reporter |
| pRRL-SFFV-BRD4(ΔET)-mTagBFP-P2A-mCherry | Protein stability reporter |
| pRRL-SFFV-BRD4(ΔBID)-mTagBFP-P2A-mCherry | Protein stability reporter |
| pRRL-SFFV-BRD4(ΔNPS)-mTagBFP-P2A-mCherry | Protein stability reporter |
| pRRL-SFFV-BRD4(Tandem BRD2-linker)-mTagBFP-P2A-mCherry | Protein stability reporter |
| pRRL-SFFV-BRD4(Tandem BRD3-linker)-mTagBFP-P2A-mCherry | Protein stability reporter |
| pRRL-SFFV-BRD4(TandemΔMotifA)-mTagBFP-P2A-mCherry | Protein stability reporter |

| sgRNA | Sequence (5' to 3') | Application |
|---|---|---|
| AAVS1 | GCTGTGCCCCGATGCACAC | Screen validation |
| DCAF16_1 | GGACTCCACAAGAGGCCAGA | Screen validation |
| DCAF16_2 | GTTCCAGTTTGGGGACACAA | Screen validation |
| DDB1 | GATGCCTGGTAAGTCAATGC | Screen validation |
| DCAF11 | GAGAGTTGGAGATCAGATACC | Screen validation |
| DCAF15_1 | TTGAGGGACACGCACACCCG | Mechanistic studies |
| DCAF15_2 | ACTCGCATACGGTCAGGTAC | Mechanistic studies |
| BRD4_1 | TGGGATCACTAGCATGTCTG | HiBiT-Tagging |
| BRD4_2 | GTGGGATCACTAGCATGTCTG | BromoTag |
| BRD4_3 | GACTAGCATGTCTGCGGAGAG | BromoTag |
| BRD3 | TCGTGGCGGTGGACATCCTC | HiBiT-Tagging |
| BRD2 | TTTGCAGCATCTTGACCGCA | HiBiT-Tagging |
| MCM4_1 | GTCCGAGCACTATGTCGTCCC | BromoTag |
| MCM4_2 | GTCCGAGCACTATGTCGTCCC | BromoTag |

**Extended Data Table 2 | Cryo-EM data collection, refinement and validation statistics**

|  | EMDB-17172<br>PDB 8OV6 |
|---|---|
| **Data collection and processing** | |
| Magnification | 190,000 |
| Voltage (kV) | 200 |
| Electron exposure (e–/Å$^2$) | 12.7 |
| Defocus range (μm) | -(1.7-3.2) |
| Pixel size (Å) | 0.74 |
| Symmetry imposed | C1 |
| Initial particle images (no.) | 564,575 |
| Final particle images (no.) | 192,014 |
| Map resolution (Å) | 3.77 |
| FSC threshold | 0.143 |
| Map resolution range (Å) | 2.5-4.5 |
|  | |
| **Refinement** | |
| Initial model used (PDB code) | 3MXF, 5FQD, 6DUV |
| Map sharpening $B$ factor (Å$^2$) | 220.7 |
| Model composition | |
| Non-hydrogen atoms | 9,289 |
| Protein residues | 1,161 |
| Ligands | U79, ZN |
| Model-Map | |
| CC (mask) | 0.79 |
| R.m.s. deviations | |
| Bond lengths (Å) | 0.002 |
| Bond angles (°) | 0.505 |
| Validation | |
| MolProbity score | 1.16 |
| Clashscore | 1.62 |
| CaBLAM outliers (%) | 1.16 |
| Rotamer outliers (%) | 0.10 |
| Cβ outliers (%) | 0.00 |
| Ramachandran plot | |
| Favored (%) | 96.31 |
| Allowed (%) | 3.69 |
| Disallowed (%) | 0 |

# Reporting Summary

## Statistics

For all statistical analyses, confirm that the following items are present in the figure legend, table legend, main text, or Methods section.

| n/a | Confirmed | |
|---|---|---|
| ☐ | ☒ | The exact sample size (*n*) for each experimental group/condition, given as a discrete number and unit of measurement |
| ☐ | ☒ | A statement on whether measurements were taken from distinct samples or whether the same sample was measured repeatedly |
| ☐ | ☒ | The statistical test(s) used AND whether they are one- or two-sided *Only common tests should be described solely by name; describe more complex techniques in the Methods section.* |
| ☒ | ☐ | A description of all covariates tested |
| ☐ | ☒ | A description of any assumptions or corrections, such as tests of normality and adjustment for multiple comparisons |
| ☐ | ☒ | A full description of the statistical parameters including central tendency (e.g. means) or other basic estimates (e.g. regression coefficient) AND variation (e.g. standard deviation) or associated estimates of uncertainty (e.g. confidence intervals) |
| ☐ | ☒ | For null hypothesis testing, the test statistic (e.g. *F*, *t*, *r*) with confidence intervals, effect sizes, degrees of freedom and *P* value noted *Give P values as exact values whenever suitable.* |
| ☒ | ☐ | For Bayesian analysis, information on the choice of priors and Markov chain Monte Carlo settings |
| ☒ | ☐ | For hierarchical and complex designs, identification of the appropriate level for tests and full reporting of outcomes |
| ☒ | ☐ | Estimates of effect sizes (e.g. Cohen's *d*, Pearson's *r*), indicating how they were calculated |

*Our web collection on statistics for biologists contains articles on many of the points above.*

## Software and code

Policy information about availability of computer code

| | |
|---|---|
| Data collection | Flow cytometry: Data was collected on a BD LSRFortessa using BD FACSDiva software (v9.0), BD FACSAria Fusion using BD FACSDiva software (v8.0.2) <br> Western blotting/SDS gels: ChemiDoc Touch imaging system (BioRad) operated on Image Lab (v2.4.0.03). <br> NGS: Illumina HiSeq3500 (https://www.illumina.com/) <br> Cell viability assays: PerkinElmer VICTOR X3 operated on PerkinElmer 2030 software (v4.0) or BMG Labtech PHERAstar (firmware v1.33). <br> HiBiT and NanoBRET endpoint assays: BMG Labtech PHERAstar (firmware v1.33) <br> Promega kinetic assays: GloMax® Discover System (software v4.0.0, firmware v4.92) <br> Mass spectrometry: Orbitrap Fusion Lumos Tribrid mass spectrometer coupled to a Dionex Ultimate 3000 RSLCnano system and operated via Xcalibur (v4.3.73.11) and Tune (v3.4.3072.18). <br> Cryo-EM: Glacios Transmission Electron Microscope (Thermo Fisher) with Falcon4i direct electron detector, operated on EPU (v3.0) software. |
| Data analysis | Flow Cytometry Analysis: Flowjo (v10.8.1) <br> FACS-based CRISPR screens: All relevant software is described in detail in the corresponding methods section. Pipelines for sgRNA quantification and statistical analysis are available on Github (https://github.com/ZuberLab/crispr-processnf/tree/566f6d46bbcc2a3f49f51bbc96b9820f408ec4a3 and https://github.com/ZuberLab/crisprmageck-nf/tree/c75a90f670698bfa78bfd8be786d6e5d6d4fc455). Used packages: fastx-toolkit (v0.0.14), Bowtie2 (v2.4.5), featureCounts (v2.0.1), MAGeCK (v0.5.9). <br> Viability-based CRISPR screen: All relevant software is described in detail in the corresponding methods section. Used packages: bcl2fastq (v2.20.0.422), cutadapt (v2.8), Bowtie2 (v2.3.0), MAGeCK (v0.5.9). SgRNAs for ubiquitin/Nedd8 focused library were selected with the Broad Institute CRISPick tool. <br> Western blot quantification: Image Lab (v6.1 build 7) |

For manuscripts utilizing custom algorithms or software that are central to the research but not yet described in published literature, software must be made available to editors and reviewers. We strongly encourage code deposition in a community repository (e.g. GitHub). See our web collection on software and code information for manuscripts.

Data compiling, processing and statistical analyses: Microsoft Excel for Microsoft 365 (v2208.16.0.15601.20526), R Studio (v2022.12.0 Build 353) with R (v4.2.2), GraphPad Prism (v9.3.1, v9.5.1)
Mass Spectrometry: Proteome Discoverer (v2.4.1.15)
Cryo-EM: Cryosparc (v4.1.2), eLBOW (in Phenix v1.20.1.4487), GRADE web server (Grade2 v1.3.0), ColabFold (v1.3), ModelAngelo (v0.2.2), WinCoot (v0.9.8.1), ISOLDE (v1.6), Phenix (v1.20.1-4487), ChimeraX (v1.6), The PyMOL Molecular Graphics System (v2.5.2)
AlphaFold: model of DCAF11 pulled from https://alphafold.ebi.ac.uk/entry/Q8TEB1 created with AlphaFold Monomer v2.0 pipeline.

Code for analysis of FACS-based screens is available on GitHub (https://github.com/ZuberLab/crispr-process-nf, https://github.com/ZuberLab/crispr-mageck-nf).

For manuscripts utilizing custom algorithms or software that are central to the research but not yet described in published literature, software must be made available to editors and reviewers. We strongly encourage code deposition in a community repository (e.g. GitHub). See the Nature Portfolio guidelines for submitting code & software for further information.

## Data

Policy information about availability of data

All manuscripts must include a data availability statement. This statement should provide the following information, where applicable:
- Accession codes, unique identifiers, or web links for publicly available datasets
- A description of any restrictions on data availability
- For clinical datasets or third party data, please ensure that the statement adheres to our policy

Source data for Fig. 1c, Fig. 2b, c, Fig. 5d, Extended Data Fig. 2a and Extended Data Fig. 6f are included in the Supplementary information files of the manuscript (Supplementary Data 1-5). Cryo-EM density maps are deposited in the EMDB with the accession code EMD-17172. The atomic model is deposited under Protein Data Bank ID 8OV6. Quantitative proteomics data have been deposited to the ProteomeXchange Consortium PRIDE repository with the accession ID PXD040570. Crystallographic or electron microscopy structures of substrate receptors bound to DDB1 and of BET inhibitors bound to BRD4 bromodomains shown for comparison in Extended Data Fig. 5b, c were obtained from the RCSB protein database (https://www.rcsb.org/) via accessions 5JK7, 5FQD, 6UD7, 7ZN7, 4E96, 5Y94, 3MXF and 6DUV. Full version of all gels and blots are provided in Supplementary Fig. 1, schematics of gating strategies applied for FACS analyses and cell sorting are provided in Supplementary Fig. 2.
All biological materials are available upon reasonable requests under material transfer agreements (MTA) with The Centre for Targeted Protein Degradation, University of Dundee, or CeMM Research Center for Molecular Medicine of the Austrian Academy of Sciences, respectively.

## Human research participants

Policy information about studies involving human research participants and Sex and Gender in Research.

| | |
|---|---|
| Reporting on sex and gender | N/A |
| Population characteristics | N/A |
| Recruitment | N/A |
| Ethics oversight | N/A |

Note that full information on the approval of the study protocol must also be provided in the manuscript.

# Field-specific reporting

Please select the one below that is the best fit for your research. If you are not sure, read the appropriate sections before making your selection.

☒ Life sciences  ☐ Behavioural & social sciences  ☐ Ecological, evolutionary & environmental sciences

For a reference copy of the document with all sections, see nature.com/documents/nr-reporting-summary-flat.pdf

# Life sciences study design

All studies must disclose on these points even when the disclosure is negative.

| | |
|---|---|
| Sample size | Sample sizes were not predetermined using statistical analyses. Sample sizes were based on prior experience in the field and our previous studies (Zengerle et al. ACS Chem Biol 2015; Gadd et al. Nat. Chem. Biol. 2017; Riching et al. ACS Chem Biol. 2018). |
| Data exclusions | In quantitative proteomics, proteins with less than three unique peptides detected were excluded from downstream analysis. For cryo-EM and data filtering is outlined in Extended Data Fig. 4 and for flow cytometry, gating schematics are shown in Supplementary Fig. 2. |
| Replication | Unless stated in figure legends or method sections, all experiments were done at least twice and the reproduction were successful. The number of technical or biological replicates and independent biological experiments are specified in the respective figure legends. |
| Randomization | No randomization was performed, as is standard for genetic, biochemical and structural studies. Internal controls were used for quantitative |

| Randomization | comparisons. |
| Blinding | No blinding was performed, as no subjective measurements were done. |

# Reporting for specific materials, systems and methods

We require information from authors about some types of materials, experimental systems and methods used in many studies. Here, indicate whether each material, system or method listed is relevant to your study. If you are not sure if a list item applies to your research, read the appropriate section before selecting a response.

## Materials & experimental systems

| n/a | Involved in the study |
|---|---|
| ☐ | ☒ Antibodies |
| ☐ | ☒ Eukaryotic cell lines |
| ☒ | ☐ Palaeontology and archaeology |
| ☒ | ☐ Animals and other organisms |
| ☒ | ☐ Clinical data |
| ☒ | ☐ Dual use research of concern |

## Methods

| n/a | Involved in the study |
|---|---|
| ☒ | ☐ ChIP-seq |
| ☐ | ☒ Flow cytometry |
| ☒ | ☐ MRI-based neuroimaging |

## Antibodies

| Antibodies used | The following primary antibodies were used for immunoblotting: BRD2 (1:1000, no. Ab139690, Abcam), BRD3 (1:2000, no. Ab50818, Abcam), BRD4 (1:1000, E2A7X, no. 13440, Cell Signaling Technology and no. Ab128874, Abcam), BromoTag (1:1000, no. NBP3-17999, Novus Biologicals), CUL4A (1:2000, no. A300-738A, Bethyl Laboratories), CUL4B (1:2000, no. 12916-1-AP, Proteintech), DDB1 (1:1000, no. A300-462A, Bethyl Laboratories), MCM4 (1:1000, no. ab4459, Abcam) RBM39 (1:1000, no. HPA001591, Atlas Antibodies), RBX1 (1:1000, D3J5I, no. 11922, Cell Signalling Technology), DCAF11 (1:2000, no. A15519, ABclonal), cleaved Caspase-3 (1:1000, D3E9, no. 9579, Cell Signalling Technology), PARP1 (1:1000, no. 9542, Cell Signalling Technology), MYC (1:500, D84C12, no. 5605, Cell Signalling Technology), β-Actin (1:10000, AC-15, no. A5441, Sigma-Aldrich), α-Tubulin (1:500, DM1A, no. T9026, Sigma-Aldrich).<br><br>The following secondary antibodies were used for immunoblotting: HRP anti-rabbit IgG (1:2500, 7074, Cell Signaling Technology), HRP anti-mouse IgG (1:5000, 7076, Cell Signaling Technology), IRDye® 680RD anti-mouse (1:5000, no. 926-68070, Li-Cor), IRDye® 800CW anti-rabbit (1:5000, no. 926-32211, Li-Cor), StarBright™ blue 520 goat anti-mouse (1:5000, no. 12005866, Biorad) and hFABTM rhodamine anti-tubulin (1:5000, no. 12004165, Biorad).<br><br>The following antibodies were used for FACS analysis and cell sorting: APC anti-mouse CD90.1/Thy-1.1 antibody (1:400, no. 202526, BioLegend), Human TruStain FcX™ Fc Receptor Blocking Solution (1:400, no. 422302, BioLegend). |
| Validation | Target specificity for the following antibodies were previously confirmed by the Ciulli group:<br>- anti-BRD2 (abcam no. ab139690): Knockdown validated, disappearance of the band in immunoblotting upon MZ1 / siRNA treatment (Zengerle et al. ACS Chem. Biol. 2015).<br>- anti-BRD3 (abcam no. ab50818): Knockdown validated, disappearance of the band in immunoblotting upon MZ1 / siRNA treatment (Zengerle et al. ACS Chem. Biol. 2015).<br>- anti-BRD4 (abcam no. ab128874): Knockdown validated, disappearance of the bands in immunoblotting upon MZ1 / siRNA treatment (Zengerle et al. ACS Chem. Biol. 2015)<br><br>Target specificity for the following antibodies were confirmed with established degrader compounds:<br>BromoTag (no. NBP3-17999, Novus Biologicals; AGB1 - Bond et al. JMedChem 2021), RBM39 (no. HPA001591, Atlas Antibodies; E7820; Uehara et al. NCB, 2017), BRD4 (2A7X, no. 13440, Cell Signaling Technology; dBET6 - Winter et al. Mol Cell, 2017), MCM4 (no. ab4459, Abcam; BromoTag degrader ABG1).<br><br>Target specificity for CUL4A (no. A300-738A, Bethyl Laboratories), CUL4B (no. 12916-1-AP, Proteintech), DDB1 (no. A300-462A, Bethyl Laboratories) and RBX1 antibodies (D3J5I, no. #11922, Cell Signalling Technology) were confirmed by western blotting following corresponding siRNA treatment (48h).<br><br>Target specificity for MYC antibody (D84C12, no. 5605, Cell Signalling Technology) was confirmed in previous work by CRISPR-based knockout (De Almeida et al. Nature, 2021).<br><br>Target specificity for DCAF11 antibody (no. A15519, ABclonal) was confirmed by western blot following inducible knockout in KBM7 iCas9 cells (Fig. 5e).<br><br>Target specificity for Cleaved Caspase 3 (D3E9, no. 9579, Cell Signalling Technology) and PARP1 antibodies was validated by the vendor via control compounds (Etoposide, Stautosporine) and blocking peptides.<br><br>Target specificity for APC anti-mouse CD90.1/Thy-1.1 antibody (no. 202526, BioLegend) was verified by ectopic overexpression. |

# Eukaryotic cell lines

Policy information about cell lines and Sex and Gender in Research

| | |
|---|---|
| Cell line source(s) | Human HEK293, HCT-116, MV-4-11 and HeLa cells were obtained from ATCC.<br>KBM7 (originally obtained from Haplogen Bioscience) iCas9 cells were a gift from Johannes Zuber (IMP - Research Institute of Molecular Pathology).<br>Lenti-X 293T cells were purchased from Clontech. |
| Authentication | All used cell lines were authenticated by short tandem repeat (STR) profiling. Successfull CRISPR-based editing of cell lines was confirmed by cell-based degradation assays (RBM39 treatment for the DCAF15 KO cell line or dBET6 treatment for HiBiT-BET cell lines) as well as by Sanger Sequencing. |
| Mycoplasma contamination | All used cell lines were routinely tested and confirmed negative for mycoplasma contamination. |
| Commonly misidentified lines<br>(See ICLAC register) | No commonly misidentified cell lines were used. |

# Flow Cytometry

## Plots

Confirm that:

☒ The axis labels state the marker and fluorochrome used (e.g. CD4-FITC).

☒ The axis scales are clearly visible. Include numbers along axes only for bottom left plot of group (a 'group' is an analysis of identical markers).

☒ All plots are contour plots with outliers or pseudocolor plots.

☒ A numerical value for number of cells or percentage (with statistics) is provided.

## Methodology

| | |
|---|---|
| Sample preparation | Sample preparation for each experiment is described in detail in the methods section.<br><br>To engineer a BRD4 protein stability reporer, KBM7 iCas9 cells were transduced with lentivirus expressing SFFV-BRD4(S)-mTagBFP-P2A-mCherry to generate stable reporter cell lines. BRD4/mCherry double positive cells were sorted on a Cytoflex SRT (Beckman Coulter). For evaluation of BRD4-TagBFP reporter degradation, cells were treated with DMSO or BRD4 degraders for 6 hours before flow cytometry analysis on an LSRFortessa (BD Biosciences).<br><br>For BRD4 stability CRISPR screens library transduced cells were induced with doxycycline for 3 days and harvested after 6 hours of treatment with DMSO or degraders. Cells were washed with PBS, stained with Zombie NIR™ Fixable Viability Dye (1:1000, BioLegend) and APC anti-mouse CD90.1/Thy-1.1 antibody (1:400, BioLegend) in the presence of Human TruStain FcX™ Fc Receptor Blocking Solution (1:400, BioLegend), and fixed with 0.5 mL methanol-free paraformaldehyde 4% (Thermo Scientific™ Pierce™) for 30 min at 4 ºC, while protected from light. Cells were washed with, and stored in FACS buffer (PBS containing 5% FBS and 1 mM EDTA) at 4 ºC over night. The next day, cells were strained trough a 35 μm nylon mesh and sorted on a BD FACSAria™ Fusion (BD Biosciences) using a 100 μm nozzle. Aggregates, dead (ZombieNIR positive), Cas9-negative (GFP) and sgRNA library-negative (Thy1.1-APC) cells were excluded, and the remaining cells were sorted based on their BRD4-BFP and mCherry levels into BRD4HIGH (8-10% of cells), BRD4MID (25-30%) and BRD4LOW (8-10%) fractions. For each sample, cells corresponding to at least 1,500-fold library representation were sorted per replicate.<br><br>To validate screen hits, BRD4 reporter cells were lentivirally transduced with an sgRNA expression plasmid (pLenti-U6-sgRNA-IT-EF1αs-Thy1.1-P2A-NeoR) at 30-50% transduction efficiency. Cas9 expression was induced with docycycline (0.4 μg ml−1) for 3days, followed by 6 hours of degrader treatment. Cells were stained for sgRNA expression with an APC conjugated anti-mouse -CD90.1/Thy1.1 antibody (#202526, Biolegend; 1:400) and Human TruStain FcX Fc receptor blocking solution (#422302, Biolegend; 1:400) for 5 minutes in FACS buffer (PBS containing 5% FBS and 1 mM EDTA) at 4°C. Cells were washed and resuspended in FACS buffer and analyzed on an LSRFortessa (BD Biosciences.<br><br>For KO/rescue studies, sgDCAF16 expressing BRD4-reporter KBM7 iCas9 cells were transduced with a lentivirus expressing an sgRNA resistant DCAF16 cDNA (pRRL-SFFV-3xFLAG-DCAF16-EF1αs-iRFP670) and analyzed as above.<br>For BRD4 degradation studies, KBM7 iCas9 cells were lentivirally transduced with mutant or truncated versions of the BRD4-TagBFP protein stability reporter and treated with DMSO or degraders for 6 hours. Reporter negative cells were excluded based on mCherry signal and reporter positive cells were analyzed.<br>Flow cytometric data analysis was performed in FlowJo v10.8.1. BFP and mCherry mean fluorescence intensity (MFI) values for were normalized by background subtraction of the respective values from reporter-negative KBM7 wild type cells. BRD4 abundance was calculated as the ratio of background subtracted BFP to mCherry MFI, and is displayed normalized to DMSO treated, sgRNA/cDNA double negative cells. |
| Instrument | All flow cytometric analyses were performed on BD LSRFortessa (4 laser, 16 detector configuration; BD Bioscience). All sorts were performed on BD FACSAria Fusion (5 lasers, 16 detectors; BD Bioscience) or CytoFLEX SRT (4 lasers, 15 detectors; Beckman Coulter) cell sorters. |
| Software | BD FACSDiva software (v8.0.2 and v9.0), Beckman Coulter CytExpert SRT (v 1.1.0.10007), Flowjo (v10.8.1) |

Cell population abundance

In BRD4 stability CRISPR screens, cells were sorted into BRD4HIGH (5-10% of cells), BRD4LOW (5-10%), and BRD4MID (25-30%) populations. All collected fractions were reanalyzed for purity and fractions with > 5% cross-contamination were discarded before further processing.

Gating strategy

In all analyses, forward scatter area vs. side scatter area plot was used to separate cell events from debris and dead cells. Forward scatter height vs. forward scatter area and/or side scatter width vs. side scatter height plots were used to separate single cells from aggregates.
For the sorting of fixed cells in CRISPR BRD4 protein stability screens, dead cells were excluded based on Zombia-NIR staining (BV786-A) vs FSC-A and sgRNA library (Thy1.1-APC-A), iCas9 (FITC-A) and reporter (PE-TexasRed-A) triple positive cells were sorted into BRD4LOW, BRD4HIGH, and BRD4MID populations based on BRD4-BFP (BV421-A) vs mCherry (PE-TexasRed-A) scatter plots. These gates were dynamically adjusted to keep the percentage at 5-10% for BRD4HIGH and BRD4LOW and 25-30% for BRD4MID populations.
A figure exemplifying the gating strategy for all FACS experiments and FACS-based screens in provided in Supplementary Figure 2.

☒ Tick this box to confirm that a figure exemplifying the gating strategy is provided in the Supplementary Information.

