## [Peer Review File · Nature]

Manuscript Title: Targeted protein degradation via intramolecular bivalent glues

Reviewer Comments & Author Rebuttals

Reviewer Reports on the Initial Version:

Referees' comments:

Referee #1 (Remarks to the Author):

In this manuscript, Hsia O. et al. report the mechanism of action of a BRD4 degrader, which is originally designed as a DCAF15-targeting PROTAC but unexpectedly acts as a DCAF16-targeting molecular glue. A unique feature of this PROTAC-like glue is that it seems to stabilize interdomain interaction within the BRD4 tandem bromodomains while leveraging an intrinsic affinity between DCAF16 and BRD4. The study is remarkably complete, ranging from the initial characterization of the compound of interest, deconvolution of the E3 ligase, biophysical analysis of the interacting partners, to final determination of the structural mechanism. The presentation is clear, logical, and concise.

Major points:

1. Figure 3d. The affinity difference between BRD4 tandem BDs and DCAF16 +/- compound 1 is surprisingly small (0.7 vs 1 μ M). If the activity of the compound is to be explained by its ability to enhance the E3-substrate affinity, the authors need to validate this subtle change using an orthogonal method (ITC, MST, AlphaLISA displacement etc.).
2. If the affinity difference is indeed subtle, the authors should be positioned to determine the structure of the BRD4-DCAF16 complex in the absence of compound 1. The authors argue that the activity of the compound might be attributable to its ability to stabilize the structural configuration of BRD4 BD1-BD2 in complex with DCAF16. It is unclear whether such a mechanism leads to an enhanced affinity between the two proteins or a preferred orientation of the two BRD4 BDs on DCAF16 for ubiquitination. Structural characterization of the compound-free protein complex structure might shed light on this.

Minor points:

1. The authors state that the resolution of the cryo-EM structure is about 4 angstrom. This might be true for DDB1. But the resolution is most likely much lower for DCAF16-BRD4. The maps shown in Fig 4 do not appear to have a resolution close to 4 angstrom. Please revise the statement and provide a figure showing the local resolution of the EM map.
2. The authors mention that the DCAF16 density in the EM map is not clear enough to build an atomic model. Is the density somewhat consistent with what AlphaFold predicts?
3. ED Figure 1e, Y axis is mislabeled?
4. Page 7, "In alphaLISA displacement assays, we found significantly enhanced affinity of 1 to BRD4Tandem in the presence of DCAF16 (IC₅₀ = 12.8 nM), as compared to 1 and BRD4Tandem alone (IC₅₀ = 462 nM; cooperativity (α) = 36; Fig. 3c, Extended Data Fig. 4b), further supporting a role of compound 1 in stabilizing the BRD4:DCAF16 complex." The last statement is not necessarily true if DCAF16 does not make direct contact with the compound. The most straightforward conclusion one can draw from the data is that DCAF16 stabilizes the BRD4-

compound 1 complex.

5. Page 10, last paragraph. The data showing the degradation resistance of the two BRD4 constructs harbouring either two copies of BD1 or BD2 (Extended Data Fig. 5e) should be mentioned somewhere else in the text instead of right after the discussion of G386 and N93. If N93 is solvent exposed, its substitution with Gly should not have any impact. The fact that BD1/BD1 and BD2/BD2 are resistant to compound-induced degradation is most likely due to the importance of the BD1-BD2 interface and their interfaces with DCAF16.

Referee #2 (Remarks to the Author):

This is a seminal paper by Ciulli and Winter's groups on the discovery of a sulfonamide-based BRD4 degrader that was previously thought to work through DCAF15 that this team discovered actually operates through DCAF16. They further demonstrate that the GNE-011 BRD4 molecular glue degrader also operates through DCAF16. The study also shows inherent affinity between BRD4 and DCAF16 which is further strengthened through the degraders. Interestingly, the authors show that the sulfonamide warhead itself does not bind to DCAF16 by itself and that the bromodomain alongside DCAF16 was required for the whole compound to bind to DCAF16. Structural, biochemical and genetic validation experiments all support their model.

Overall, this is an interesting study that not only demonstrates the mechanism of action of previously reported PROTAC/molecular glue degraders of BRD4 as operating through DCAF16, but also potentially reveals rational design strategies that one can take for future molecular glue degrader design beyond BRD4 through exploitation of existing weak protein-protein interactions and strengthening them through these types of molecular glue degraders. The studies described are rigorously performed and should be accepted for publication.

Referee #3 (Remarks to the Author):

Summary of the key results:

In this study, Drs. Alessio Ciulli, Georg E. Winter and colleagues reported their extensive investigation of a molecule glue BRD2/4 degrader (compound 1) recently disclosed in patent publications (WO2021157684). Although compound 1 was synthesized by linking JQ1, a ligand for BET proteins and E7820, a ligand for DCF15, its induced BRD2/4 protein degradation is independent of DCF15. Interestingly, GNE-0011, a previously reported BRD2/4 degrader, also induced degradation of BRD2/4 with a similar but yet different mechanism of action. Of note, GNE-0011 has a propargylamine tail extending from the para-position of the phenyl ring and lacks the ligand for DCF15. Through CRISPR/Cas9 screen and other assays, the authors identified and confirmed that the induced degradation of BRD2/BRD4 by compound 1 and GNE-0011 depends upon DCAF16 and associated degradation machinery, in contrast to dBET6, which is a PROTAC BET degrader and depends upon cereblon/cullin4A for degradation. While compound 1 binds to DCF15 in vitro, it lacks the apparent affinity to DCAF16. In ternary complex formation assays, the authors showed that while compound 1 and GNE-0011 have no obvious interactions with DCF16, BRD4 and DCF16 have intrinsic interactions, which are further enhanced by both compound 1 and GNE-0011, albeit with different degrees. The authors further showed that presence of two tandem bromodomains (BD1 and BD2) are necessary and sufficient for induced BRD2/4 degradation by compound 1 and GNE-0011. The authors then determined the ternary complex formed between BRD4Tandem, 1 and DCAF16:DDB1ΔBPB at ~4 Å resolution by . Although the resolution is not very high, the structure clearly reviews that compound 1 binds to BD2 through its JQ-1 moiety and to BD1 through its indole-sulfonamide moiety to link the two domains and stabilize their relative

conformation in a bidentate fashion. While compound 1 does not bind directly with DCF16, it facilitates a multivalent gluing interaction between the two bromodomains of BRD4 and DCAF16. Therefore, the authors have discovered a unique mode of action for a new type of molecule glues. Previously reported molecule glues first bind to an E3 ligase and then recruit neo-substrate proteins for degradation. In contrast, compound 1 binds concurrently to two different domains of BRD2/4 proteins, bringing these two domains together and stabilizing their conformations to recruit an E3 ligase (DCF16) for degradation. In addition, the authors have identified the residues responsible for the degradation selectivity of BRD2/4 over BRD3 protein.

Originality and significance: Overall, this study is of high significance for the field of induced proximity in general and molecule glue design more specifically. First and most importantly, this study uncovers a very novel mechanism for a new type of molecule glues, in contrast to more traditional molecule glue molecules such as lenalidomide and its analogues and indisulam and E7820, which engage with an E3 ligase first and then recruit neo-substrates for degradation. Second, the authors have convincingly demonstrated that the tandem BD1 and BD2 domains are necessary and sufficient for induced degradation by compound 1. Third, the authors have elucidated the residues responsible for the degradation selectivity for BRD2/4 over BRD3.

Data & methodology: The data presented in this study are of very high quality in nature. The authors use a large number of cutting-edge technologies and methods to carry out an elegant and thorough investigation. The presentation is very concise and clear.

Appropriate use of statistics and treatment of uncertainties: Appropriate.

Conclusions: Overall, the conclusions are very solid.

Suggested improvements: In the study, the authors initially investigated GNE-0011, together with compound 1. The authors showed that these two molecule glues have similar but somewhat different mechanisms of actions in inducing BRD2/4 degradation. Consistently, GNE-0011 should not be able to engage currently with both BD1 and BD2 domains based upon its chemical structure. It will be very interesting for the authors to further investigate or at least further discuss what are the proposed mechanism of action for GNE-0011.

In Figure 2f, while DCAF16 knockout has no effect for the cell growth inhibition activity of dBET6, consistent with its independence of DCF16. In comparison, DCAF16 knockout has a very large reduction for the cell growth inhibition activity of compound 1. However, DCAF16 knockout has very modest impact on the activity of GNE-0011, which is very surprising. Please discuss the major difference between compound 1 and GNE-0011.

In Figure 3d, the authors showed that in the TR-FRET proximity assay, while K_d value for BRD4 BD1-Bd2 (Tandem) and DCF16 is 1 μ M, addition of compound 1 only improves the K_d value to 700 nM, a very minimal enhancement. This piece of data is very different from the alphaLISA displacement assay (Figure 3c). Please discuss the discrepancy between these two different assays.

References: appropriate.

Clarity and context: The paper is well written, concise and clear.

Author Rebuttals to Initial Comments:

Referees' comments:

Summary of major updates since the pre-print

- Expansion of the intramolecular bivalent glue concept to a second system, reinforcing generality: Discovery and comprehensive cellular and *in vitro* characterization of a distinct, BRD4-selective intramolecular bivalent glue degrader (IBG4) which utilizes the same molecular mechanism but glues via a different and structurally unrelated E3 ligase (CRL4^{DCAF11}) that also intrinsically interact with BRD4.
- Rational, structure-guided design of improved intramolecular glues consisting of double-JQ1 compounds (IBG2 and IBG3) with remarkable potency reaching <10 pM DC₅₀ values, surpassing potencies reported in the literature for PROTACs or canonical molecular glues and highlighting the real-world potential for this new gluing modality to deliver lead-quality therapeutic compounds.
- Higher resolution Cryo-EM structure (3.77Å) and fully built model of the ternary complex including DCAF16, allowing the identification of key residues involved in DCAF16:BRD4 and intra-bromodomain interfaces.
- Additional biophysical experiments characterizing the mechanism of action of IBG1 (formerly compound **1**) via ITC, TR-FRET and NanoBRET bromodomain conformation biosensor.
- Rescue of degradation via reporter system for BRD3^{Tandem} using E -> G mutation.

*A note to the editor and reviewers: In the revised manuscript and our response to the reviewer comments, we have renamed compound **1** to IBG1 to better highlight the novel mechanism of intramolecular bivalent glue degraders, and refer to newly introduced compounds of this class as IBG2, IBG3 and IBG4. Truncations of IBG1, formerly named compounds **2a-d** and **3a-c** are now compounds **1a-d** and **1e-g** and the bumped version of IBG1 utilized in BromoTag approaches is now called bIBG1.*

*The original comments from the reviewers are in **black**, and our responses are in blue.*

Referee #1 (Remarks to the Author):

In this manuscript, Hsia O. et al. report the mechanism of action of a BRD4 degrader, which is originally designed as a DCAF15-targeting PROTAC but unexpectedly acts as a DCAF16-targeting molecular glue. A unique feature of this PROTAC-like glue is that it seems to stabilize interdomain interaction within the BRD4 tandem bromodomains while leveraging an intrinsic affinity between DCAF16 and BRD4. The study is remarkably complete, ranging from the initial characterization of the compound of interest, deconvolution of the E3 ligase, biophysical analysis of the interacting partners, to final determination of the structural mechanism. The presentation is clear, logical, and concise.

Major points:

1. Figure 3d. The affinity difference between BRD4 tandem BDs and DCAF16 +/- compound 1 is surprisingly small (0.7 vs 1 μ M). If the activity of the compound is to be explained by its ability to enhance the E3-substrate affinity, the authors need to validate this subtle change using an orthogonal method (ITC, MST, AlphaLisa displacement etc.).
2. If the affinity difference is indeed subtle, the authors should be positioned to determine the structure of the BRD4-DCAF16 complex in the absence of compound 1. The authors argue that the activity of the compound might be attributable to its ability to stabilize the structural configuration of BRD4 BD1-BD2 in complex with DCAF16. It is unclear whether such a mechanism leads to an enhanced affinity between the two proteins or a preferred orientation of the two BRD4 BDs on DCAF16 for ubiquitination. Structural characterization of the compound-free protein complex structure might shed light on this.

We thank the reviewer for raising these important points. Regarding the nature of the DCAF16:BRD4 interaction in the absence of compound, our efforts to structurally resolve a DCAF16:BRD4 complex in the absence of IBG1 (formerly compound 1) have been unsuccessful. We prepared many cryo-EM grids with varying ratios of DCAF16:DDB1 Δ BPB:DDA1 to BRD4^{Tandem} construct with BRD4 in molar excess to maximise chances of obtaining the complex. Unfortunately, we could only ever see either DCAF16:DDB1 Δ BPB particles or empty DDB1 Δ BPB particles.

Given the inability to structurally characterise the intrinsic protein-protein complex in the absence of degrader compound, we proceeded, as rightly suggested by the reviewer, to further validate and characterise the protein-protein complex in the absence and presence of compound via orthogonal biophysical assays:

Firstly, to validate the K_D value we obtained by TR-FRET, we performed ITC experiments titrating DCAF16 into a preformed complex of BRD4:IBG1 at a 1:1.1 molar ratio and obtained a highly comparable K_D value of ~600 nM (new Fig. 3a; Reviewer Fig. 1b). The improved intramolecular bivalent glue degrader that we generated via structure-guided design, IBG3, yielded a similar titration curve with a 3-fold enhanced binding affinity (K_D ~180 nM, new Fig. 4j, Reviewer Fig. 1c), consistent with its more potent cellular degradation activity. At similar protein concentrations, we detected a much smaller heat change for DCAF16 titrated into BRD4 in the absence of compounds (new Extended Data Fig. 3d, Reviewer Fig. 1a), and importantly observed an exothermic binding isotherm (ΔH negative, -8.2 kJ/mol) as opposed to endothermic in the presence of compounds (ΔH positive, +38 kJ/mol). The resulting binding

curve for the DCAF16:BRD4 intrinsic interaction yields a K_D value in the single-digit micromolar range ($K_D \sim 4 \mu\text{M}$), in overall agreement with our TR-FRET results ($K_D \sim 1 \mu\text{M}$). Overall, this data is consistent with a less specific or less stable protein-protein interaction in the absence compared to presence of IBG compounds, and suggests that the interaction between DCAF16 and BRD4 differs significantly when BRD4 is bound to IBG1/3.

We furthermore applied a distinct TR-FRET assay format, termed complex formation assay (new Fig. 4i, Reviewer Fig. 1d), that independently confirmed not only the compound-induced formation of a ternary complex, but also the enhanced gluing effect of the improved intramolecular bivalent glue IBG3, in line with more potent BRD4 degradation (new Extended Data Fig. 6b, Reviewer Fig. 1e)

So together, based on our previous data and the new data outlined above, we conclude that:

- 1) the IBGs increase the binding affinity of BRD4 for DCAF16 (TR-FRET, ITC),
- 2) DCAF16 increases the binding affinity of IBGs for BRD4 (alphaLISA), and
- 3) IBGs potently induce bromodomain dimerization, ternary complex formation, ubiquitination and degradation (SEC, NanoBRET BD dimerization assay, TR-FRET ternary complex formation assay, FACS stability reporter and HiBiT degradation).

In summary, IBGs stabilise an explicit and specific high-affinity arrangement of the tandem bromodomains within the ternary structure that underpins their potent degradation activity.

Reviewer Fig. 1.: Biochemical characterization of BRD4:DCAF16 interaction.

a-c, ITC data for DCAF16 titration into BRD4 (**a**), BRD4:IBG1 (**b**) and BRD4:IBG3 (**c**). *Data as in new Extended Data Fig. 3d, new Fig. 3a and new Fig. 4j, respectively.* **d**, TR-FRET ternary complex formation

assay. Anti-His-europium antibody bound to BRD4Tandem was incubated with equimolar Cy5-labelled DCAF16:DDB1 Δ BPB:DDA1 and increasing concentrations of IBG1, IBG3 or JQ1. *Data as in new Fig. 4i, e*, HiBiT-BRD4 degradation. HEK293 HiBiT knock-in cells were treated with IBG1 or IBG3 for 5 h and levels of BRD4-HiBiT proteins were quantified via luminescence. *Data as in new Extended Data Fig. 6b.*

Minor points:

1. The authors state that the resolution of the cryo-EM structure is about 4 angstrom. This might be true for DDB1. But the resolution is most likely much lower for DCAF16-BRD4. The maps shown in Fig 4 do not appear to have a resolution close to 4 angstrom. Please revise the statement and provide a figure showing the local resolution of the EM map.

We thank the reviewer for pointing this out. We have now further refined our structure and improved the quality of the map and the overall resolution to 3.77 Å (new Extended Data Fig. 4a, Reviewer Fig. 2a), which enabled the confident construction of an atomic model for DCAF16 (new Extended Data Fig. 5a, Reviewer Fig. 3). As suggested by the referee, we additionally now provide a Fourier Shell Correlation (FSC) curve of the refined EM map (new Extended Data Fig. 4b, Reviewer Fig. 2b) and local resolution estimation (new Extended Data Fig. 4c, Reviewer Fig. 2c), as well as angular distributions and posterior directional distribution plots for the final local refinement (new Extended Data Fig. 4d-e, Reviewer Fig. 2d-e).

Reviewer Fig. 2.: Cryo-EM data processing.

a, Workflow for Cryo-EM data processing. **b**, Gold-standard Fourier shell correlation at a cut-off of 0.143. **c**, Local resolution estimation on the unsharpened map. **d**, **e**, Angular distribution plot (**d**) and posterior position directional distribution plot (**e**) for the final local refinement. All panels as in new Extended Data Fig. 4.

2. The authors mention that the DCAF16 density in the EM map is not clear enough to build an atomic model. Is the density somewhat consistent with what AlphaFold predicts?

As mentioned above, based on our improved map we were able to confidently model DCAF16 using model-angelo, colabfold and manual building. In our model, DCAF16 adopts a unique fold consisting of 8 helices and several loops (new Extended Data Fig. 5a, Reviewer Fig. 3a), in a tertiary structure that differs significantly from the low to very low confidence AlphaFold prediction. A structure comparison search on the Dali server turns up no clear structural homologs, showing DCAF16 adopts a unique fold previously uncharacterised structurally and explaining the poor AlphaFold prediction. A preprint submitted back-to-back with our initial manuscript has in the meantime reported a cryo-EM structure of DCAF16 bound to isolated BRD4^{BD2}, stabilized by a covalent molecular glue degrader (Li, Ma, Hassan *et al.*, 2023, BioRxiv). This independently derived structure shows high complementarity to our own work, thus cross validating our model (RMSD for alignment of DCAF16 = 0.534 Å, Reviewer Fig. 3b).

Reviewer Figure 3.: Atomic model and comparison of DCAF16:BRD4 ternary structures.

a, Structure of DCAF16 coloured rainbow from N- to C-terminus, as in new Extended Data Fig. 5a. **b**, Overlay of ternary structure from this work (PDB code 8OV6, DDB1 blue, DCAF16 yellow, BRD4^{BD1} orange, BRD4^{BD2} red, IBG1 light grey) and ternary structure from Li, Ma, Hassan *et al.* (PDB code 8G46, all dark grey, structure does not include BRD4^{BD1}).

3. ED Figure 1e, Y axis is mislabeled?

We thank the reviewer for noticing this mistake. Indeed, the CellTiterGlo assays provided in old Extended Data Figure 1e quantifies cell viability, and we depict the curves normalized to DMSO. To improve clarity, we changed the axis label to ‘Viability (% normalized to DMSO)’ (new Extended Data Fig. 1c).

4. Page 7, “In alphaLISA displacement assays, we found significantly enhanced affinity of 1 to BRD4Tandem in the presence of DCAF16 (IC₅₀ = 12.8 nM), as compared to 1 and BRD4Tandem alone (IC₅₀ = 462 nM; cooperativity (α) = 36; Fig. 3c, Extended Data Fig. 4b), further supporting a role of compound 1 in stabilizing the BRD4:DCAF16 complex.” The last statement is not necessarily true if DCAF16 does not make direct contact with the compound. The most straightforward conclusion one can draw from the data is that DCAF16 stabilizes the BRD4-compound 1 complex.

We thank the reviewer for raising this point and believe the improved map and model of the ternary complex, together with the expanded biophysical characterisation presented in our revised manuscript allow us to clarify this. The refined structure shows that the JQ1 moiety as well as the aromatic linker of IBG1 indeed engage in multiple direct interactions with DCAF16 (new Fig. 4b, new Extended Data Fig. 5d). We thus believe that IBG1 not only stabilizes the relative arrangement of bromodomains, but also directly contributes to ternary complex stability. We hope that this model is outlined clearly in our revised manuscript.

5. Page 10, last paragraph. The data showing the degradation resistance of the two BRD4 constructs harbouring either two copies of BD1 or BD2 (Extended Data Fig. 5e) should be mentioned somewhere else in the text instead of right after the discussion of G386 and N93. If N93 is solvent exposed, its substitution with Gly should not have any impact. The fact that BD1/BD1 and BD2/BD2 are resistant to compound-induced degradation is most likely due to the importance of the BD1-BD2 interface and their interfaces with DCAF16.

We fully agree with this notion. As pointed out above, we believe that ternary complex formation requires an explicit arrangement of BRD4 bromodomains relative to each other to enable recognition by DCAF16 (or DCAF11). This is also supported by a complete lack of BET degradation observed for bivalent BET inhibitors, that also dimerize BET bromodomains but employ different linker length and architecture and thus likely induce distinct bromodomain arrangements (Reviewer Fig. 4a-c, new Extended Data Fig. 6i-j, new Extended Data Fig. 7b). We are thus fully aligned with the referee's suggestion, that resistance of BD1/BD1 and BD2/BD2 bromodomain tandems to IBG1 and IBG3-induced degradation is most likely due to an incompatibility of the bromodomain interface with the arrangement required for recognition by DCAF16. Indeed, compared to BD1, BD2 has a 3-residue insertion in the ZA loop that lies in the BD1:BD2 interface and might explain the observations described above. Even though space constraints did not allow us to discuss this in such detail in the main text, we hope that the revised manuscript nonetheless better captures these thoughts.

Reviewer Fig. 4.: Bivalent BET bromodomain inhibitors.

a, Structures of bivalent BET inhibitors MT1 and MS645. **b**, NanoBRET bromodomain dimerization assay. Indicated compounds were titrated into transiently expressed BRD4^{Nluc-Tandem-HaloTag} in HEK293 cells. *Data as in new Extended Data Fig. 7b*. **c**, HiBiT-BRD4 degradation activity of bivalent BET inhibitors MT1 and MS645 compared to IBG1 after treatment for 24 hours. *Data as in new Extended Data Fig. 6j*.

Referee #2 (Remarks to the Author):

This is a seminal paper by Ciulli and Winter's groups on the discovery of a sulfonamide-based BRD4 degrader that was previously thought to work through DCAF15 that this team discovered actually operates through DCAF16. They further demonstrate that the GNE-011 BRD4 molecular glue degrader also operates through DCAF16. The study also shows inherent affinity between BRD4 and DCAF16 which is further strengthened through the degraders. Interestingly, the authors show that the sulfonamide warhead itself does not bind to DCAF16 by itself and that the bromodomain alongside DCAF16 was required for the whole compound to bind to DCAF16. Structural, biochemical and genetic validation experiments all support their model.

Overall, this is an interesting study that not only demonstrates the mechanism of action of previously reported PROTAC/molecular glue degraders of BRD4 as operating through DCAF16, but also potentially reveals rational design strategies that one can take for future molecular glue degrader design beyond BRD4 through exploitation of existing weak protein-protein interactions and strengthening them through these types of molecular glue degraders. The studies described are rigorously performed and should be accepted for publication.

We thank the reviewer for the generous feedback and fully agree that the intramolecular bivalent glues we describe in our study represent a new type of degraders that might be broadly applicable to target proteins beyond BRD4. In our revised manuscript we now present an additional intramolecular bivalent glue chemotype, IBG4, that degrades BRD4 via a similar mode of action but engages the structurally unrelated E3 ligase DCAF11 (Reviewer Fig. 5a-c; new Fig. 5, new Extended Data Fig. 7). We hope that the referee agrees that this additional example clearly underpins the above notion of general applicability.

We also agree that the molecular and structural mechanism we uncover for DCAF16-dependent intramolecular bivalent glue degrader IBG1 (formerly compound **1**) provided the basis for the rational design of novel degraders, which we exemplify in our revised manuscript by the generation and characterization of two improved compounds, IBG2 and IBG3, that work via the same intramolecular bivalent glue mechanism and degrade BET proteins with potency in the low picomolar range (Reviewer Fig. 5d, new Fig. 4g-j, new Extended Data Fig. 6). We hope that together, these new examples and datasets further strengthen our manuscript and conclusions.

Reviewer Fig. 5.: Novel intramolecular bivalent glue degraders.

a, Bromodomain tandem requirement of IBG4. BRD4^{Tandem}, BD1 or BD2 dual fluorescence reporter KBM7 cells were treated for 6 hours with increasing concentrations of IBG4 and protein degradation was evaluated via flow cytometry. **b**, DCAF11 dependency of IBG4. AAVS1 control, DCAF11 or DCAF16 knockout KBM7 cells were treated with DMSO, IBG4 (100 nM) or IBG3 (0.1 nM) for 6 hours and BRD4 levels were analysed via immunoblotting. **c**, TR-FRET complex stabilization assay. 100 nM His-tagged BRD4^{Tandem} bound to anti-His europium antibody, was incubated with increasing concentrations of Cy5-labelled DCAF11:DDB1ΔBPB:DDA1 and 500 nM IBG4 or DMSO. **d**, HiBiT degradation assay. HEK293 HiBiT knock-in cells were treated with the indicated compounds for 5 h and levels of BRD4-HiBiT protein was quantified via luminescence.

Referee #3 (Remarks to the Author):

Summary of the key results:

In this study, Drs. Alessio Ciulli, Georg E. Winter and colleagues reported their extensive investigation of a molecule glue BRD2/4 degrader (compound 1) recently disclosed in patent publications (WO2021157684). Although compound 1 was synthesized by linking JQ1, a ligand for BET proteins and E7820, a ligand for DCF15, its induced BRD2/4 protein degradation is independent of DCF15. Interestingly, GNE-0011, a previously reported BRD2/4 degrader, also induced degradation of BRD2/4 with a similar but yet different mechanism of action. Of note, GNE-0011 has a propargylamine tail extending from the para-position of the phenyl ring and lacks the ligand for DCF15. Through CRISPR/Cas9 screen and other assays, the authors identified and confirmed that the induced degradation of BRD2/BRD4 by compound 1 and GNE-0011 depends upon DCAF16 and associated degradation machinery, in contrast to dBET6, which is a PROTAC BET degrader and depends upon cereblon/cullin4A for degradation. While compound 1 binds to DCF15 in vitro, it lacks the apparent affinity to DCAF16. In ternary complex formation assays, the authors showed that while compound 1 and GNE-0011 have no obvious interactions with DCF16, BRD4 and DCF16 have intrinsic interactions, which are further enhanced by both compound 1 and GNE-0011, albeit with different degrees. The authors further showed that presence of two tandem bromodomains (BD1 and BD2) are necessary and sufficient for induced BRD2/4 degradation by compound 1 and GNE-0011. The authors then determined the ternary complex formed between BRD4Tandem, 1 and DCAF16:DDB1ΔBPB at ~4 Å resolution by . Although the resolution is not very high, the structure clearly reviews that compound 1 binds to BD2 through its JQ-1 moiety and to BD1 through its indole-sulfonamide moiety to link the two domains and stabilize their relative conformation in a bidentate fashion. While compound 1 does not bind directly with DCF16, it facilitates a multivalent gluing interaction between the two bromodomains of BRD4 and DCAF16. Therefore, the authors have discovered a unique mode of action for a new type of molecule glues. Previously reported molecule glues first bind to an E3 ligase and then recruit neo-substrate proteins for degradation. In contrast, compound 1 binds concurrently to two different domains of BRD2/4 proteins, bringing these two domains together and stabilizing their conformations to recruit an E3 ligase (DCF16) for degradation. In addition, the authors have identified the residues responsible for the degradation selectivity of BRD2/4 over BRD3 protein.

Originality and significance: Overall, this study is of high significance for the field of induced proximity in general and molecule glue design more specifically. First and most importantly, this study uncovers a very novel mechanism for a new type of molecule glues, in contrast to more traditional molecule glue molecules such as lenalidomide and its analogues and indisulam and E7820, which engage with an E3 ligase first and then recruit neo-substrates for degradation. Second, the authors have convincingly demonstrated that the tandem BD1 and BD2 domains are necessary and sufficient for induced degradation by compound 1. Third, the authors have elucidated the residues responsible for the degradation selectivity for BRD2/4 over BRD3.

Data & methodology: The data presented in this study are of very high quality in nature. The authors use a large number of cutting-edge technologies and methods to carry out an elegant and thorough investigation. The presentation is very concise and clear.

Appropriate use of statistics and treatment of uncertainties: Appropriate.

Conclusions: Overall, the conclusions are very solid.

We thank the reviewer for the detailed summary and positive comments and similarly share the excitement about discovering a new and distinct modality for targeted protein degradation. We hope the reviewer agrees that our additional work on rational optimization of DCAF16-dependent intramolecular bivalent glue degraders as well as the characterization of a novel DCAF11-recruiting intramolecular glue will further generalize and strengthen these findings.

Suggested improvements: In the study, the authors initially investigated GNE-0011, together with compound 1. The authors showed that these two molecule glues have similar but somewhat different mechanisms of actions in inducing BRD2/4 degradation. Consistently, GNE-0011 should not be able to engage currently with both BD1 and BD2 domains based upon its chemical structure. It will be very interesting for the authors to further investigate or at least further discuss what are the proposed mechanism of action for GNE-0011.

We thank the referee for raising these valid concerns. While our observations of a critical involvement of DCAF16 in GNE-0011-based degradation of BRD4 has in the meantime been independently confirmed by other research groups (Shergalis et al., 2023, ACS Chem. Biol., Li, Ma, Hassan et al., 2023, BioRxiv), we agree that our work does not definitively solve the mode of action of this compound. Unfortunately, our extensive efforts to structurally determine the mode of action of GNE-0011 via cryo-EM were unsuccessful. We thus modelled GNE-0011 docked into BD1 and BD2 and overlaid the resulting models with the experimentally derived BRD4:IBG1:DCAF16 ternary complex. This suggested that two molecules of GNE-0011 can be accommodated in the structure, filling a space equivalent to one molecule of IBG1 (Reviewer Fig. 6a). We note that the positively charged propargylamine groups of both GNE-0011 molecules sit adjacent to one another, suggesting a less stable arrangement compared to IBG1, which would be in line with the lower degree of cooperativity observed in alphaLISA for GNE-0011 compared to IBG1 (Reviewer Fig. 6b) and the less potent BET degradation (Reviewer Fig. 6c, new Extended Data Fig. 2g). Nevertheless, we believe that in comparison to the meticulously characterized mode of action of IBG1, these data do not definitively resolve the mechanism of GNE-0011 and we thus decided to focus the revised manuscript on the structure-guided design of more potent intramolecular bivalent glue degraders (IBG2 and IBG3) in addition to characterizing a DCAF11-recruiting compound employing the same mechanism (IBG4), while deprioritizing GNE-0011, and hope this is in line with the reviewer's and editorial board's opinion.

Reviewer Fig. 6. GNE-0011.

a, Structural model of DCAF16-BRD4 ternary complex with two copies of GNE-0011 modelled (DCAF16 yellow, BD1 orange, BD2 red). **b**, alphaLISA displacement of biotinylated JQ1 probe from BRD4 by GNE-0011 ± DCAF16. **c**, BRD4 degradation efficiency. KBM7 cells were treated with increasing concentrations of IBG1 or GNE-0011 16 h and levels of BRD4, MYC, cleaved PARP1 and cleaved Caspase-3 were analysed via immunoblotting. Data in c as in new Extended Data Fig. 2g, Reviewer Fig. 7b.

In Figure 2f, while DCAF16 knockout has no effect for the cell growth inhibition activity of dBET6, consistent with its independence of DCF16. In comparison, DCAF16 knockout has a very large reduction for the cell growth inhibition activity of compound 1. However, DCAF16 knockout has very modest impact on the activity of GNE-0011, which is very surprising. Please discuss the major difference between compound 1 and GNE-0011.

We thank the reviewer for making this observation. Indeed, while DCAF16 knockout mediated resistance to both GNE-0011 and IBG1 (formerly compound 1), the effect was much more pronounced for IBG1 (IC₅₀ increase = 67-fold and 4-fold for IBG1 and GNE-0011, respectively). Nonetheless, both compounds also showed cytotoxicity in DCAF16 knockout cells, with almost identical IC₅₀ values (IC₅₀ = 25.2 nM and 23.7 for IBG1 and GNE-0011, respectively, Reviewer Fig. 7a). We hypothesized that the cytotoxicity observed in the absence of DCAF16 is mainly due to BET protein inhibitory activity of the JQ1 warheads contained in both compounds, while the increased killing observed in the presence of DCAF16 represents the added effect of BRD2 and BRD4 degradation.

To experimentally address this hypothesis, we quantified MYC protein levels as proxy for BET protein activity, as well as apoptosis induction as a phenotypic marker for BRD4 degradation via immunoblotting in DCAF16 wild-type and knockout cells after treatment with increasing concentrations of IBG1 and GNE-0011 (new Extended Data Fig. 2g, Reviewer Fig. 7b-c). As expected, in wild type cells both compounds potently induced BRD4 degradation, with IBG1 showing increased efficacy compared to GNE-0011, while neither compound affected BRD4 levels in DCAF16 knockout cells. In wild-type cells, MYC protein levels closely correlated with BRD4 degradation, and IBG1 resulted in decreased MYC expression at much lower concentrations than GNE-0011. In line with a DCAF16-independent inhibition of BET proteins, both compounds also induced MYC downregulation in DCAF16 knockout cells. However, in comparison to wild-type cells, both compounds showed similar potency in the absence of DCAF16, in line with the near-identical IC₅₀ values observed in this context.

Apoptosis, on the other hand, was only potently induced by IBG1 and only in WT cells, while GNE-0011 or IBG1 treatment in DCAF16 knockout cells did not induce relevant levels of Caspase-3 or PARP1 cleavage. Taken together, these data show that the increased cytotoxicity of IBG1 compared to GNE-0011 can be attributed to apoptotic cell death that goes hand in hand with potent DCAF16-dependent BRD2/4 degradation, while in the absence of DCAF16 both compounds elicit comparable levels of BET inhibition, causing MYC downregulation and growth inhibition.

Even though we have deprioritized GNE-0011 in our updated manuscript, as stated above, we nonetheless included data on DCAF16-dependent induction of apoptosis and DCAF16-independent BET inhibition in the manuscript as new Fig. 2f and new Extended Data Fig. 2g.

Reviewer Fig. 7.: DCAF16-dependent and independent effects.

a, Viability assay. KBM7 iCas9 WT or DCAF16 knockout cells were treated with increasing doses of IBG1 or GNE-0011 for 72 hours and cell viability was evaluated by CellTiterGlo assay. Dose-response curves fitted using non-linear regression. *Data for IBG1 as in new Fig. 2g.* **b-c**, Apoptosis induction and MYC downregulation. KBM7 iCas9 WT (**b**) or DCAF16 knockout cells (**c**) were treated with increasing concentrations of IBG1, GNE-0011 or dBET6 for 16 h and levels of BRD4, MYC, cleaved PARP1 and cleaved Caspase-3 were analysed via immunoblotting. *Data for b, c as new Extended Data Fig. 2g*

In Figure 3d, the authors showed that in the TR-FRET proximity assay, while K_D value for BRD4 BD1-Bd2 (Tandem) and DCAF16 is 1 μM , addition of compound 1 only improves the K_D value to 700 nM, a very minimal enhancement. This piece of data is very different from the alphaLISA displacement assay (Figure 3c). Please discuss the discrepancy between these two different assays.

We thank the reviewer for raising these points.

The first point about the apparent minimal enhancement of affinity in TR-FRET between the PPI in the presence versus absence of compound is as raised by Reviewer 1 and addressed above. As per above, to validate the K_D value we obtained by TR-FRET, we performed ITC experiments titrating DCAF16 into a preformed complex of BRD4:IBG1 at a 1:1.1 molar ratio and obtained a highly comparable K_D value of ~600 nM (new Fig. 3a; Reviewer Fig. 1b). The improved intramolecular bivalent glue degrader that we generated via structure-guided design, IBG3, yielded a similar titration curve with a 3-fold enhanced binding affinity (K_D ~180 nM, new Fig. 4j, Reviewer Fig. 1c), consistent with its more potent cellular degradation activity. At similar protein concentrations, we detected a much smaller heat change for DCAF16 titrated into BRD4 in the absence of compounds (new Extended Data Fig. 3d, Reviewer Fig. 1a), and importantly observed an exothermic binding isotherm (ΔH negative, -8.2 kJ/mol) as opposed to endothermic in the presence of compounds (ΔH positive, +38 kJ/mol). The resulting binding curve for the DCAF16:BRD4 intrinsic interaction yields a K_D value in the single-digit micromolar range (K_D ~4 μM), in overall agreement with our TR-FRET results (K_D ~1 μM). Overall, this data is consistent with a less specific or less stable protein-protein interaction in the absence compared to presence of IBG compounds, and suggests that the interaction between DCAF16 and BRD4 differs significantly when BRD4 is bound to IBG1/3.

We furthermore applied a distinct TR-FRET assay format, termed complex formation assay (new Fig. 4i, Reviewer Fig. 1d), that independently confirmed not only the compound-induced formation of a ternary complex, but also the enhanced gluing effect of the improved intramolecular bivalent glue IBG3, in line with more potent BRD4 degradation (new Extended Data Fig. 6b, Reviewer Fig. 1e)

So together, based on our previous data and the new data outlined above, we conclude that:

- 1) the IBGs increase the binding affinity of BRD4 for DCAF16 (TR-FRET, ITC),
- 2) DCAF16 increases the binding affinity of IBGs for BRD4 (alphaLISA), and
- 3) IBGs potently induce bromodomain dimerization, ternary complex formation, ubiquitination and degradation (SEC, NanoBRET BD dimerization assay, TR-FRET ternary complex formation assay, FACS stability reporter and HiBiT degradation).

In summary, IBGs stabilise an explicit and specific high-affinity arrangement of the tandem bromodomains within the ternary structure that underpins their potent degradation activity.

With regards to the second point raised, we understand the reviewer here is referring to the apparent discrepancy in cooperativity values measured by the TR-FRET and alphaLISA assays, respectively. We agree that this difference seems initially confusing. As the reviewer points out, the cooperativity (α) value measured in the TR-FRET complex stabilisation assay (new Fig. 3c; Reviewer Fig. 8a) was ~1.4, whereas IBG1 showed a very strong effect on probe displacement in the alphaLISA displacement assay (new Extended Data Fig. 3f; Reviewer Fig. 8b), with cooperativity α ~ 40.

These two cooperativity values relate to different binding events and equilibria:

- 1) TR-FRET: the affinity of DCAF16 for BRD4 in the presence or absence of IBG1 ($\alpha = K_D$ of DCAF16 binding BRD4 divided by the K_D of DCAF16 binding BRD4:IBG1)
- 2) alphaLISA: the affinity of IBG1 for BRD4 in the presence or absence of DCAF16 as measured by probe displacement from BRD4 ($\alpha =$ the affinity (IC_{50}) of IBG1 for BRD4 divided

by the affinity of IBG1 for BRD4 in the presence of saturating DCAF16). Thus, our TR-FRET and alphaLISA assays measure the cooperativity of different binding events.

Furthermore, in a modified version of the alphaLISA assay we measured displacement of the biotinylated JQ1 probe from BRD4 by titration of DCAF16 alone with an $IC_{50} = 3 \mu\text{M}$ and maximal displacement of $\sim 50\%$ of the probe (Reviewer Fig. 8c). This intrinsic affinity resulting in probe displacement may inflate the cooperativity in the alphaLISA, as compared to traditional PROTAC alphaLISA displacement assays where the two proteins (e.g. VCB and BRD4) have no intrinsic affinity and no intrinsic displacement of the probe. To investigate this further, we compared the cooperativity using different concentrations of DCAF16 and observed no major difference in the cooperativity using DCAF16 at $1 \mu\text{M}$ through to 10 nM (Reviewer Fig. 8d). We thus conclude that the larger cooperativity measured in alphaLISA compared to TR-FRET is reflecting of the two different binding assay set-ups.

We hope that these data sufficiently address the reviewer's pertinent remarks.

Reviewer Fig. 8.: alphaLISA Cooperativity.

a, TR-FRET complex stabilization assay. 200 nM His-tagged $\text{BRD4}^{\text{Tandem}}$ or BRD4^{BD1} , bound to anti-His-europium antibody, were incubated with increasing concentrations of Cy5-labelled DCAF16:DDB1 Δ BPB:DDA1 complex in the presence or absence of $1 \mu\text{M}$ IBG1 **b**, alphaLISA displacement assay. His- $\text{BRD4}^{\text{Tandem}}$ or His- BRD4^{BD1} were preincubated with a biotinylated JQ1 probe and titrated against increasing concentrations of IBG1 in the presence or absence of DCAF16. **c**, Intrinsic affinity of DCAF16 for BRD4. **d**, Effect of DCAF16 concentration on cooperativity. Assay was performed using a new batch of DCAF16:DDB1 Δ BPB:DDA1 but under the same condition as reported in the original manuscript.

References: appropriate.

Clarity and context: The paper is well written, concise and clear.

Reviewer Reports on the First Revision:

Referees' comments:

Referee #1 (Remarks to the Author):

In this revision, the authors have included more data to address the concerns this reviewer raised in the last round of review. Although the authors still could not resolve the structure of compound-free BRD4-DCAF16 complex, the technical difficulty could be explained by the relatively weak intrinsic affinity between the two proteins, which has now been determined to be around 4 μ M by ITC. The near 10-fold affinity difference between the compound-free and compound-enhanced PPI makes more sense. The improved map quality and the more comprehensive info on cryo-EM data processing are also highly appreciated. This reviewer supports the immediate publication of this manuscript in Nature.

Referee #3 (Remarks to the Author):

In the revised manuscript, the authors have done an outstanding job in addressing all the points raised by three reviewers. Specifically, the authors have refined the Cryo-EM structure of BRD4 (BD1-BD2):IBG1:DCAF16 to a high resolution (3.77 Å), allowing them to identify key residues involved in DCAF16:BRD4 and intra-bromodomain interfaces. Importantly, the authors have discovered a distinct, BRD4-selective intramolecular bivalent glue degrader (IBG4), which utilizes the same molecular mechanism but glues via a different and structurally unrelated E3 ligase (CRL4DCAF11). Furthermore, the authors have designed IBG2 and IBG3 as new analogs of IBG1, which are capable of inducing degradation of BRD4 with pico-molar potencies. Moreover, additional experiments have been performed to ambiguously define the cellular MOA of this type of molecule glues.

Overall, the revised manuscript is of very high quality and has a broad scientific impact for the field of chemical biology and therapeutic discovery. I fully support the publication of this revised manuscript in Nature.

Referee #4 (Remarks to the Author):

Note: the following review was requested after an initial round of reviews was completed. The comments provided below are directed towards the revised manuscript submitted by the authors following the initial reviews.

In the revised manuscript by Hsia et. al., the authors elucidate the mechanism for BRD2/4 degradation by the recently patented compound IBG1. The bifunctional IBG1 contains an E7820 sulfonamide motif that binds DCAF15, suggesting a PROTAC-like mechanism for degradation. Surprisingly, the authors find that DCAF15 is not responsible for BRD2/4 degradation and employ a myriad of techniques, including genetic, biochemical, biophysical, and cell-based methods, to conclusively identify DCAF16 as the primary effector of degradation. Furthermore, during the course of these experiments, the authors show that the E7820 sulfonamide motif does not bind directly to DCAF16 in a PROTAC-like intermolecular manor, but instead binds to the tandem bromodomain of BRD4 (BD1) to form an intramolecular bridging interaction. This bridging interaction appears to stabilize the conformation of the two bromodomains to produce an unexpected molecular glue-like interaction with DCAF16.

The molecular interactions underpinning this unique "intramolecular bivalent glue" (IBG) mechanism are detailed with a well-resolved cryo-EM structure (3.77 Å) of the tandem bromodomains of BRD4 bound to DCAF16:DDB1, along with the complex-stabilizing IBG1. The

authors exploit this structural information to generate two new molecules, IBG2 and IBG3, that form a stronger bridge between the tandem bromodomains of BRD2/4. Using this approach, the authors identify novel IBG degraders with exquisite potency (Fig. 4h), demonstrating the value of the structural and mechanistic insights uncovered by their work.

Finally, the authors identify an additional JQ1 derivative from the patent literature, IBG4, and show that this compound acts via a similar mechanism. However, in yet another surprising finding, the authors show that DCAF16 is not responsible for IBG4-induced degradation. Repeating the suite of genetic, biochemical, and cell-based assays described above, the authors identified DCAF11 as the key substrate adapter responsible for IBG4-induced degradation.

Main Points:

Overall, the findings reported in this manuscript are highly interesting, novel, often surprising, and clearly described. The authors provide exceptional experimental evidence to support their claims, with multiple orthogonal assays corroborating each finding. The breadth and quality of work reported here is impressive.

Although the general concept of molecular glues is well established, there are currently no examples of bivalent glues that bridge two domains of a target protein and thereby stabilize a conformation that results in new functional protein-protein interactions (here, ubiquitin transfer, but could extend to other functions). As such, this work constitutes a new subclass of molecule glues which could provide new opportunities for the TPD community.

The main shortcoming of this manuscript is that the IBG mechanism for degradation is only exemplified for BRD2/4 targets. In the final paragraph of the discussion, the authors state that the mechanism "could outline a generalizable strategy" for targeted protein degradation. However, this could also represent an isolated example that is poorly generalizable, given that BRD2/4 are highly primed for degradation via multiple pathways. The authors do not make exaggerated claims of generality throughout the manuscript, but this closing proposal should include a clause indicating that additional examples need to be identified before the generality of the mechanism can be evaluated. Indeed, it seems likely that this will be a more restricted approach to protein degradation than a broadly applicable modality. Nevertheless, this landmark paper will provide the community with a high-quality example of the tools and techniques available to characterize additional IBGs.

Minor Points:

- i) The legend of Fig. 4 is missing a description for panel (c) and the descriptions that follow are therefore offset and do not match the figure panels.
- ii) The ITC plots in several figures look very atypical. For example, each injection in Fig. 3a contains both an exo- and endothermic phase? Fig. 4j transitions from endothermic to exothermic? The authors should include a discussion of the ITC results to clarify these data. It would also be helpful to include a discussion of the dramatic differences observed between ITC titrations for DCAF16 into BRD4 with and without IBG1 (Fig. 3a vs Extended Fig. 3d). Why does the addition of compound shift complex assembly from exothermic to endothermic ($\Delta H = -8.2$ without compound; $\Delta H = 38.2$ with compound)?
- iii) Given the similarity in names, it is tempting for readers to assume that DCAF15 / 16 / 11 must show some sequence or structural similarity, which minimizes the novelty of the authors findings. The authors do state that these are unrelated substrate adapters, but it may be helpful to further emphasize just how different these proteins are. Extended Fig. 5b compares available DDB1 substrate adapter structures, but the overlays make this figure difficult to understand without detailed examination. The figure would be clearer if the DCAF16 structure was shown first, followed by the relevant comparison structures. Labeling of protein components could also help.

Author Rebuttals to First Revision:

Summary of changes:

- The Discussion section has been shortened and the generality of IBGs has been qualified further, in-line with Reviewer 4's points raised.
- The main text section describing the ITC results has been revised to address one of the points raised by Reviewer 4.
- Extended Data Figure 5b has now been changed to address Reviewer 4's comment.
- Figure 4 legend has been fixed.
- Chemistry synthesis methods have been moved to the Supplementary Information document.
- Section sub-headings have been shortened.
- The main text has been shortened without changing the narrative.
- Figures have been slightly re-worked to reduce the overall dimensions.

Referees' comments:

Referee #1 (Remarks to the Author):

In this revision, the authors have included more data to address the concerns this reviewer raised in the last round of review. Although the authors still could not resolve the structure of compound-free BRD4-DCAF16 complex, the technical difficulty could be explained by the relatively weak intrinsic affinity between the two proteins, which has now been determined to be around 4 μ M by ITC. The near 10-fold affinity difference between the compound-free and compound-enhanced PPI makes more sense. The improved map quality and the more comprehensive info on cryo-EM data processing are also highly appreciated. This reviewer supports the immediate publication of this manuscript in Nature.

Response:

We thank the referee for their vote of support! We agree that the visualization of a compound-free structure for the intrinsic DCAF16:BRD4 interaction would have been desirable but the failure to do so is in alignment with the biological and biophysical data presented in our manuscript consistent with it being a poorly stable complex that is challenging to structural studies. Gaining low-resolution insights in this intrinsic interaction will be an important focus of our future studies. We once again want to thank the reviewer for their valuable input during revisions and for their support of our manuscript.

Referee #3 (Remarks to the Author):

In the revised manuscript, the authors have done an outstanding job in addressing all the points raised by three reviewers. Specifically, the authors have refined the Cryo-EM structure of BRD4 (BD1-BD2):IBG1:DCAF16 to a high resolution (3.77 Å), allowing them to identify key residues involved in DCAF16:BRD4 and intra-bromodomain interfaces. Importantly, the authors have discovered a distinct, BRD4-selective intramolecular bivalent glue degrader (IBG4), which utilizes the same molecular mechanism but glues via a different and structurally unrelated E3 ligase (CRL4DCAF11). Furthermore, the authors have designed IBG2 and IBG3 as new analogs of IBG1, which are capable of inducing degradation of BRD4 with pico-molar potencies. Moreover, additional experiments have been performed to ambiguously define the cellular MOA of this type of molecule glues. Overall, the revised manuscript is of very high quality and has a broad scientific impact for the field of chemical biology and therapeutic discovery. I fully support the publication of this revised manuscript in Nature.

Response:

We thank the reviewer for their overwhelmingly positive feedback and are happy that they agree that the additional data added during revisions have improved our manuscript.

Referee #4 (Remarks to the Author):

Note: the following review was requested after an initial round of reviews was completed. The comments provided below are directed towards the revised manuscript submitted by the authors following the initial reviews.

In the revised manuscript by Hsia et. al., the authors elucidate the mechanism for BRD2/4 degradation by the recently patented compound IBG1. The bifunctional IBG1 contains an E7820 sulfonamide motif that binds DCAF15, suggesting a PROTAC-like mechanism for degradation. Surprisingly, the authors find that DCAF15 is not responsible for BRD2/4 degradation and employ a myriad of techniques, including genetic, biochemical, biophysical, and cell-based methods, to conclusively identify DCAF16 as the primary effector of degradation. Furthermore, during the course of these experiments, the authors show that the E7820 sulfonamide motif does not bind directly to DCAF16 in a PROTAC-like intermolecular manner, but instead binds to the tandem bromodomain of BRD4 (BD1) to form an intramolecular bridging interaction. This bridging interaction appears to stabilize the conformation of the two bromodomains to produce an unexpected molecular glue-like interaction with DCAF16.

The molecular interactions underpinning this unique “intramolecular bivalent glue” (IBG) mechanism are detailed with a well-resolved cryo-EM structure (3.77 Å) of the tandem bromodomains of BRD4 bound to DCAF16:DDB1, along with the complex-stabilizing IBG1. The authors exploit this structural information to generate two new molecules, IBG2 and IBG3, that form a stronger bridge between the tandem bromodomains of BRD2/4. Using this approach, the authors identify novel IBG degraders with exquisite potency (Fig. 4h), demonstrating the value of the structural and mechanistic insights uncovered by their work.

Finally, the authors identify an additional JQ1 derivative from the patent literature, IBG4, and show that this compound acts via a similar mechanism. However, in yet another surprising finding, the authors show that DCAF16 is not responsible for IBG4-induced degradation. Repeating the suite of genetic, biochemical, and cell-based assays described above, the authors identified DCAF11 as the key substrate adapter responsible for IBG4-induced degradation.

Main Points:

Overall, the findings reported in this manuscript are highly interesting, novel, often surprising, and clearly described. The authors provide exceptional experimental evidence to support their claims, with multiple orthogonal assays corroborating each finding. The breadth and quality of work reported here is impressive.

Although the general concept of molecular glues is well established, there are currently no examples of bivalent glues that bridge two domains of a target protein and thereby stabilize a conformation that results in new functional protein-protein interactions (here, ubiquitin transfer, but could extend to other functions). As such, this work constitutes a new subclass of molecule glues which could provide new opportunities for the TPD community.

The main shortcoming of this manuscript is that the IBG mechanism for degradation is only exemplified for BRD2/4 targets. In the final paragraph of the discussion, the authors state that the mechanism “could outline a generalizable strategy” for targeted protein degradation. However, this could also represent an isolated example that is poorly generalizable, given that BRD2/4 are highly primed for degradation via multiple pathways. The authors do not make exaggerated claims of generality throughout the manuscript, but this closing proposal

should include a clause indicating that additional examples need to be identified before the generality of the mechanism can be evaluated. Indeed, it seems likely that this will be a more restricted approach to protein degradation than a broadly applicable modality. Nevertheless, this landmark paper will provide the community with a high-quality example of the tools and techniques available to characterize additional IBGs.

We thank the reviewer for their kind remarks regarding the quality of our work and manuscript.

We have now reworked the Discussion to better reflect the Reviewer's remark that applications to other targets beyond BET proteins, but also other modalities besides induced protein degradation will be important future goals for solidifying the generality of this new mechanism. As the Reviewer points out, this novel category of molecular glues may represent a new and exciting avenue for the field of induced-proximity pharmacology, and we are happy that the reviewer shares our excitement that this work provides a strong blueprint for the discovery and design of other intramolecular bivalent glues.

Specifically, we have revised the final paragraph of the manuscript as follows:

"In conclusion, we show that structurally distinct BET degraders converge on a shared novel mechanism of action: intramolecular dimerization of two domains to modify protein surface and modulate protein-protein-interactions. This concept is so far limited to degradation of a single target protein family and generalizability to other targets remains to be shown. However, protein surface modulation via intramolecular, chemical bridging of binding sites in cis could outline a strategy to pharmacologically utilize intrinsic interactions with diverse effector proteins and rewire cellular circuits beyond protein degradation."

Minor Points:

i) The legend of Fig. 4 is missing a description for panel (c) and the descriptions that follow are therefore offset and do not match the figure panels.

We thank the referee for pointing out this error and have amended the legend.

ii) The ITC plots in several figures look very atypical. For example, each injection in Fig. 3a contains both an exo- and endothermic phase? Fig. 4j transitions from endothermic to exothermic? The authors should include a discussion of the ITC results to clarify these data.

The referee correctly points out that there is both negative DP (exothermic) and positive DP (endothermic) contributions to the heat observed for injections in the binding isotherms shown in Fig. 3a and Fig. 4j for DCAF16 titration into BRD4^{Tandem} bound to IBG1 and IBG3, respectively. In these titrations, the positive DP contribution is due to the absorbed (endothermic) heat of binding of DCAF16 to the BRD4^{Tandem}:IBG1/3 complex. In contrast, the negative DP contribution is due to the released (exothermic) heat of dilution of DCAF16 into the buffer. In these titrations, the heat of dilution (negative peak) is largely or completely offset by the heat of binding (positive peak) for each of the injections until the binding is saturated towards the end of the curve, where we only see the negative peak corresponding to the heat of dilution as the binding to form the ternary complex has reached saturation. The observed transition from apparent endothermic to apparent exothermic in Fig. 4j exactly reflects this.

From our extensive experience with performing ITC experiments, these kinds of plots exhibiting competing heat phases (i.e. endothermic heat of binding plus exothermic heat of

dilutions, or *vice versa*) are seen quite often and indeed we have observed these types of behaviours in titrations included in many of our previous manuscripts (see PMIDs: 23872845, 25505247, and 28288108 for representative examples).

It would also be helpful to include a discussion of the dramatic differences observed between ITC titrations for DCAF16 into BRD4 with and without IBG1 (Fig. 3a vs Extended Fig. 3d). Why does the addition of compound shift complex assembly from exothermic to endothermic ($\Delta H = -8.2$ without compound; $\Delta H = 38.2$ with compound)?

This is because here we are looking at two different complexes: 1) the intrinsic interaction between DCAF16 and BRD4^{Tandem} and 2) the interaction between DCAF16 and the BRD4^{Tandem}:IBG1/3 complex. As discussed in previous comments from the other referees, the exact nature of the intrinsic interaction between BRD4^{Tandem} and DCAF16 remains unclear. What is clear is that the species present in the ITC cell, i.e. BRD4^{Tandem} apo and BRD4^{Tandem}:IBG1/3 are significantly different, as also evidenced by their distinct SEC traces. Thus, the large difference in heat of binding, one being exothermic while the other one being endothermic, is not necessarily surprising, as we are measuring a very different binding event in the presence of compound, which intramolecularly dimerises the bromodomains of BRD4. The comparison of the two ITC data in itself therefore points not only to the interaction being significantly strengthened (i.e. “glued”) in the presence of the IBG compound (resulting in lower K_d value in the presence vs absence of compound), but also overall quite different (thus resulting in very different ΔH values). As suggested, we have included the following sentences in the revised manuscript to make this point:

“Comparison of the ITC titrations for DCAF16 into BRD4 apo or IBG1-bound reveals the interaction is not only strengthened (c.f. K_D of 0.6 vs 4 μ M) but also thermodynamically altered (c.f. ΔH of 38 vs -8 kJ/mol), suggesting a change in complex architecture.”

“Such a conformational change would also be in line with the thermodynamic shift from exo- to endothermic binding of BRD4 to DCAF16 in presence of IBG1 observed in ITC (Fig. 3a, Extended Data Fig. 3d).”

iii) Given the similarity in names, it is tempting for readers to assume that DCAF15 / 16 / 11 must show some sequence or structural similarity, which minimizes the novelty of the authors findings. The authors do state that these are unrelated substrate adapters, but it may be helpful to further emphasize just how different these proteins are. Extended Fig. 5b compares available DDB1 substrate adapter structures, but the overlays make this figure difficult to understand without detailed examination. The figure would be clearer if the DCAF16 structure was shown first, followed by the relevant comparison structures. Labeling of protein components could also help.

We thank the referee for this valuable suggestion and have amended Extended Data Fig. 5b to now show side-by-side comparisons of the structurally distinct DCAF family members, DCAF1 and DCAF15, as well as CRBN in comparison to DCAF16, in cartoon representation bound to DDB1 in surface representation and shown from identical orientations (see also Reviewer Fig. 1). Additionally, as DCAF11 is not yet structurally resolved, we provide an AlphaFold prediction for comparison in new Extended Data Figure 7g. We hope this addition helps to clarify the point raised and makes the figure more clearly interpretable.

Reviewer Fig. 1.: Comparison of DCAF16 and structurally distinct CRL4 substrate receptors DCAF1, DCAF15, CRBN (PDB entries 5JK7, 6UD7, and 5FQD, respectively) and DCAF11 (AlphaFold prediction) bound to DDB1 (blue).

Reviewer Reports on the Second Revision:

Referee #4 (Remarks to the Author):

****Please see attached Word document****
(figures are included which required attachment)

Response to Authors' Revision:

[Please note, the following is the attachment Referee 4 refers to on the previous page]

In this revised manuscript, the authors have addressed the main point raised during review regarding over-stating the generality of the approach. The addition of the following statement sufficiently qualifies the author's claims: "The concept is so far limited to degradation of a single target family and generalizability to other targets remains to be shown." Thank you for including this additional transparency.

Regarding the minor points, the revised Extended Data Fig. 5b makes the structural comparison much easier to visualize and clearly demonstrates how distinct the relevant substrate adapter proteins are – which enforces the novelty of the authors' findings.

Regarding the ITC data, we suggested that the authors elaborate on the dramatic changes observed in the thermodynamics of complex formation in the absence and presence of compounds. This behavior is unusual, interesting, and highly informative. In response, the authors added the following statement:

"Comparison of the ITC titrations for DCAF16 into BRD4 apo or IBG1-bound reveals the interaction is not only strengthened (c.f. K_D of 0.6 vs 4 μM) but also thermodynamically altered (c.f. ΔH of 38 vs -8 kJ/mol), suggesting a change in complex architecture."

However, the thermodynamic data indicates much more than a simple "change in complex architecture." Here is an overview of what the authors present for IBG1:

The DCAF16-BRD4 complex formation goes from slightly exothermic ($\Delta H = -8.17$ kJ/mol; ie. enthalpically favorable) to highly endothermic ($\Delta H = 38.2$ kJ/mol; ie. enthalpically unfavorable) with the addition of IBG1. If IBG1 strengthens the ternary complex, then this observation does not make sense – this should enhance the enthalpy of binding. Instead, the addition of compound makes complex formation enthalpically unfavorable, but this is more than compensated for by the dramatic increase in entropy ($-\Delta S = -22.5$ kJ/mol without compound; $-\Delta S = -73.9$ kJ/mol with compound). Forming a protein complex increases the order of the system and is thus entropically unfavorable, yet both ITC experiments found that complex formation was entropically favorable (ie. complex formation increases the disorder of the system). This can only mean one thing: *DCAF16-BRD4 complex formation buries hydrophobic surface area and releases water back to bulk solvent!*

Furthermore, the much larger entropy value in the presence of compound can be attributed to two factors. First, the entropic penalty for stabilizing the conformation of the bromodomains of BRD4 is paid for by compound binding prior to complex formation with DCAF16. Second, all the IBG compounds reported in this manuscript have highly hydrophobic biphenyl linkers that would be solvent exposed, further increasing the hydrophobic surface area that is buried upon complex formation.

Finally, all these observations drawn from the thermodynamic data, are clearly corroborated by the authors' structure, which shows a large hydrophobic surface area buried at the ternary complex interface (the authors do point out several hydrophobic features of the interface):

My point is that a careful examination of the thermodynamic data, corroborated by structural data, provides a great deal of information regarding the key factors driving complex formation. Since the novelty of this manuscript comes from the discovery of intramolecular bivalent glues, a thorough understanding of this mechanism seems important. The authors' data provide a great deal of information, but they have not highlighted these key findings. I believe this could be helpful for readers who may not be experts in ternary complex thermodynamics or may be confused by the exothermic to endothermic transition with IBG1 addition.

Finally, a quick and clear understanding of the ITC data is complicated by the ITC plot show in Fig. 3a, which has both exo- and endo-thermic phases. As the authors note in their response, the exothermic component comes from the heat of dilution (which I believe is true). However, heat of dilution is typically controlled for by performing injections into buffer and subtracting this signal from the complex formation experiment. It looks like this was done in Fig. 4j for IBG3 binding, which does not have these odd exo- and endo-thermic peaks and is much clearer. Since Fig.3a is a primary figure in a manuscript being considered for Nature, I believe this could be improved (made comparable to Fig.4j).

As stated previously, this is a very high-quality manuscript that includes a vast amount of data and many interesting findings. I support publication in Nature, but believe the discussion points outlined above would add to the clarity and accessibility of the manuscript to a broader audience.

Author Rebuttals to Second Revision:

Response to Authors' Revision:

In this revised manuscript, the authors have addressed the main point raised during review regarding over-stating the generality of the approach. The addition of the following statement sufficiently qualifies the author's claims: "*The concept is so far limited to degradation of a single target family and generalizability to other targets remains to be shown.*" Thank you for including this additional transparency.

We thank the referee for prompting us to add this important qualification.

Regarding the minor points, the revised Extended Data Fig. 5b makes the structural comparison much easier to visualize and clearly demonstrates how distinct the relevant substrate adapter proteins are – which enforces the novelty of the authors' findings.

We thank the referee for this suggested improvement to the manuscript and are happy that it now accentuates the novelty of our findings.

Regarding the ITC data, we suggested that the authors elaborate on the dramatic changes observed in the thermodynamics of complex formation in the absence and presence of compounds. This behavior is unusual, interesting, and highly informative. In response, the authors added the following statement:

"Comparison of the ITC titrations for DCAF16 into BRD4 apo or IBG1-bound reveals the interaction is not only strengthened (c.f. K_d of 0.6 vs 4 μM) but also thermodynamically altered (c.f. ΔH of 38 vs -8 kJ/mol), suggesting a change in complex architecture."

However, the thermodynamic data indicates much more than a simple "change in complex architecture." Here is an overview of what the authors present for IBG1:

The DCAF16-BRD4 complex formation goes from slightly exothermic ($\Delta H = -8.17$ kJ/mol; ie. enthalpically favorable) to highly endothermic ($\Delta H = 38.2$ kJ/mol; ie. enthalpically unfavorable) with the addition of IBG1. If IBG1 strengthens the ternary complex, then this observation does not make sense – this should enhance the enthalpy of binding. Instead, the addition of compound makes complex formation enthalpically unfavorable, but this is more than compensated for by the dramatic increase in entropy ($-T\Delta S = -22.5$ kJ/mol without compound; $-T\Delta S = -73.9$ kJ/mol with compound). Forming a protein complex increases the order of the system and is thus entropically unfavorable, yet both ITC experiments found that complex formation was entropically favorable (ie. complex formation increases the disorder of the system). This can only mean one thing: *DCAF16-BRD4 complex formation buries hydrophobic surface area and releases water back to bulk solvent!*

Furthermore, the much larger entropy value in the presence of compound can be attributed to two factors. First, the entropic penalty for stabilizing the conformation of the bromodomains of BRD4 is paid for by compound binding prior to complex formation with DCAF16. Second, all the IBG compounds reported in this manuscript have highly hydrophobic biphenyl linkers that would be solvent exposed, further increasing the hydrophobic surface area that is buried upon complex formation.

Finally, all these observations drawn from the thermodynamic data, are clearly corroborated by the authors' structure, which shows a large hydrophobic surface area buried at the ternary complex interface (the authors do point out several hydrophobic features of the interface):

My point is that a careful examination of the thermodynamic data, corroborated by structural data, provides a great deal of information regarding the key factors driving complex formation. Since the novelty of this manuscript comes from the discovery of intramolecular bivalent glues, a thorough understanding of this mechanism seems important. The authors' data provide a great deal of information, but they have not highlighted these key findings. I believe this could be helpful for readers who may not be experts in ternary complex thermodynamics or may be confused by the exothermic to endothermic transition with IBG1 addition.

Finally, a quick and clear understanding of the ITC data is complicated by the ITC plot show in Fig. 3a, which has both exo- and endo-thermic phases. As the authors note in their

response, the exothermic component comes from the heat of dilution (which I believe is true). However, heat of dilution is typically controlled for by performing injections into buffer and subtracting this signal from the complex formation experiment. It looks like this was done in Fig. 4j for IBG3 binding, which does not have these odd exo- and endo-thermic peaks and is much clearer. Since Fig.3a is a primary figure in a manuscript being considered for Nature, I believe this could be improved (made comparable to Fig.4j). As stated previously, this is a very high-quality manuscript that includes a vast amount of data and many interesting findings. I support publication in Nature, but believe the discussion points outlined above would add to the clarity and accessibility of the manuscript to a broader audience.

We thank the referee for the positive, detailed feedback and pertinent discussion points, and we similarly share this referee's enthusiasm for the ITC data! We have made the following changes to sections of the text which we hope will now provide more context in-line with the excellent points raised by the reviewer above:

“Comparison of the ITC titrations for DCAF16 into unbound vs IBG1-bound BRD4^{Tandem} reveals that IBG1 strengthens (c.f. K_D of 0.6 vs 4 μ M) and thermodynamically alters the BRD4:DCAF16 interaction. While IBG1 changes the binding from exothermic to endothermic (c.f. ΔH -8 vs 38 kJ/mol), this unfavourable enthalpy change is more than compensated for by a substantial change in the entropic term which becomes much more favourable in the presence of IBG1 (c.f. $T\Delta S$ 22.5 vs 73.9 kJ/mol). This enthalpy-entropy compensation, a well-known phenomenon in biological systems²³, leads to a greater binding energy in the presence vs absence of IBG1 (c.f. ΔG -35.7 vs -30.6 kJ/mol; resulting in a favourable $\Delta\Delta G$ of -5.1 kJ/mol). Together, these dramatic differences in thermodynamic behaviour are consistent with a different mode of DCAF16 binding for IBG1-bound compared to unbound BRD4^{Tandem}.”

“Such a conformational change would also explain the dramatic increase in entropy observed in ITC for BRD4 binding to DCAF16 in the presence of IBG1 (Fig. 3a, Extended Data Fig. 3d), as the entropic penalty for intramolecularly engaging and stabilizing the bromodomains is paid for by IBG1 binding prior to complex formation with DCAF16.”

“The ternary complex is further stabilized by intramolecular contacts between the two bromodomains including the sandwiching of M442 between W81 and P375 in the WPF shelves of BD1 and BD2, respectively (Extended Data Fig. 5g). This series of interactions buries a large hydrophobic surface area upon complex formation, which is consistent with the highly entropically favourable interaction of IBG1-bound BRD4^{Tandem} with DCAF16 (Fig. 3a).”

We thank the reviewer again for the vote of confidence in our manuscript and for supporting publication. We feel that the suggested changes have materially improved the quality and accessibility of our story.

The final version has been seen by the referee.